# *Glis2* is an early effector of polycystin signaling and a target for therapy in polycystic kidney disease

Chao Zhang [1,7], Michael Rehman[1,7], Xin Tian[1,7], Steven Lim Cho Pei [1], Jianlei Gu[2], Thomas A. Bell 3rd[3], Ke Dong [1], Ming Shen Tham [1], Yiqiang Cai [1], Zemeng Wei [1], Felix Behrens [1], Anton M. Jetten[4], Hongyu Zhao [2,5,6], Monkol Lek [5] & Stefan Somlo [1,5] ✉

Mouse models of autosomal dominant polycystic kidney disease (ADPKD) show that intact primary cilia are required for cyst growth following the inactivation of polycystin-1. The signaling pathways underlying this process, termed cilia-dependent cyst activation (CDCA), remain unknown. Using translating ribosome affinity purification RNASeq on mouse kidneys with polycystin-1 and cilia inactivation before cyst formation, we identify the differential 'CDCA pattern' translatome specifically dysregulated in kidney tubule cells destined to form cysts. From this, *Glis2* emerges as a candidate functional effector of polycystin signaling and CDCA. In vitro changes in *Glis2* expression mirror the polycystin- and cilia-dependent changes observed in kidney tissue, validating *Glis2* as a cell culture-based indicator of polycystin function related to cyst formation. Inactivation of *Glis2* suppresses polycystic kidney disease in mouse models of ADPKD, and pharmacological targeting of *Glis2* with antisense oligonucleotides slows disease progression. *Glis2* transcript and protein is a functional target of CDCA and a potential therapeutic target for treating ADPKD.

Autosomal dominant polycystic kidney disease (ADPKD) is a common genetic disease most often caused by mutations in the genes encoding polcysytin-1 (PC1) or polycystin-2 (PC2)[1]. ADPKD presents with expanding fluid-filled cysts arising from kidney tubules that enlarge and deform the kidney, usually over the course of decades[2]. Cysts grow by altering kidney tubule cell shape and function, taking on a secretory rather than resorptive phenotype while also remodeling the surrounding basement membrane and interstitium and undergoing low-level cyst cell proliferation. These changes are associated with increased inflammatory stimuli and fibrotic changes in the surrounding kidney tissue. ADPKD presents with several clinical symptoms including hypertension, chronic and acute pain, urinary infections, and kidney stones[2]. Cysts in the liver arising from the bile ducts are a common extrarenal manifestation while intracranial aneurysms are a less common but potentially catastrophic association that may have familial risk underpinnings[3,4]. In the kidney, ADPKD results in impaired kidney function requiring dialysis or transplantation by the fifth decade of life for about half of affected individuals[5]. While the disease is inherited as a dominant trait, cyst initiation requires somatic second hit mutations that further reduce or eliminate the functional dosage of the respective polycystin protein in kidney tubule or bile duct cells[6]. ADPKD can also manifest in individuals inheriting recessive

[1]Department of Internal Medicine, Yale School of Medicine, New Haven, CT, USA. [2]Department of Biostatistics, Yale University School of Public Health, New Haven, CT, USA. [3]Ionis Pharmaceuticals, Inc., Carlsbad, CA, USA. [4]Cell Biology Section, Immunity, Inflammation and Disease Laboratory, National Institute of Environmental Health Sciences, National Institutes of Health, Research Triangle Park, NC, USA. [5]Department of Genetics, Yale School of Medicine, New Haven, CT, USA. [6]Computational Biology and Bioinformatics Program, Yale University, New Haven, CT, USA. [7]These authors contributed equally: Chao Zhang, Michael Rehman, Xin Tian. ✉e-mail: stefan.somlo@yale.edu

hypomorphic alleles for *PKD1*[7]. The nature of the germline mutations affects the clinical severity of the disease with truncating mutations in *PKD1* typically having a severe course and minimally hypomorphic mutations in *PKD1* and most mutations in *PKD2* having a more indolent course[8,9]. However, the significant intrafamilial variation in severity of ADPKD indicates that additional factors beyond the inherited mutation have important roles in prognosis. Larger kidneys at an earlier age are at present the most accepted biomarker suggesting a more severe clinical course[9].

In more than two decades since the identification of *PKD1* and *PKD2*, there has been significant progress in understanding the functions of the polycystin proteins[10]. Several signaling pathways have been reported as aberrantly affected in ADPKD cystic epithelia including cyclic-AMP signaling[11,12], mammalian target of rapamycin (mTOR) signaling[13], G-protein coupled receptor signaling[14], and extracellular matrix signaling[5] to name a few. Several cellular and organ level processes including metabolic pathways, mitochondrial function and immune modulation have been found to be affected by polycystin function and to affect the course of ADPKD[5,14,15]. Despite, or perhaps because of, these pleiotropic findings there is a lack of consensus on the functional roles of the polycystins that are most directly related to the ADPKD phenotype in the kidney. There are, however, several areas of general agreement. There is consensus that in vivo mouse models based on *Pkd1* and *Pkd2*, the orthologs of the respective human disease genes, are the best laboratory-based models for validation of functional pathways and putative therapeutic targets for ADPKD. There is also agreement that polycystin proteins are localized to the membrane overlying the primary cilium, a solitary microtubule based cellular sensory organelle protruding from the apical luminal surface of almost all kidney tubule and cyst cells. Primary cilia play a central role in the pathogenesis of ADPKD[16–18]. The genetic relationship between cilia and polycystins in ADPKD was first defined in a mouse study showing that cyst growth following the loss of polycystins is suppressed by concomitant genetic removal of cilia[19]. Cyst growth was also suppressed in adult models of ADPKD following inactivation of *Tulp3*, which affects the membrane associated protein composition of cilia without dismantling the organelle entirely[20,21]. These series of findings have suggested the existence of a cilia-dependent cyst activating (CDCA) pathway that can be defined as signaling pathway that is regulated by the polycystin complex located in cilia and whose full function requires the presence of intact cilia[19,22]. CDCA is a putative signaling pathway that requires structurally and functionally intact primary cilia to drive cyst growth following inactivation of polycystin-1 or polycystin-2. The lack of convergence toward a unifying functional pathway in polycystin biology and the persistence of gaps in understanding of polycystin function in vivo suggest that critical components the polycystin-CDCA signaling cascades have not yet been identified. The molecular events most closely related to the primary function of polycystins in cilia are potentially the most effective and most specific targets for therapy in ADPKD.

The current study began with the premise that these gaps in understanding can be approached through unbiased discovery strategies followed by validation steps in mouse models based on orthologous genes. We based the discovery step on kidney tubule selective Translating Ribosome Affinity Purification (TRAP) transcriptional profiling[23–26]. We applied TRAP to models of polycystin and cilia inactivation in mouse kidneys and made use of the unique feature of mouse models that allows interrogation of genetically defined kidney tubule cell populations before the onset of cyst formation and its attendant confounding secondary non-cell autonomous biological effects. We identified a group of 167 transcripts shared in male and female mice whose expression in vivo is specifically altered in the same direction of change in *Pkd1* knockout kidney tubule cells destined to form cysts when compared to both wild type cells and cells with dual inactivation of *Pkd1* and cilia which are protected from cyst formation.

We refer to this pattern of gene dysregulation as the "CDCA pattern". We applied biologically informed selection criteria to further refine this list to 73 genes from which we identified 11 genes with the highest statistical significance and biological plausibility. We selected one candidate gene, *Glis2*, for biological validation. *Glis2* encodes one of three Gli–similar (Glis1-3) Krüppel-like zinc finger transcription factor proteins[27–29]. *Glis2* is most abundantly expressed in the kidney along the entire nephron[30–32] and is the causative gene for nephronophthisis type 7 (*NPHP7*)[33,34]. We found that *Glis2* transcript and protein expression changes in primary kidney cell cultures are an in vitro surrogate for in vivo gene expression changes associated with polycystin-dependent cyst formation. Inactivation of *Glis2* in early onset and adult mouse models of *Pkd1* and *Pkd2* resulted in suppression of cyst formation. Furthermore, pharmacological targeting of *Glis2* using antisense oligonucleotides suppressed cyst formation and associated secondary changes in the mouse kidney. In aggregate, we have developed a discovery platform for transcriptional changes related to polycystin function and discovered Glis2 as a potentially tractable therapeutic target for treatment of ADPKD.

## Results

### Translating ribosome affinity purification transcriptional profiling in early-stage cyst cells in vivo

We sought to discover early-stage changes in gene expression signatures that are uniquely associated with loss of polycystin-1 and cyst formation in vivo. To this end, we used TRAP to interrogate actively translating mRNA expression (the "translatome") in a cell-specific manner in mouse kidney tubules after they were genetically induced to form cysts, but at a time point before cysts formed. To achieve cell specificity for kidney tubules, we used the *Pax8*<sup>rtTA</sup>;*TetO*<sup>Cre</sup> digenic doxycycline inducible kidney selective system[19,35] to inactivate target genes and contemporaneously turn on expression of the L10a-EGFP ribosomal fusion protein used to isolate ribosomal complexes only from cells in which Cre recombinase had been active[23]. The adult induced Cre recombinase expression pattern and natural history of polycystic kidney disease progression in this model has been described[19,36]. To identify differentially expressed transcripts that are most specifically correlated with cyst formation, we combined analysis from mice with three genotypes: *Pkd1* single mutants (Pkd1<sup>KO</sup>) that are genetically destined to develop polycystic kidney disease[19], *Pkd1* and *Kif3a* (Pkd1<sup>KO</sup>+cilia<sup>KO</sup>) double knockouts that are protected from cyst growth despite inactivation of *Pkd1*[19,22], and *Pkd1* heterozygous mice that behave like wild type ("noncystic") (Fig. 1a). We sought to identify polycystin-dependent cell autonomous transcriptional changes. Since cyst formation is associated with tissue level responses such as inflammation which can manifest with non-cell-autonomous "outside-in" signaling in cyst cells, we selected a time point at which polycystin-1 and cilia had disappeared from kidney tubule cells[19] but when cysts had not yet formed to limit the contributions of such secondary inputs. Using comparisons amongst the three genotypes, we identified differentially expressed genes that showed significant "same relative direction change" in Pkd1<sup>KO</sup> when compared to both Pkd1<sup>KO</sup>+cilia<sup>KO</sup> and noncystic controls–i.e., significantly upregulated in Pkd1<sup>KO</sup> compared to both other genotypes or downregulated in Pkd1<sup>KO</sup> compared to both (Fig. 1a). Finally, while not a feature of human ADPKD, significant sex dimorphism in cyst growth in *Pkd1* adult mouse models has been described with female mice showing slower disease progression[37]. We therefore prespecified independent evaluation of differentially expressed transcriptional profiles in male and female mice.

To achieve these objectives, we used mice with the following genotypes: *Pkd1*<sup>fl/fl</sup>; *R26*<sup>Rpl10a</sup>; *Pax8*<sup>rtTA</sup>; *TetO*<sup>Cre</sup> (Pkd1<sup>KO</sup>), *Pkd1*<sup>fl/fl</sup>; *Kif3a*<sup>fl/fl</sup>; *R26*<sup>Rpl10a</sup>; *Pax8*<sup>rtTA</sup>; *TetO*<sup>Cre</sup> (Pkd1<sup>KO</sup> + cilia<sup>KO</sup>), *Pkd1*<sup>fl/+</sup>; *R26*<sup>Rpl10a</sup>; *Pax8*<sup>rtTA</sup>; *TetO*<sup>Cre</sup> ("noncystic"). All experimental mice were hemizygous for the *R26*<sup>Rpl10a</sup>, *Pax8*<sup>rtTA</sup> and TetO<sup>Cre</sup> alleles to control for dosage effects. All mice received oral doxycycline from postnatal days 28 to 42 (P28–P42)

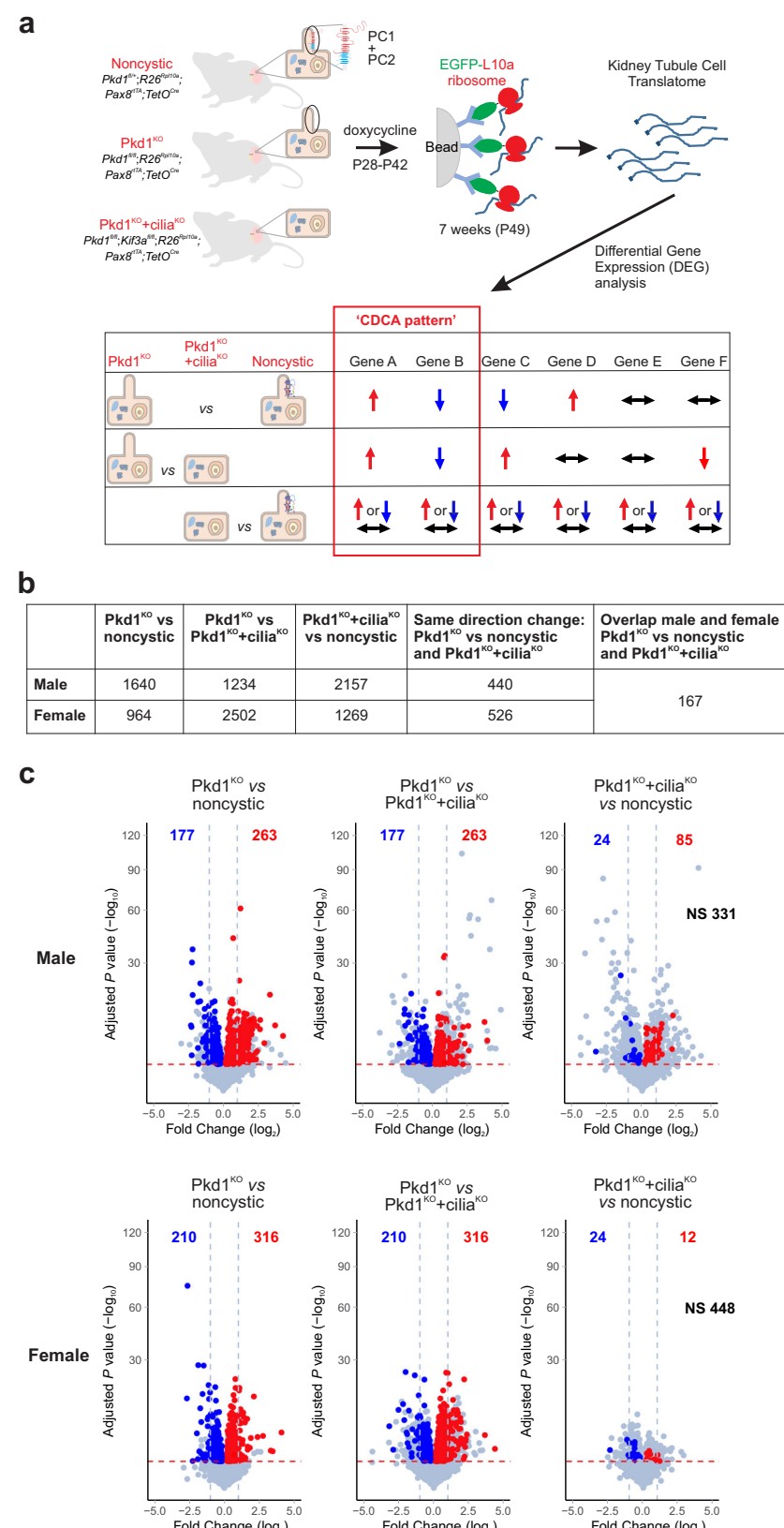

to induce *TetO*[Cre19]. Fresh kidney tissue was harvested for ribosomal pulldown at P49 (7 weeks) when mice had minimal histologic changes associated with cyst formation (Supplementary Fig. 1a, b). We used 6–8 mice for each genotype and sex, for a total of 45 TRAP RNASeq samples in the 6 groups with at least 100 million reads in each sample (Supplementary Data 1). The efficiency of *Pkd1*[fl] allele deletion in kidneys across all genotypes and in both sexes was 60–70% (Supplementary Fig. 1c). TRAP RNASeq raw sequences were processed and aligned to the mouse reference genome (Supplementary Fig. 2a). Since the TRAP is directed at actively translating mRNA, we constrained our analysis to protein coding gene transcripts. TRAP identified 15,991 and 15,968 protein coding genes in male and female mice, respectively

**Fig. 1 | Translating ribosome affinity purification (TRAP) RNASeq in *Pkd1* mouse models. a** Schematic outline of the TRAP RNASeq strategy. Male and female mice with the indicated genotypes were induced with doxycycline from P28–P42 to inactivate the respective conditional alleles and to activate the EGFP-L10a ribosomal fusion protein in a Cre recombinase dependent cell-specific manner. Labeled ribosomes were isolated from fresh kidney tissues, frozen and processed together in a single batch by RNASeq. Bioinformatic analysis of three pairwise comparisons for each sex identified differentially expressed genes (DEG). In the example shown, only Genes A and B show the same relative direction change in both the Pkd1[KO] vs. noncystic and the Pkd1[KO] vs. the Pkd1[KO]+cilia[KO] comparisons. Genes A and B are said to have the "CDCA pattern" of expression whereas Genes C–F do not. **b** Table showing numbers of DEGs identified for each sex in the indicated pairwise analyses and in Pkd1[KO] compared to both other groups. Complete gene lists are provided

Supplementary Data 2. **c** Volcano plots of DEGs in males (upper panels) and females (lower panels) for the indicated pairwise comparisons. Genes with significant differential expression (FDR ≤ 0.05) and same change direction in Pkd1[KO] compared to both noncystic and Pkd1[KO]+cilia[KO] groups are indicated by red dots and numbers (upregulated) and blue dots and numbers (downregulated). The sum of the numbers of significant DEGs with the same direction change, 440 for males and 526 for females, correspond to the data in (**b**). Gray dots and NS represent genes with FDR > 0.05. Red dashed line, adjusted $p$ value ≤ 0.05 threshold; blue dashed line, two-fold change threshold. Detection of DEGs was done with the DESeq2 R package (version 1.30.1) using a negative binomial generalized linear model. The Benjamini–Hochberg procedure was used for multiple test correction with FDR ≤ 0.05 used as the threshold for statistical significance. Additional data are provided in Supplementary Figs. 1–6.

(Supplementary Fig. 2b). The TRAP RNASeq data is available in a browsable format at https://pkdgenesandmetabolism.org/. Analysis by standard PCA and robustPCA[38] did not suggest significant outliers (Supplementary Fig. 3a); however, the previously reported sex differences were apparent (Supplementary Fig. 3b). Differentially expressed protein coding genes (DEG) were identified using the DESeq2 package. The Benjamini–Hochberg procedure was used for multiple test adjustment with false discovery rate (FDR) ≤0.05 selected as the threshold for statistical significance. Pairwise comparison between Pkd1[KO] single mutant and noncystic kidneys identified 1640 and 964 DEGs in male and female mice, respectively, whereas Pkd1[KO] compared with Pkd1[KO]+cilia[KO] double mutant kidneys identified 1234 and 2502 DEGs for male and female mice, respectively (Fig. 1b and Supplementary Data 2). Male and female mice respectively had 440 and 526 DEG that were differentially expressed with the "same relative direction of change" in Pkd1[KO] when compared to both Pkd1[KO] + cilia[KO] double knockouts and noncystic controls (Fig. 1b, c, Supplementary Fig. 4a and Supplementary Data 2). These two DEG sets are comprised of 799 unique genes since 167 DEG are shared between male and female mice (Fig. 1b, Supplementary Fig. 4b and Supplementary Data 2). Pearson correlation analysis of the 799 unique DEGs showed high correlation coefficients across all samples with particularly close grouping for all Pkd1[KO] samples which further clustered based on sex (Supplementary Fig. 5). The 167 shared DEGs follow a sex-independent "CDCA-pattern" defined as being selectively dysregulated in cyst-prone Pkd1[KO] tubule cells at the precystic stage when compared to both Pkd1[KO]+cilia[KO] double mutants that lack propensity for cyst formation and noncystic controls. We chose to focus on DEGs shared by male and female mice as a more stringent selection criterion to provide the highest confidence for definition of the "CDCA pattern" transcriptional changes. This does not, however, preclude potential functional importance for other genes, e.g., those showing significant "CDCA pattern" change only in male mice which may be associated with more rapid progression. Nonetheless, we hypothesize that this 167 gene set has the highest likelihood of containing subsets of genes that define a bona fide transcriptional response to polycystin loss in vivo. This gene set likely contains genes whose dysregulation is functionally related to in vivo cyst progression in ADPKD.

## "CDCA pattern" transcriptional candidates

Biologic pathway enrichment analysis using Metascape[39] applied to the male, female, and combined overlap "CDCA pattern" gene sets with 440, 526 and 167 genes, respectively, showed a strong enrichment of cell cycle related processes only in male mice[40], which progress more rapidly compared to female mice (Supplementary Fig. 6 and Supplementary Data 3). The 167 gene overlap group showed enrichment primarily of metabolic pathways although the Metascape term "cilium organization" also appeared (Supplementary Fig. 6 and Supplementary Data 3). Notably, these analyses did not identify pathways previously implicated in ADPKD pathogenesis, suggesting an opportunity for discovery of previously unknown transcriptional responses to

polycystin-1 function from this unbiased cell-specific in vivo analysis. We further prioritized the "CDCA pattern" DEGs based on biological knowledge. All kidney tubule segments and tubule epithelial cell types, with the likely exception of intercalated cells which lack cilia[41], have the capacity to form cysts following polycystin inactivation and which can be suppressed by inactivation of cilia[19]. Therefore, the TRAP RNASeq DEG data was cross-referenced with microdissected nephron segment-specific bulk RNASeq expression data[31] to identify transcripts expressed along the entire tubule as would be expected of candidate transcriptional changes related to CDCA. Based on these considerations we prioritized genes with expression in at least seven nephron segments where *Pax8*[rtTA] is active in our model[19] (proximal tubule S1 and S2 segments, medullary thick ascending limb, cortical thick ascending limb, distal convoluted tubule, connecting tubule, cortical collecting duct). In addition, the role of genes with statistically significant differential expression but low absolute expression in the kidney is uncertain. We therefore further prioritized the gene set to those with transcripts per million (TPM) > 1.0 in all seven microdissected segments in normal kidney[31]. These criteria yielded 185 out of 440 genes in male and 252 out of 526 genes in the female "CDCA pattern" DEGs (Supplementary Fig. 7 and Supplementary Data 4). The overlap between male and female for this subset of "CDCA pattern" DEG contained 73 genes (Fig. 2a and Supplementary Data 4). This group of 73 "CDCA pattern" genes show robust expression in all nephron segments of the normal kidney, with 61 of 73 genes having TPM > 1.0 in all 14 microdissected segments analyzed[31] (Supplementary Data 4).

The biological relevance of at least some elements of this gene set, which was selected without bias regarding functional roles, is supported by the inclusion of *Pkd2* which is significantly upregulated[42] in response to inactivation of *Pkd1* in vivo (Fig. 2a–c). Several other genes in this group have potential roles in cilia-related functions or phenotypes including *Glis2* (NPHP7)[33], *Ptpdc1*[43], *Anks3*[44], and *Rab23*[45,46] which are significantly upregulated in Pkd1[KO] and *Tmem67*[47,48] which is downregulated (Fig. 2a–c). Other genes on this list show strong statistical association with the Pkd1[KO] genotype, albeit without clear functional hypotheses. These include *Cables2*, *Chpf*, and *Tspan5* which are upregulated with Pkd1[KO] and *Lad1*, *Ntn4*, and *Spns2* which are downregulated in Pkd1[KO] relative to both other genotypes (Fig. 2a–c and Supplementary Data 2).

We next assessed the robustness of the in vivo transcriptional changes using primary cell cultures from kidneys of *Pkd1*[fl/fl]; *Pax8*[rtTA]; *TetO*[Cre] (Pkd1[KO]) mice. Cells were either treated with doxycycline to inactivate *Pkd1* during cell culture or left untreated; all cells were grown under conditions to form cilia prior to analysis. Cells were examined by quantitative reverse transcriptase PCR (qRT-PCR) to assess expression of *Cables2*, *Chpf*, *Tspan5*, *Anks3*, *Ptpdc1 Lad1*, and *Ntn4*. Expression of all 7 genes was significantly changed in the *Pkd1* knockout cells relative to the non-knockout cells in the same direction as in the TRAP RNASeq, thereby recapitulating the polycystin-1 genotype-dependent transcriptional changes found in vivo (Fig. 2d). Based on the strong association of these genes with the "CDCA pattern" of

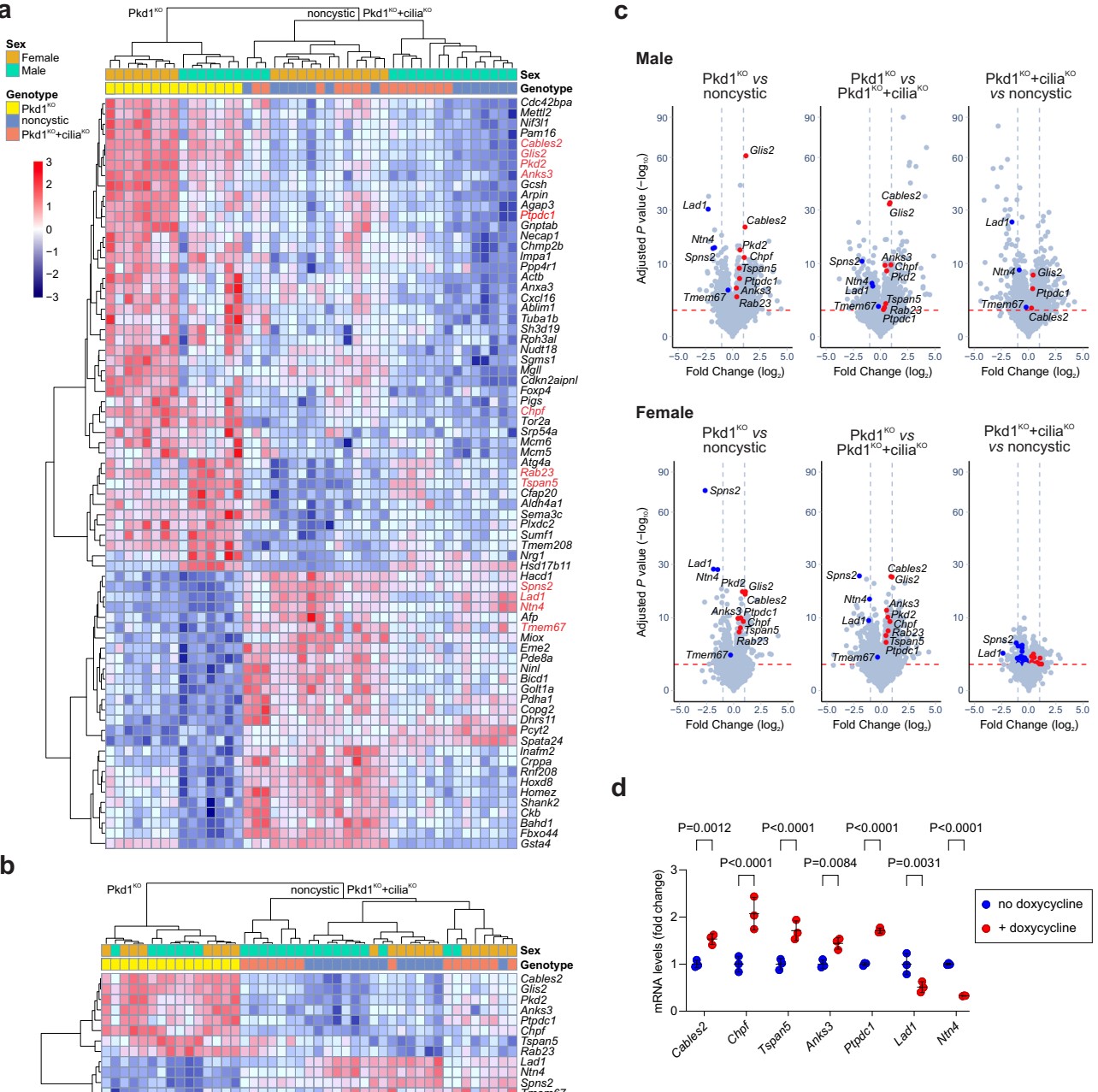

**Fig. 2 | Core group of 73 "CDCA pattern" identified by TRAP RNASeq. a** Heat map showing unsupervised hierarchical clustering of TRAP RNASeq expression data for 73 genes common to male and female mice that showed significant same relative direction change in expression in Pkd1^KO compared to noncystic and Pkd1^KO+cilia^KO and that have expression TPM > 1.0 reported in a majority of microdissected kidney tubule segments (see text for details; Supplementary Data 4). Color scale indicates relative gene expression value. Selected genes discussed in the text and shown in red in (**a**) are presented in a heat map with unsupervised hierarchal clustering (**b**), and labeled on volcano plots (**c**) with the indicated pairwise comparisons in male (top) and female (bottom) mice. Detection of DEGs was done with the DESeq2 R package (version 1.30.1) using a negative binomial generalized linear model. The

Benjamini–Hochberg procedure was used for multiple test correction with FDR ≤ 0.05 used as the threshold for statistical significance. **d** Quantitative RT-PCR (qRT-PCR) for the indicated genes from primary cells cultured from kidneys of *Pkd1^fl/fl; Pax8^rtTA; TetO^Cre* mice. Cells were either treated with doxycycline (red) to knockout *Pkd1* or left untreated (blue) without *Pkd1* knockout. *n* = 3 biological replicates for each gene and condition. Statistical significance for each gene was determined by unpaired two-tailed Student's *t* test and presented as mean ± s.e.m. Expression of each gene following *Pkd1* knockout recapitulated the relative direction change compared to non-knockout controls as was observed by TRAP RNASeq in vivo (**a**–**c**). Source data for (**d**) are provided as a Source Data file. Additional data are provided in Supplementary Fig. 7.

expression, their robust expression along the entire nephron, and the reproducibility of genotype-dependent expression changes in primary cells, we propose that a subset of this gene set will provide a transcriptional signature for polycystin function. Beyond that, this "CDCA pattern" gene set presents opportunities for discovery of previously unknown functional relationships with polycystic kidney disease mechanisms.

### *Glis2* is a target of polycystin signaling in vivo and in vitro

To explore potential functional relationships in this gene set, we selected *Glis2* for further validation. *Glis2* is among the most statistically significantly dysregulated genes in Pkd1^KO in both male and female mice compared to the other two genotype groups (Fig. 2a–c and Supplementary Data 2). It has broad expression across all segments of the nephron (Supplementary Data 4)[31]. *Glis2* is most highly

expressed in the kidney relative to other tissues and inactivation of *Glis2* results in primarily a kidney phenotype[30]. Finally, *Glis2* is a causative gene for nephronophthisis, a ciliopathy related kidney disease[33]. To determine whether the changes in *Glis2* expression found in *Pkd1* models are extensible to other preclinical models of ADPKD, we used a mouse model in which polycystic kidney disease resulting from inactivation of *Pkd2* is rapidly reversed by re-expression of *Pkd2*[36]. *Glis2* expression was elevated in 13-week-old *Pkd2* mutant kidneys showing that the transcriptional upregulation observed in *Pkd1* knockouts extends to *Pkd2* (Fig. 3a). Beginning 2 days after starting *Pkd2* re-expression, *Glis2* transcripts returned to levels that did not differ significantly from noncystic kidneys (Fig. 3a). This rapid normalization of *Glis2* expression after PC2 re-expression parallels the rapid resolution of the polycystic kidney phenotype in vivo[36]. *Glis2* is an attractive candidate for CDCA-related functions.

Glis2 protein had previously been reported to be localized in cilia[33] so we considered the hypothesis that loss of PC1 affected the presence of Glis2 in cilia. We set out to test this by first establishing that IMCD3 cells with knockout of *Pkd1* grown under conditions to form cilia show upregulation of *Glis2* and *Pkd2* mRNA compared to wild type cells (Supplementary Fig. 8a), thereby mimicking the in vivo transcriptional changes for both genes in response to *Pkd1* knockout. This suggests that these cells should be adequate models for assessing PC1-dependent Glis2 subcellular expression. Unfixed ciliated IMCD3 cells expressing Glis2-EGFP and the fluorescent fusion protein cilia marker Nphp3$^{(1-200)}$-mApple under live cell imaging conditions showed nuclear localization of Glis2-EGFP but undetectable Glis2-EGFP expression in the Nphp3$^{(1-200)}$-mApple labeled cilia (Supplementary Fig. 8b and Supplementary Movie 1). Glis2-FLAG was similarly absent from cilia in fixed HEK293A cells, suggesting that neither cell type nor epitope label accounted for the absence from cilia (Supplementary Fig. 8c). As an independent technical and positive control, Glis3-EGFP fusion protein was readily detectable in cilia in similar heterologous expression studies (Supplementary Fig. 8d). Finally, we tested whether inactivation of PC1 resulted in Glis2 localization in cilia. IMCD3 cells with inactivation of PC1[49] also failed to show Glis2-EGFP expression in cilia (Supplementary Fig. 8e, f and Supplementary Movies 2, 3). These data show that Glis2 is localized to the nucleus but is not detectable in cilia and the latter is unaffected by *Pkd1* mutation status.

To allow comprehensive evaluation of changes in *Glis2* protein expression, we developed and validated a polyclonal antibody to Glis2 (Supplementary Fig. 9a–d). The anti-Glis2 antibody (YNG2) recognizes epitope tagged over-expressed mouse and human Glis2 by immunoblotting (Supplementary Fig. 9a, b). The specificity of YNG2 is further indicated by detection of native Glis2 in nuclear fractions of cell lines and tissue lysates from kidneys of wild type mice, but not in kidney cell lines and lysates from *Glis2* null mice[50] (Supplementary Fig. 9c, d). The bulk of Glis2 protein in both cultured cells and kidney tissue lysates is expressed in the nucleus, with little detected in the cytosolic fraction (Supplementary Fig. 9c, d). We next used the developing kidney to evaluate correlation of native tissue mRNA[51] and protein expression for *Glis2*. *Glis2* showed elevated expression of both transcript and protein in postnatal kidneys at P1 and P10 followed by downregulation of both transcript and protein at P20 and P40 (Fig. 3b, c). Next, we examined Pkd genotype dependence of *Glis2* expression in primary cells cultured from kidneys of mice with allele combinations that allowed for conditional inactivation of target genes following in vitro treatment with doxycycline. All cells were studied under conditions that promoted cilia formation prior to mRNA and protein extraction. Primary cell cultures from kidneys of *Pkd1$^{fl/fl}$; Pax8$^{rtTA}$; TetO$^{Cre}$* (Fig. 3d, e) and *Pkd2$^{fl/fl}$; Pax8$^{rtTA}$; TetO$^{Cre}$* (Fig. 3f, g) mice were treated with doxycycline in cell culture to induce inactivation of the respective Pkd gene (Fig. 3g and Supplementary Fig. 10). *Glis2* transcript expression increased in both *Pkd1* and *Pkd2* knockout cells compared to cells not treated with doxycycline (Fig. 3d, f). Doxycycline treatment alone, without Pkd

gene knockout had no effect on *Glis2* expression. *Pkd1* mutant primary cells also showed upregulation of *Pkd2* transcripts, reproducing another feature of the in vivo TRAP RNASeq and published data[42] (Fig. 3d). *Pkd1* transcript expression was upregulated following *Pkd2* inactivation in the primary cell system suggesting possible coordinate transcriptional regulation of the polycystin genes (Fig. 3f). At the protein level, Glis2 showed increased nuclear expression following doxycycline induced *Pkd1* or *Pkd2* inactivation (Fig. 3e, g).

Primary cell cultures were also produced from kidneys of littermate mice with genotypes supporting inducible in vitro inactivation of *Pkd1* alone or *Pkd1* and *Kif3a* (Pkd1+cilia) together. These cells required longer time in culture after doxycycline treatment to allow for disappearance of cilia (Supplementary Fig. 11a). Cells with *Pkd1* inactivation alone again showed upregulation of both transcript and protein expression for *Glis2* and *Pkd2* (Fig. 3h, i). Cells with dual inactivation of Pkd1+cilia showed no difference in expression of either *Glis2* or *Pkd2* compared to control cells without *Pkd1* knockout (Fig. 3h, i). Suppression of Glis2 protein expression was also observed with *Pkd2* and *Ift88* (cilia) double knockout cells compared to *Pkd2*-only knockout primary cultures (Fig. 3j and Supplementary Fig. 11b). Adult inducible *Pkd1* and *Tulp3* double knockout mice have a similar suppressive effect on kidney cyst formation as Pkd1+cilia knockouts, but without loss of cilia[20]. Glis2 and PC2 protein expression in *Pkd1;Tulp3* double knockout cells remained unchanged from non-knockout controls further supporting the strict correlation of Glis2 levels in vitro with polycystin-dependent cyst forming potential in vivo (Fig. 3k). *Glis2* transcript and Glis2 protein expression are molecular markers that recapitulate the "CDCA pattern" in vitro.

Finally, we performed in vivo validation experiments using mouse kidney tissues in two different models to determine whether changes in protein expression correlated with changes in transcript levels. In the first model, kidney tissue lysates from *Pkd1$^{fl/fl}$; Pax8$^{rtTA}$; TetO$^{Cre}$* mice showed increased Glis2 protein as well as PC2 expression following doxycycline induced *Pkd1* knockout compared to mice with the same genotype that did not have doxycycline induced knockout (Fig. 3l). This knockout-dependent change in expression was abrogated in doxycycline treated *Pkd1$^{fl/fl}$; Kif3a$^{fl/fl}$; Pax8$^{rtTA}$; TetO$^{Cre}$* double knockout mouse kidney lysates (Fig. 3l). The increase in Glis2 protein expression following *Pkd1* inactivation in *Pkd1$^{fl/fl}$; Pax8$^{rtTA}$; TetO$^{Cre}$* mice was primarily in the nuclear fraction of the tissue (Fig. 3m). In the second model, Glis2 protein upregulation in response to Pkd1$^{KO}$ was markedly suppressed in *Pkd1$^{fl/fl}$; Tulp3$^{fl/fl}$; Pax8$^{rtTA}$; TetO$^{Cre}$* mice following dual gene inactivation that is known to prevent cyst progression in adult inducible models[20] (Fig. 3m). In aggregate, these data confirm the strong "CDCA pattern" correlation between Glis2 protein nuclear expression and propensity for cyst formation in vivo. We conclude that polycystin dependent transcriptional changes in *Glis2* manifest as correlated changes in protein expression with most of the protein showing nuclear localization in vivo and in vitro. *Glis2* shows genotype and phenotype dependent changes in expression in both *Pkd1* and *Pkd2* primary kidney cells that recapitulate the expression changes discovered by the TRAP RNASeq studies in vivo. *Glis2* transcript and protein expression changes in primary cells are an in vitro surrogate indicator of in vivo cyst forming potential resulting from polycystin inactivation.

## Inactivation of *Glis2* slows cyst progression in mouse models of ADPKD

To investigate whether *Glis2* has a phenotypic effect in ADPKD, we used a series of mouse allele combinations to determine whether inactivation of *Glis2* affected polycystin-dependent cyst progression. We first introduced a knock-out allele of *Glis2*[50] into the early-onset *Pkd1$^{fl/fl}$;Pkhd1$^{Cre}$* mouse model[41] (Fig. 4a–d and Supplementary Fig. 12). *Pkhd1$^{Cre}$* results in complete inactivation of *Pkd1* in principal cells of the collecting duct by P7[19]. Mice have significant polycystic kidney disease

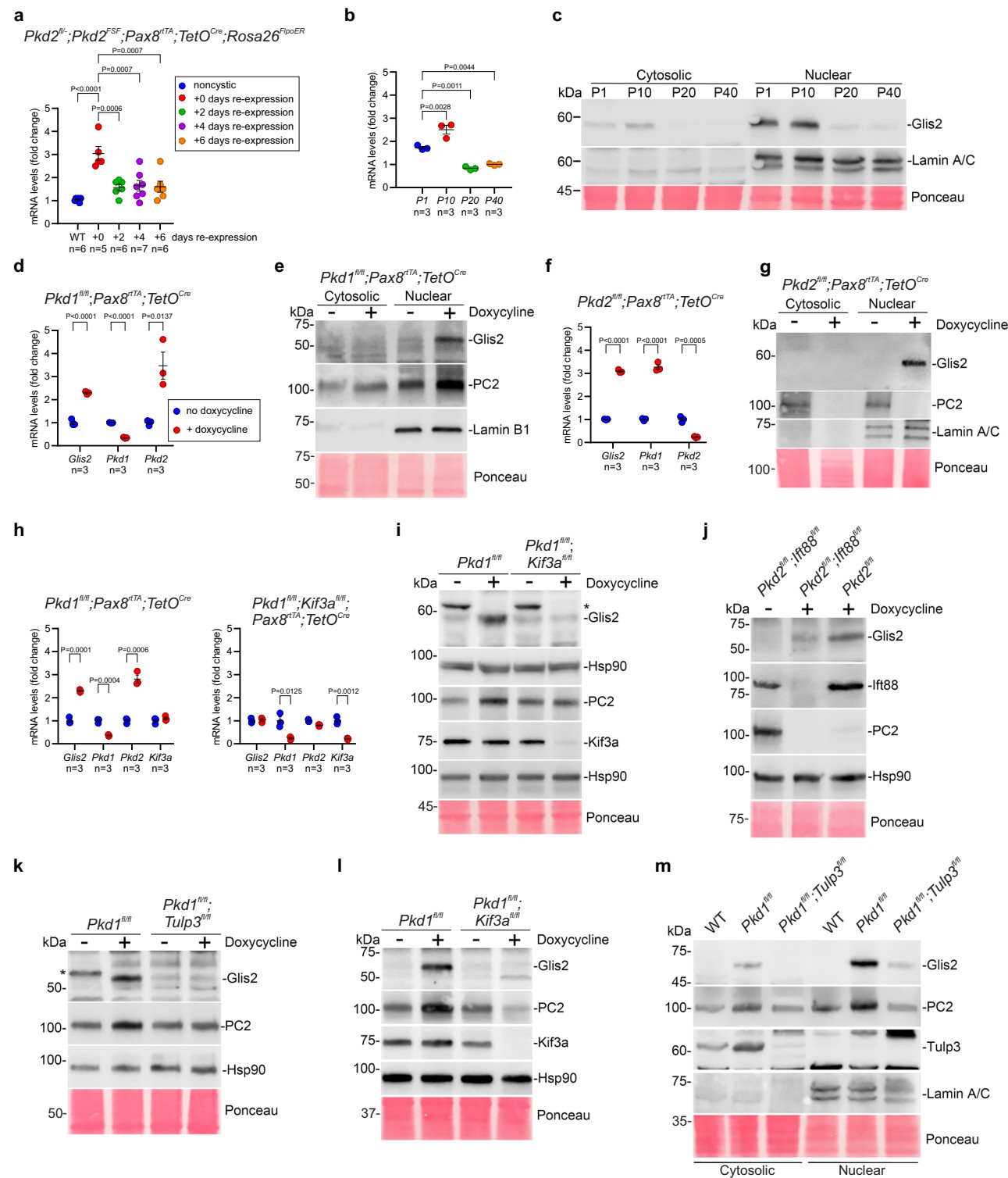

at P24 manifest by the structural parameters of increased kidney-to-body weight ratio and percent cystic area (cystic index) and the kidney functional measure of increased blood urea nitrogen (BUN)[19]. *Glis2* null mice at that age show subtle microscopic changes of mild cortical tubule atrophy and thickening of the glomerular basement membrane with normal BUN[50]. *Glis2*[−/−]; *Pkd1*[fl/fl]; *Pkhd1*[Cre] were significantly protected from polycystic kidney disease compared to *Pkd1*[fl/fl]; *Pkhd1*[Cre] mice at P24 (Fig. 4a–d and Supplementary Fig. 12). We further evaluated the durability of this rescue by examining mice with these same allele combinations at P49 (Supplementary Fig. 13). Germline null *Glis2*

supported persistent relative improvement of polycystic kidney disease with normal kidney-to-body weight ratio and significantly milder cystic index and BUN elevation when compared to *Pkd1*[fl/fl]; *Pkhd1*[Cre] mice at P49. To examine whether the protective effect of *Glis2* loss on cyst growth is also applicable to other nephron segments as would be expected of factors related to CDCA, *Glis2*[−/−] knockout was combined with *Pkd1*[fl/fl]; *Pax8*[rtTA]; *TetO*[Cre] in another early onset mouse model. Oral doxycycline was administered to nursing female mice from P0–P14 and kidneys of the progeny litter were analyzed at P14[20]. Absence of *Glis2* resulted in significant protection from cyst growth following *Pkd1*

**Fig. 3 | Glis2 expression is an in vitro indicator of polycystin dependent cyst forming potential. a** qRT-PCR of *Glis2* from kidney tissue lysates of noncystic (blue) and cystic *Pkd2* knockout (red) 13 week old mice and from *Pkd2* knockout kidneys following 2, 4 and 6 days of re-expression begun at 13 weeks. All mice received oral doxycycline from P28–42. Noncystic mice lack *Pax8^rtTA*. Fold-change *Glis2* expression relative to that in noncystic mice which is set to 1.0. **b** qRT-PCR and (**c**) immunoblot of kidney tissue lysates from wild type mice at postnatal days 1, 10, 20 and 40. **b** Fold-change *Glis2* relative to its expression at P40 which is set to 1.0. **c** Glis2 expression in cytosolic and nuclear fractionated kidney tissue lysates. **d, f** qRT-PCR and (**e, g**) immunoblots from primary kidney cells with *Pkd1* (**d, e**) and *Pkd2* (**f, g**) knockout following. doxycycline treatment during in vitro cell culture. **d, f** Fold change following doxycycline mediated inactivation (red) relative to expression of the same gene in cells without doxycycline (blue) which is set to 1.0. **h** qRT-PCR and (**i–k**) immunoblots from lysates of primary kidney cell cultures following in vitro doxycycline inducible inactivation of *Pkd1* and *Kif3a* (**h, i**), *Pkd2* and *Ift88* (**j**), and *Pkd1* and *Tulp3* (**k**). All mice have *Pax8^rtTA*;*TetO^Cre* in addition to the indicated alleles. **h** Fold change following doxycycline mediated inactivation (red) relative to expression of the same gene without doxycycline (blue) which is set to 1.0. **i, k** *, non-specific band. **l, m** Immunoblots from total kidney tissue lysates from mice with indicated allele combinations. All mice also have *Pax8^rtTA*;*TetO^Cre* alleles and received oral doxycycline from P28–42; kidney tissue was obtained at 10 weeks (**l**) and 12 weeks (**m**). Experiments in (**c, g, j, m**) were done two times with primary cell cultures from different mice; **i, k** were done three times; **e, l** were done more than three times. **a, b** Multiple-group comparisons using one-way ANOVA followed by Tukey's multiple-comparison test and presented as the mean ± s.e.m. **d, f, h** Statistical significance for each gene is determined by unpaired two-tailed Student's *t* test and presented as mean ± s.e.m. **c, g, m** Lamin A/C and (**e**) Lamin B1, loading control and relative enrichment for nuclear fractions. **i–l** Hsp90 loading control. **c, e, g, i–m** Ponceau S serves as another loading control. Source data for exact values and images of uncropped blots are provided as a Source Data file. Additional data are provided in Supplementary Figs. 8–11.

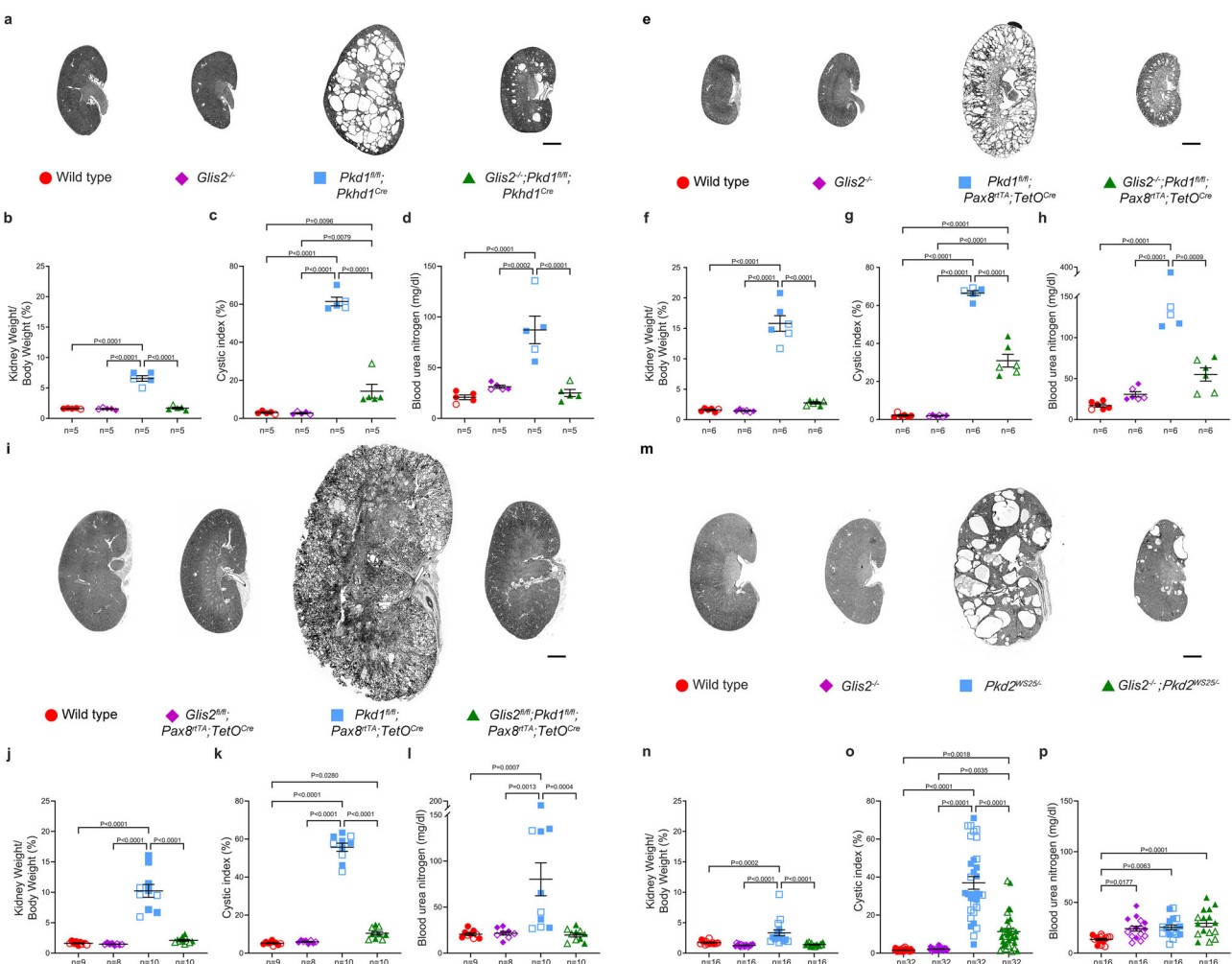

**Fig. 4 | Glis2 inactivation suppresses progression in mouse models of polycystic kidney disease. a, e, i, m** Representative images of kidneys with the specified genotypes and **b–d, f–h, j–l, n–p** show corresponding kidney structural (**b, f, j, n** kidney weight to body weight ratio; **c, g, k, o** cystic index) and functional (**d, h, l, p** blood urea nitrogen) measures for male (closed symbols) and female (open symbols) mice. *n*, number of mice except in (**o**), where it is the number of kidneys. Multiple-group comparisons were performed by one-way ANOVA followed by Tukey's multiple-comparison test, presented as mean ± s.e.m. **a, e, i, m** Scale bars, 1 mm. **a–d** *Pkd1* models based on the collecting duct selective *Pkhd1-Cre* at P24. Genotypes in (**b–d**) correspond to symbol color and shape coding in (**a**). **e–h** *Pkd1* models based on *Pax8^rtTA* induced with oral doxycycline from P1–14 and examined at P14. Genotypes in (**f–h**) correspond to symbol color and shape coding in (**e**). **i–l** *Pkd1* models based *Pax8^rtTA* induced with oral doxycycline from P28–42 and examined at 18 weeks. Genotypes in (**j–l**) correspond to symbol color and shape coding in (**i**). **m–p** *Pkd2* models based on the *Pkd2^WS25* allele examined at 14 weeks. Genotypes in (**n–p**) correspond to symbol color and shape coding in (**m**). Source data for exact values are provided as a Source Data file. Additional data are provided in Supplementary Figs. 12–18.

inactivation as indicated by significant improvements in kidney-to-body weight ratio, cystic index and BUN levels (Fig. 4e–h and Supplementary Fig. 14). These findings suggest that elevated expression of *Glis2* is a permissive factor for cyst progression in vivo and absence of *Glis2* improves renal prognosis in early onset ADPKD models based on *Pkd1*.

Cyst growth in adult onset ADPKD models is slower than in early onset models[52,53] and may more closely reflect the course of ADPKD in most patients. Since germline null *Glis2*[−/−] mice develop progressive nephronophthisis-like chronic kidney disease that is evident at 4 months age[50], we developed an inducible conditional *Glis2*[fl/fl]; *Pkd1*[fl/fl]; *Pax8*[rtTA]; *TetO*[Cre] model to evaluate the effects of contemporaneous *Glis2* inactivation on cyst growth in later onset mouse models. All mice were administered doxycycline from P28–P42, and severity of polycystic kidney disease was examined at 18 weeks age. Unlike germline null *Glis2* inactivation, inducible inactivation of *Glis2* alone beginning at P28 did not result in any discernible kidney phenotype when assessed for structural changes and evidence of collagen deposition at 18 weeks (Supplementary Fig. 15a, b). *Pkd1*[fl/fl]; *Pax8*[rtTA]; *TetO*[Cre] mice at 18 weeks showed the expected structural and fibrotic changes of polycystic kidney disease along with elevated BUN indicative of kidney function impairment (Fig. 4i–l and Supplementary Figs. 15c–f and 16). *Glis2* and *Pkd1* double mutant mice showed a much milder phenotype than *Pkd1* single mutants at 18 weeks age when evaluated by kidney-to-body weight ratio, cystic index, BUN and histologic criteria (Fig. 4i–l and Supplementary Figs. 15c–f and 16). *Glis2* and *Pkd1* dual inactivation was accompanied by normalization of kidney epithelial cell proliferation measured by EdU incorporation and Ki67 in both proximal and distal nephron segments compared to cystic *Pkd1* single mutants (Supplementary Fig. 17). Germline inactivation of *Glis2* also results in senescence in kidney epithelial cells[54]. We used senescence-associated β-galactosidase (SA-β-gal) staining and qRT-PCR for *Cdkn1* and *Cdkn2a* to determine whether cellular senescence occurs in kidneys of 18-week-old mice with adult inducible inactivation of *Glis2* (Supplementary Fig. 18). *Glis2*[ΔEx3/ΔEx3] null mice, using the germline null allele generated from our *Glis2*[fl] allele, showed extensive cortical SA-β-gal staining and elevated mRNA levels for *Cdkn1a* and *Cdkn2a* confirming previous reports of increased cellular senescence in *Glis2* germline null mice (Supplementary Fig. 18a, b, f, g)[54]. Kidneys from *Pkd1*[fl/fl]; *Pax8*[rtTA]; *TetO*[Cre] mice showed only occasional cysts with SA-β-gal staining and no increase in *Cdkn1a* or *Cdkn2a* mRNA expression (Supplementary Fig. 18c, f, g). This sporadic SA-β-gal staining may indicate increased lysosomal activity in a few cysts, but is not specific for increased senescence, which is not present at the whole organ level. Adult *Glis2*[fl/fl]; *Pkd1*[fl/fl]; *Pax8*[rtTA]; *TetO*[Cre] double knockout and *Glis2*[fl/fl]; *Pax8*[rtTA]; *TetO*[Cre] single knockout kidneys did not show any evidence of SA-β-gal staining or elevated *Cdkn1a* and *Cdkn2a* transcript expression (Supplementary Fig. 18d–g). These findings indicate that the association of *Glis2* inactivation with epithelial cell senescence in the kidney is a feature of germline null nephronophthisis phenotypes and not a feature of postnatal kidney tubule selective inactivation of *Glis2*. Overall, adult inducible tubule specific inactivation of *Glis2* in kidney epithelial cells suppresses kidney cyst growth following *Pkd1* inactivation without evidence of untoward effects on kidney structure or function.

Finally, CDCA related factors are expected to affect ADPKD due to *Pkd2* as well as *Pkd1*[19]. We examined the effects of *Glis2*[−/−] null alleles on polycystic kidney disease progression in the *Pkd2*[WS25/−] model[55]. The spontaneous stochastic second hit mutations in *Pkd2*[WS25/−] most closely recapitulate the mechanism of human ADPKD. Since there is no Cre recombinase in this model, we used the *Glis2*[−/−] null mice. We selected 14 weeks as the endpoint for the analysis to allow enough time for the highly variable *Pkd2*[WS25/−] model to develop polycystic kidney disease while limiting the progressive chronic kidney disease that occurs in Glis2[−/−] mice[50]. *Glis2*[−/−]; *Pkd2*[WS25/−] mice showed reduced cyst growth in kidneys as reflected by significant reduction in kidney-to-body weight

ratio and cystic index (Fig. 4m–p and Supplementary Fig. 19). The same degree of mild impairment of kidney function was observed in *Glis2*[−/−], *Pkd2*[WS25/−] double mutant mice as in the respective single mutant *Glis2*[−/−] and *Pkd2*[WS25/−] mice. The underlying impairment of kidney function caused by the *Glis2*[−/−] null mutation precludes clear interpretation of the effects in *Glis2*[−/−]; *Pkd2*[WS25/−] mice, but the absence of an additive worsening of BUN in *Glis2*[−/−]; *Pkd2*[WS25/−] mice is permissive for the interpretation that renal impairment due to polycystic kidney disease was limited with the reduction in cyst growth. *Glis2* inactivation is effective in reducing polycystic kidney disease progression in adult models based on both *Pkd1* and *Pkd2*. The *Pkd* genotype-dependent changes in *Glis2* transcription and protein expression have a causative role in progression of polycystic kidney disease that can be ameliorated by inactivation of the *Glis2* gene.

## *Glis2* is a target for therapy in ADPKD

We next sought to evaluate whether *Glis2* is a suitable target for preclinical pharmacological intervention to reduce polycystic kidney disease progression. We designed an optimized antisense oligonucleotide (ASO)[56] targeting degradation of mouse *Glis2* (Glis2-ASO) and a scrambled sequence control ASO (control-ASO). Since ASO are most effectively taken up in the proximal tubule in the kidney[57,58], we tested the Glis2-ASO effect in the tamoxifen inducible *Pkd1*[fl/fl]; *UBC*[Cre-ERT2] model[19] which only has Cre activity in the proximal tubule in the kidney when evaluated with the *ROSA*[mT/mG] reporter (Supplementary Fig. 20). Mice received daily intraperitoneal (IP) tamoxifen from P28-P35 to inactivate *Pkd1* in proximal tubule cells. Mice were treated with control- or Glis2-ASO at a dose of 50 mg/kg by IP injection from 5 weeks of age to 18 weeks age. Mice were injected twice during week 5 and then once per week thereafter for a total of 15 doses; polycystic kidney disease progression was examined at 18 weeks of age. Kidneys of Glis2-ASO treated mice had significantly reduced kidney-to-body weight ratio, cystic index, and BUN (Fig. 5a–d and Supplementary Fig. 21). The data showed statistically significant improvement with Glis2-ASO compared to control-ASO treatment with both sexes combined. However, the previously reported sex dimorphism in *Pkd1* disease severity in adult mouse models[37] notably affected the distribution of the results. Male mice developed severe polycystic kidney disease with mean kidney-to-body weight ratio ~8.8% that was associated with mean cystic index ~48% and BUN 77 mg/dl in control-ASO treated mice. These parameters were significantly improved by Glis2-ASO treatment to respective means of ~4.3%, ~25% and 43 mg/dl in the male mice (Supplementary Fig. 22a–c). Female mice developed minimal polycystic kidney disease with mean kidney-to-body weight ratio of only ~3.3%, cystic index ~22% and BUN 29 mg/dl. These parameters all decreased slightly to ~2.6%, ~19% and 27 mg/dL, respectively, with treatment but the findings were not powered to achieve statistical significance in female mice given the mild control-ASO treated disease state (Supplementary Fig. 22d–f).

We assessed the "target engagement" efficiency of proximal tubule *Glis2* mRNA depletion by the Glis2-ASO compared to control-ASO in 18-week-old kidneys of male mice using two independent semiquantitative fluorescent in situ hybridization (FISH) approaches—single molecule FISH (smFISH) and RNAScope-FISH (Fig. 5e–n and Supplementary Fig. 23). Two color FISH with probes directed against the proximal tubule specific endocytic receptor megalin (*Lrp2*) allowed assessment of *Glis2* mRNA expression in the region of proximal tubule cells (Fig. 5e, i). The specificity of both FISH methods was confirmed by absence of hybridization of the *Glis2* probes on *Glis2*[−/−] null kidney tissue sections (Fig. 5f, j). Glis2-ASO treatment resulted in reduced *Glis2* FISH signal compared to control-ASO treatment by both detection methods (Fig. 5g, h, k, l). To quantitate these differences, regions of interest (ROI) near the proximal tubules in multiple images from kidney sections from control- and Glis2-ASO treated mice were manually selected based solely on the megalin FISH channel (Supplementary Fig. 23).

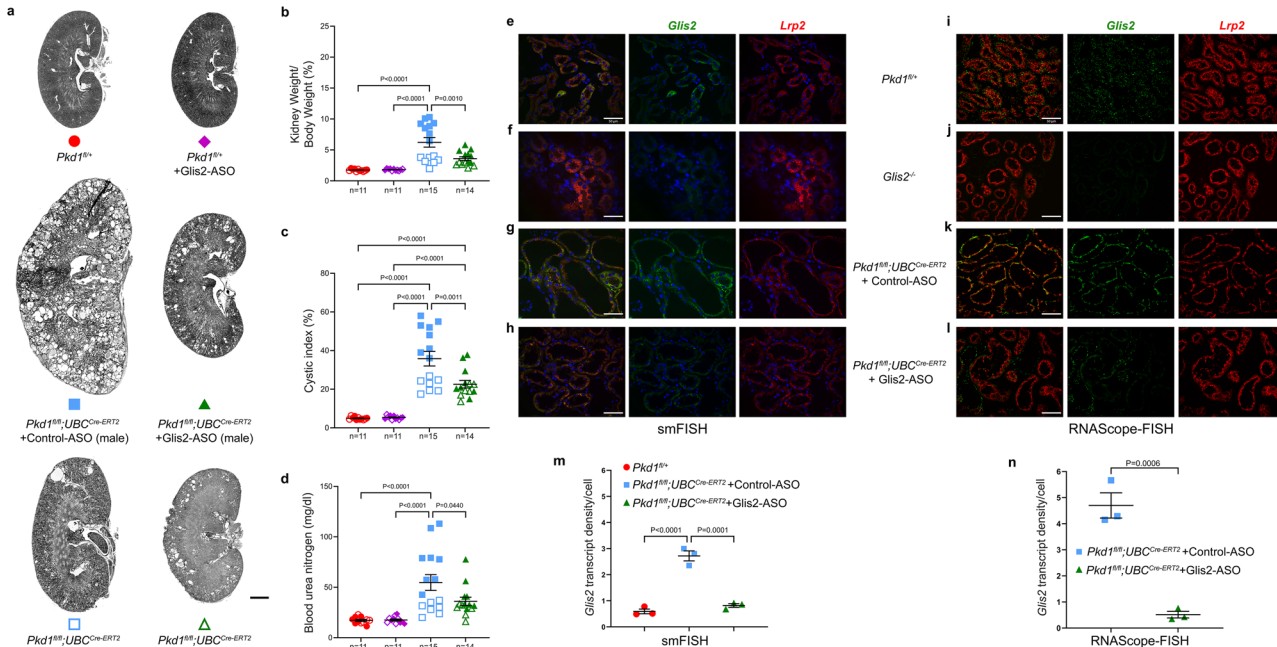

**Fig. 5 | *Glis2* antisense oligonucleotide (ASO) reduces kidney cyst growth in an adult onset ADPKD model. a** Representative kidney images, **b** kidney weight to body weight ratio, **c** cyst index and **d** blood urea nitrogen from mice with the indicated genotypes in (**a**) and matched symbol color and shape codes in (**b**–**d**) at 18 weeks age. *Pkd1* inactivation was induced with tamoxifen administration from P28–35 and control- or Glis2-ASO was administered twice in week 5 and then weekly through 18 weeks age. All mice received tamoxifen regardless of genotype. One group of noncystic controls also received Glis2-ASO; the other did not. **e–l** Representative images of two-color fluorescent in situ hybridization (FISH) for *Glis2* mRNA (green) and the proximal tubule marker gene *Lrp2* (red) encoding megalin. **b–d** Male mice, closed symbols; female mice, open symbols. *n* number of mice. **e–h** Single molecule FISH (smFISH); **i–l** RNAScope-FISH. *Glis2* expression is detected by both methods in control mouse kidneys (**e**, **i**) and the specificity of both probes is demonstrated by the absence of signal for *Glis2* in kidneys from

*Glis2⁻/⁻* mice (**f**, **j**). *Glis2* signal by both FISH methods is increased relative to control kidney in *Lrp2* positive proximal tubule derived cysts in mice treated with control-ASO (**g**, **k**). *Glis2* signal by both FISH approaches is decreased relative to control-ASO in proximal tubules of mice treated with Glis2-ASO (**h**, **l**). All mice, except (**f**, **j**), received tamoxifen IP daily from P28-35; all kidneys were examined at age 18 weeks. Treatment with Glis2-ASO significantly decreased *Glis2* mRNA levels in proximal tubules of *Pkd1ᶠˡ/ᶠˡ;UBCᶜʳᵉ⁻ᴱᴿᵀ²* male mice compared to control-ASO treated counterparts when measured both smFISH (**m**) and RNAScope-FISH (**n**). *n* = 3 for all groups. Multiple-group comparisons (**b–d**, **m**) were performed by one-way ANOVA followed by Tukey's multiple-comparison test. Comparison between two groups (**n**) was performed by one-tailed unpaired Student's *t* test. All data are presented as mean ± s.e.m. Scale bar, 1 mm (**a**); 50 µm (**e–l**). Source data for exact values are provided as a Source Data file. Additional data are provided in Supplementary Figs. 19–24.

The numbers of *Glis2* FISH signal points and the number of DAPI stained nuclei as a surrogate for cell numbers in these ROI were quantified using CellProfiler and the *Glis2* transcript density per cell was used for relative quantitation (Fig. 5m, n). Treatment with the Glis2-ASO significantly decreased *Glis2* mRNA in regions of the cortex expressing *Lrp2* (i.e., proximal tubules) when measured by both smFISH and RNAScope-FISH (Fig. 5m, n).

Glis2-ASO treatment in noncystic kidneys did not result in discernible evidence of senescence when evaluated by SA-β-gal staining (Supplementary Fig. 24a, b). Kidneys from *Pkd1ᶠˡ/ᶠˡ; UBCᶜʳᵉ⁻ᴱᴿᵀ²* mice treated with control-ASO again showed occasional cysts with SA-β-gal staining as previously noted in untreated mice with polycystic kidney disease (Supplementary Fig. 24c). Glis2-ASO treated *Pkd1ᶠˡ/ᶠˡ; UBCᶜʳᵉ⁻ᴱᴿᵀ²* mice showed similar presence of sporadic SA-β-gal cell staining that was qualitatively reduced compared to untreated cystic kidneys indicating that Glis2-ASO did not result in increased lysosomal activity as detected by SA-β-gal staining (Supplementary Fig. 24d). Finally, we assessed several additional markers of polycystic kidney disease severity in response to Glis2-ASO treatment in male mice. Cyst cell proliferation is a common feature of ADPKD. Male *Pkd1ᶠˡ/ᶠˡ; UBCᶜʳᵉ⁻ᴱᴿᵀ²* polycystic mice treated with control-ASO showed increased proximal tubule cell proliferation measured by EdU incorporation and Ki67 staining, and this proliferation was significantly reduced in mice treated with Glis2-ASO (Fig. 6a–d). Polycystic kidneys are also characterized by inflammatory changes. The presence of increased interstitial macrophages in control-ASO treated cystic kidneys was

significantly reduced following Glis2-ASO treatment (Fig. 6e, f). We had previously shown that the cytokine TNF-α and the NLRP3 inflammasome effector cleaved-caspase-1 are elevated in the interstitial regions of polycystic kidneys[36]. In keeping with these observations, control-ASO treated cystic kidneys showed elevated levels of the TNF-α and cleaved caspase-1; both were significantly reduced in mice treated with Glis2-ASO (Fig. 6g–j). Another hallmark of progressive polycystic kidney disease is tubulointerstitial fibrosis. The fibrotic response can be reduced or prevented in ADPKD but once it is established, it can result in irreversible kidney damage[36]. Control-ASO treated kidneys showed increased myofibroblast activation by immunoblotting and immunohistochemistry for α-smooth muscle actin (α-SMA) and platelet-derived growth factor receptor β (PDGFRβ) (Fig. 6k–n). Treatment with Glis2-ASO significantly reduced the levels of both markers of fibrosis compared to the control-ASO treatment (Fig. 6k–n). Overall, the studies presented validate a paradigm for in vivo translatome discovery in ADPKD preclinical models and show that upregulation of *Glis2* discovered by this approach is a marker for polycystin function both in vivo and in vitro. Beyond being a marker for polycystin activity, Glis2 is shown in preclinical studies to be therapeutic target for treatment of ADPKD.

## Discussion

The polycystins were initially discovered as complex, novel proteins with uncertain function. They were subsequently localized to the primary cilium. The precise molecular composition and function of the

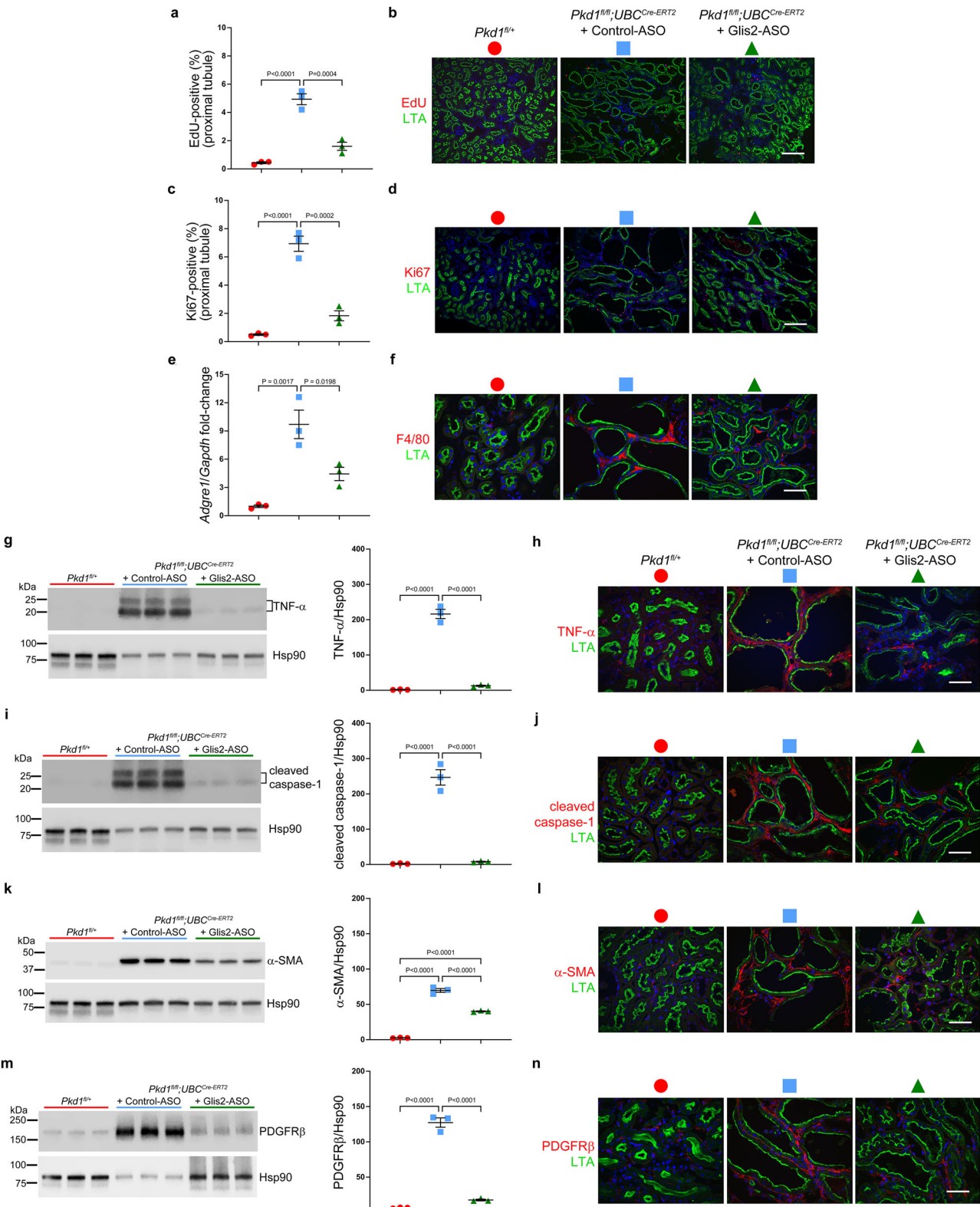

polycystin complex in primary cilia in the metanephric kidney remains incompletely understood. While many known cellular pathways have been shown to be altered in ADPKD, molecular mechanistic connections between these dysregulated pathways and polycystin function in cilia has been lacking (reviewed in refs. 59–61). The novel nature of the polycystins and of their organellar compartment, coupled with the absence of biological mechanistic precedents for connecting these features to in vivo phenotypes, suggest that critical functional

relationships in the polycystin effector pathways are yet to be discovered. In the current study, we adopted strategies to produce an unbiased discovery platform based on early tubule cell-specific transcriptional changes specifically related to the in vivo propensity for *Pkd1*-dependent cyst formation. First, we used TRAP RNASeq to selectively capture the kidney tubule cell type-specific in vivo translatome only from cells with *Pkd1* inactivation[23,62]. Second, we used a tripartite comparison of genotypes that included Pkd1[KO]+cilia[KO] double

**Fig. 6 | Glis2-ASO treatment decreases cyst cell proliferation, inflammation, and fibrotic changes in a model of ADPKD. a–n** Analyses of kidney tissues from noncystic mice (red symbols) and polycystic *Pkd1;UBC^Cre-ERT2* mice treated with control-ASO (blue) or Glis2-ASO (green). All mice are male; all received tamoxifen from P28–35; treatment groups received the respective ASO from weeks 5 to 18. Kidney tissue was examined at 18 weeks. **a, c** Aggregate quantitative data and (**b, d**) representative images showing the percentage of EdU-positive nuclei (**a, b**) and Ki67-positive nuclei (**c, d**) in the Lotus tetragonolobus agglutinin (LTA) positive proximal tubules. **e** qRT-PCR for *Adgre1*, encoding F4/80, from kidney lysates normalized to *Gapdh*, expressed as fold-change relative to the mean for control kidneys which is set to 1.0. **f** Representative images of F4/80 expression near proximal tubules. Changes in inflammatory responses indicated by changes in TNF-α (**g, h**) and cleaved caspase-1 (**i, j**). Immunoblots are shown with quantitation using densitometric ratios (**g, i**) that include two bands each for TNF-α (**g**) and cleaved caspase-1 (**i**) normalized to Hsp90. Fold-change is shown relative to the mean of the ratio in the non-cystic samples, which is set to 1.0. Representative images of TNF-α (**h**) and cleaved caspase-1 (**j**). **k–n** Changes in fibrotic responses indicated by changes in α-SMA (**k, l**) and PDGFRβ (**m, n**) expression following control- or Glis2-ASO treatment. Immunoblots are shown with quantitation using densitometric ratios (**k, m**) for α-SMA (**k**) and PDGFRβ (**m**) normalized to Hsp90. Fold-change is shown relative to the mean of the ratio in the non-cystic samples, which is set to 1.0. Representative images of α-SMA (**l**) and PDGFRβ (**n**). **b, d, f, h, j, l, n** At least one section from 3 kidneys for each group was examined for all representative images. Scale bars: 150 μm (**b, d**); 50 μm (**f, h, j, l, n**). **a, c, e, g, i, k, m** Multiple-group comparisons were performed by one-way ANOVA followed by Tukey's multiple-comparison test, presented as mean ± s.e.m. Full-length blots are provided as source data. Source data for exact values and images of uncropped blots are provided as a Source Data file.

knockouts to enhance specificity for transcriptional changes associated with cyst formation which is markedly attenuated in Pkd1^KO+cilia^KO kidneys despite inactivation of *Pkd1*[19,22]. Third, we selected a time point before discernible cystic tubule dilation following *Pkd1* inactivation to reduce the impact of non-cell autonomous "outside-in" signals. To increase the utility of these TRAP RNASeq and related data to the broader research community, we have set up the "Metabolism and Genomics in Cystic Kidney (MAGICK)" (https://pkdgenesandmetabolism.org/) website making the data readily browsable. The in vivo TRAP translatome derived in these studies should offer broad opportunities for focused hypothesis generation and discovery in polycystin- and cilia-related kidney phenotypes.

A transcriptional signature for PKD based on bulk RNASeq data and comprised of 1,515 commonly dysregulated genes (~7.5% of protein coding genes) has been previously proposed[63]. This set contained 92 transcription factors which included Glis2, although it was not prioritized[64]. The challenge with such broad gene lists is that it is difficult to prioritize focus for biological investigation and, because it is based on bulk RNASeq, it incorporates many changes in the polycystic kidney that are extrinsic to cyst cells. In an effort to further focus the gene set in the TRAP translatome, we applied parameters based on biological context. To identify the highest confidence minimal "CDCA pattern" transcriptional changes, we focused on DEG that were in common between male and female mice with the knowledge, confirmed in our current studies, that female mice are relatively phenotypically protected in cyst progression[37]. Notably, sex dimorphism as seen in adult mice is not observed in humans. In focusing on the highest confidence DEG that appear in both sexes in the TRAP, it is recognized that there are likely other genes that respond in the "CDCA pattern," but which are excluded from our core list due to the stringency of our criteria. For example, male mice showed a strong cell cycle related transcriptional signature at 7 weeks, as we previously observed at 10 weeks in bulk RNASeq studies using *Pkd2* models[40]. The earlier appearance of the enhanced cyst cell proliferation phenotype in males may be correlated with the more rapid polycystic kidney disease progression in male mice. It can be hypothesized that this transcriptional signal will appear in female mice at a later stage when further progression toward cysts has occurred, but it is absent from our male-female overlap gene set at 7 weeks. The set of 167 DEG common to both sexes that follow the CDCA transcriptional pattern represents a high confidence set of early ADPKD-related in vivo alterations in the translatome, but not a comprehensive list.

The "CDCA pattern" DEG were further prioritized based on expression along the entire tubule with TPM > 1.0[31]. The latter subset of 73 DEG contained several genes with cilia-related functions as well as genes with undefined functions but with high degrees of statistical association with the "CDCA pattern". The finding that among nearly 17,000 transcripts sequenced, the 73 DEG identified by the above unbiased paradigm included *Pkd2* provides a degree of assurance that this set of DEG contain transcripts with significant functional relevance

to ADPKD pathobiology. Beyond that, several of these DEG retain *Pkd1* genotype-dependent differential expression in primary ciliated kidney cell cultures. The concordance of these in vitro expression patterns with the in vivo cell-specific transcriptional changes suggests that with further selection, optimization and validation, the "CDCA pattern" translatome will define a modest sized "multigenic transcriptional signature" for polycystin function as it relates specifically to polycystic kidney disease, agnostic to knowledge of specific functional mechanisms. This in vivo validated transcriptional signature can also be applied to validation of other polycystic kidney disease model systems such as organoids[65].

*Glis2* was selected for biological validation based on its robust expression along the entire nephron, its association with a ciliopathic nephronophthisis phenotype and its strong statistical association with the "CDCA pattern" TRAP gene set. *Glis2* had not previously been considered a target of CDCA activity or an effector of polycystin-related functional pathways in ADPKD. Our finding and validation of *Glis2* in these roles highlights the value of the unbiased in vivo TRAP RNASeq approach to fill gaps in understanding of ADPKD mechanisms. Although *Glis2* has been implicated in an array of biological functions [e.g. refs. 66,67], it is best characterized in its role in the mammalian kidney[27,29]. We found that Glis2 protein expression is highest in the first two postnatal weeks in the mouse kidney, confirming that the previously reported increased transcript levels during development are correlated with protein levels[51]. Although of unknown significance, it is noteworthy that this pattern parallels the developmental regulation of polycystin-1 expression[68] and the developmental switch in the pace of cyst growth following *Pkd1* inactivation[52]. Initial reports suggested that Glis2 may be localized in cilia in addition to the nucleus[33], offering an attractive parallel with Gli transcription factors in the Hedgehog pathway; however, the cilia localization of Glis2 was not corroborated in later studies[69] nor in the cilia proteome[70] nor by our data. Glis2 has been proposed as a regulator of Hedgehog signaling required for maintaining kidney tubular cells in a differentiated state[51]; however, we have previously shown that the Hedgehog pathway has no role in ADPKD cyst progression in orthologous models[71] so it is unlikely that this proposed role for Glis2 is related to its effector function in polycystin dependent signaling.

Interestingly, *Glis2* inactivation suppresses cyst formation due to loss of cilia in an early onset model of kidney selective inactivation of *Kif3a*[54]. While the cadence and mechanisms of cyst formation due to cilia inactivation differ from polycystic kidney disease resulting from inactivation of *Pkd1* or *Pkd2*[19], this observation does raise the intriguing possibility that Glis2 may be a common downstream effector for both mechanisms. Although a speculative formulation, it is possible that the absence of cilia supports low-level activity of cellular effectors of CDCA due to a general loss of regulation; this contrasts with the strong maladaptive cyst-driving signal that CDCA emanating from intact cilia lacking polycystins offers. Glis2 may be an effector of both the high amplitude CDCA signal in ADPKD (intact cilia, absent polycystin) and

the low amplitude signal that occurs when cilia are lost (absent cilia, intact polycystin). Such a hypothesis raises the possibility that transcriptional targets of Glis2 are common elements of kidney cyst formation from diverse genetic causes.

*Glis2*[-/-] mice develop progressive nephronophthisis-like tubulointerstitial fibrotic kidney disease associated with epithelial cell senescence, but with relatively few reported extra-renal defects[33,50,54,67,72]. Similarly, patients with *NPHP7* develop early onset end stage kidney disease, and although it is possible that subtle extrarenal manifestations also occur, these have not been reported to date[33,34]. *Glis2* has been reported to be upregulated in mouse models of nonalcoholic steatohepatitis (NASH), and knockdown has reversed NASH associated transcriptional programs[66]. In our studies, kidney selective conditional inactivation of *Glis2* or systemic treatment with Glis2-ASO did not result in any discernible deleterious kidney phenotype. The absence of tubulointerstitial disease and changes suggestive of cellular senescence in *Glis2* adult inactivation and ASO inhibition models indicates that these features may only occur with homozygous germline null mutations when Glis2 is absent throughout development. Overall, these data leave open the possibility that pharmacological inhibition of *Glis2* in adults may be well tolerated both systemically and in the kidney.

One of the factors limiting progress in understanding polycystin function is the lack of in vitro model systems that reliably recapitulate in vivo polycystic kidney disease-related molecular phenotypes as a "readout". Cultured cells are an adequate model for assessing trafficking and biophysical properties of polycystin proteins[73-75]. More complex systems such as three-dimensional cell culture[76] or organoids[65,77] that enable assessment of integrated biological phenotypes (i.e., cysts) have to date not enabled discovery of unique molecular events directly related to polycystin-dependent cyst formation in vivo. *Glis2* transcript and nuclear protein expression is an in vitro molecular readout that is directly correlated with genetic factors that determine in vivo cyst formation. Inactivation of either polycystin results in upregulation of Glis2 in vivo and in vitro while genetic interventions that suppress cyst growth in vivo, including mutation of several cilia-related genes, similarly suppress increased expression of Glis2 in vitro. These findings also show that the primary cell culture systems, grown under conditions that support cilia formation, recapitulate elements of the in vivo cilia dependent transcriptional phenotype associated with cyst formation. The discovery of the value of both the primary cells as an assay system and at least *Glis2* as a dependable, biologically relevant readout, should facilitate in vitro discovery screens for modulators of polycystin function.

A transcriptional change related directly to polycystin function as provided by *Glis2* does not necessarily indicate functional importance. We established the functional importance of *Glis2* in ADPKD pathogenesis through a series of genetic experiments in mouse models of ADPKD. Inactivation of *Glis2* suppressed cyst formation in two early onset models of ADPKD based on *Pkd1* and two adult models based on either *Pkd1* or *Pkd2*. The protection in the adult inducible model of *Pkd1* was particularly complete and could be interpreted in the absence of confounding by the nephronophthisis phenotype that occurs in germline *Glis2*[-/-] mice. Since Glis2 is a transcription factor, it is reasonable to hypothesize that its effects on cyst growth are related to its transcriptional targets. Glis2 has been proposed to act as a transcriptional repressor[29,78] although it is important to note that its function, like that of many transcription factors, may be context dependent. A critical next step will be to identify targets of Glis2 activity specifically in the context of inactivation of the polycystins. We also addressed an inherent limitation of models that use the Cre/*lox*P system which results in simultaneous inactivation of large swaths of contiguous cells along the kidney tubule. This mechanism differs from the more spatially discrete sporadic somatic inactivation events thought to underlie cyst initiation in human ADPKD. We used the

*Pkd2*[WS25/-] mice which develop cysts following sporadic random second hit events akin to the human disease to extend validation that *Glis2* inactivation suppresses cyst formation in non-Cre dependent models. These genetic interaction studies establish *Glis2* not only as a marker for CDCA, but also as an effector of CDCA. We do not have a direct mechanism connecting polycystin function to *Glis2* expression so there are certainly additional intervening molecular components of CDCA, some likely in cilia, that remain essential to identify. If these intervening steps involve signaling events linked to posttranslational modifications, for example, they will not be discovered by interrogating transcriptional changes. Nonetheless, knowledge of a second critical molecular component of CDCA downstream of polycystins, i.e., *Glis2*, offers a significant step forward in devising strategies to fill in the remaining gaps.

We assessed whether *Glis2* is a suitable therapeutic target in preclinical model systems. We found that an ASO mediating degradation of *Glis2* mRNA was effective in reducing polycystic kidney disease severity and attendant tissue level changes including cyst cell proliferation and inflammatory and pro-fibrotic changes in the kidney. These findings suggest that inhibition of *Glis2*, to an extent achievable by pharmacologic rather than genome level genetic inactivation, can significantly impact the course of ADPKD in an orthologous gene model system. Furthermore, a drug with specificity for *Glis2* may be tolerated given the limited known extra-renal effects even in a germline null background. There are a few notable elements to our preclinical study design. Since we anticipated that the ASO would be most effective in reducing *Glis2* transcripts in the proximal tubule[57,58], we selected a model in which *Pkd1* was only inactivated in the proximal tubule in the kidney. Given that our *Glis2* dual genetic inactivation data was effective in reducing cysts in the proximal and distal nephron, we expect that compounds able to achieve *Glis2* inhibition along the entire nephron would likewise be effective in suppressing cyst growth along the entire nephron. Lastly, we began treatment with ASO soon after inactivation of *Pkd1* before cysts had formed. This choice eliminated potential confounding variation in drug availability in more advanced cysts that may not receive glomerular filtrate. This was beneficial in a proof-of-concept studies, but future studies should determine if inhibition of Glis2 at later stages of ADPKD can nonetheless slow disease progression.

Overall, this study fills several gaps in the ADPKD field. First, it provides a translatome dataset that most closely defines the early, cell autonomous in vivo changes associated with polycystic kidney disease. Second, it identifies *Glis2* transcript and protein as an in vivo and in vitro marker of polycystin function specifically related to polycystic kidney disease progression. Third, it defines primary cells cultured from mouse kidneys as a validated in vitro system in which to assay Glis2 expression as an indicator of polycystin function. Fourth, it establishes *Glis2* as a functional target of CDCA cyst-promoting activity following polycystin inactivation. Finally, it offers a proof of concept that pharmacological targeting of *Glis2* can suppress polycystic kidney disease based on studies in preclinical models.

## Materials and methods
### Mouse strains and procedures
This research was conducted in strict accordance with the guidelines and regulations for the care and use of laboratory animals as set forth by the Institutional Animal Care and Use Committee (IACUC) at Yale University. All experimental protocols involving animals were reviewed and approved by the IACUC. Efforts were made to minimize animal suffering and to reduce the number of animals used.

Following strains of mice were used in this study: *Pkd1*[fl 41], *Pkd2*[fl 79], *Kif3a*[fl 80], *Pax8*[rtTA] (JAX Strain # 007176), *TetO*[Cre] (JAX Strain # 006234), *Tulp3*[fl 20] (kindly provided by Karel Liem, Yale University), *Pkd2*[+/- 81], *Pkd2*[WS25 55], *Ift88*[fl] (JAX Strain # 022409)[82], *Glis2*[-/- 50], *UBC*[Cre-ERT2] (JAX Strain # 008085), *Pkhd1*[Cre 83], *ACTB*[Cre] (JAX Strain # 003376), *ACTB*[Flp]

(JAX Strain # 005703), *ROSAᵐᵀ/ᵐᴳ* (JAX Strain # 007676), *R26ᴿᵖˡ¹⁰ᵃ* (JAX Strain # 024750). *Glis2ᶠˡ* mice were produced in this study. Three *Glis2*-targeted *Glis2ᵗᵐ¹ᵃ⁽ᴱᵁᶜᴼᴹᴹ⁾ᴴᵐᵍᵘ* embryonic stem (ES) cell lines (HEPD0539_2_B10; HEPD0539_2_D09; HEPD0539_2_G11) were obtained from the European Mouse Mutant Cell Repository (EuMMCR, Munich, Germany). All three ES cell lines have identically targeted *Glis2*, which in its final form has exon 3 is flanked by *loxP* sites. Two ES cell clones (HEPD0539_2_B10, HEPD0539_2_D09) were expanded and injected into blastocysts to obtain founders. F1 progeny with the *Glis2* targeted allele were crossed with *ACTBᶠᴸᴾ* mice to remove the LacZ-neomycin cassette through the germline. An exon 3 deleted *Glis2ᐃᴱˣ³⁻* null allele was obtained from *Glis2ᶠˡ* by mating with *ACTBᶜʳᵉ*. Mating strategies for allele combination and doxycycline and tamoxifen Cre induction protocols have been previously described[19,36]. All strains were back-crossed at least four generations with C57BL/6J and are expected to be at least 90% congenic C57BL/6J. Mice of both sexes were used. All animals were maintained in secure, intact, clean, fully assembled, barcoded, static micro-isolator cages. Temperature was maintained between 20 °C and 26.1 °C and humidity between 30% and 70%, with 12 h:12 h light:dark cycles. All animals were used in accordance with scientific, humane, and ethical principles and in compliance with regulations approved by the Yale University Institutional Animal Care and Use Committee (IACUC). Genotyping was done on DNA isolated from toe clips. Mice were euthanized according to standard protocols approved by Yale IACUC. Experimental mice were euthanized with ketamine (100 mg/kg) and xylazine (10 mg/kg). Blood was collected by ventricular puncture. One kidney was snap-frozen for protein and mRNA extraction, and the other kidney was fixed in situ by perfusion through the heart with 4% paraformaldehyde (PFA, MP Biomedicals, Cat. no. 0215014601) in 1× PBS. Sera were separated using Plasma Separator Tubes with lithium heparin (BD Biosciences, BD Vacutainer, Cat. No. 364606). Serum urea nitrogen was analyzed by the George M. O'Brien Kidney Center at Yale. Genotyping primer sequences are provided in Supplementary Data 5.

## Copy number determination of Pax8ʳᵗᵀᴬ and TetOᶜʳᵉ

Mouse genomic DNA was diluted to 50 ng/μl and mixed with iTaq Universal SYBR Green Supermix (BioRad, Cat. no. 1725121) and qPCR primers specific to *Pax8ʳᵗᵀᴬ* or *TetOᶜʳᵉ* and 18S rRNA. The primer sequences used were:

Pax8E1QF1 5'-GGGAAGAGAAGGGGTTGAAGG-3'
Pax8E1QR1 5'-ACTCAGCAGGCCAGGAAGTA-3'
TetCreQF2 5'-CACGCTGTTTTGACCTCCAT-3'
TetCreQR2 5'-CTCTGCCCCTCGACTCTAGA-3'

qPCR was performed with a CFX96 Touch Real-Time PCR Detection System (BioRad). The expression of *Pax8ʳᵗᵀᴬ* and *TetOᶜʳᵉ* was normalized to *18S* by the $2^{-\Delta\Delta CT}$ method.

## Pkd1 deletion efficiency in kidney tissues

Mice were euthanized and one kidney was snap-frozen in liquid nitrogen. DNA was extracted with DNeasy Blood & Tissue kit (Qiagen, Cat. no. 69504). DNA concentration was measured by Nanodrop 2000 (Thermo Fisher Scientific, Cat. no. ND2000). An equal starting amount of DNA (50 ng/μl) was used for all samples. A common forward primer (Pkd1X1F1) was designed to the region upstream of the first *loxP* site within intron 1. A reverse primer (Pkd1X1R1) was designed to the region immediately downstream of the first *loxP* site also within intron 1. A second reverse primer (Pkd1X2R4) was designed to the region downstream of the second *loxP* site within intron 4. The following primer were used in *Pkd1* deletion efficiency assay:

Pkd1X1F1: 5'-TCACGGAAGAGCAGCCTGCCTT-3'
Pkd1X1R1: 5'-TCTGTGTACTGGGGCACAGCCT-3'
Pkd1X2R4: 5'-AGCACCTGAGCTGTTGTCAGGG-3'

*Cre* activation results in deletion of exons 2 to 4 of *Pkd1*. The product from primer set Pkd1X1F1/ Pkd1X1R1 yields a 119 bp PCR in wild type (no *loxP*) DNA and a 319 bp in the undeleted *loxP* allele. Pkd1X1F1/ Pkd1X2R4 will result in a PCR product of 425 bp only obtained after Cre mediated recombination. The PCR products were resolved on a 1.5% (w/v) agarose gel (Sigma-Aldrich, Cat. no. A9539), imaged with ChemiDoc XRS+ Imaging System (BioRad, Cat. No. 1708265) and the intensity of the PCR bands in captured TIFF images were quantitated using ImageJ (NIH). A ratio of the densitometric intensity of the *loxP*-deleted 425 bp band to the sum of the densitometric intensities of the *loxP*-undeleted 319 bp band plus the deleted 425 bp band was taken as indicating the deletion efficiency for each sample.

## Translating ribosome affinity purification (TRAP)

TRAP was performed largely by the method described by[62] with a few modifications. Briefly, fresh kidney in lysis buffer [20 mM HEPES-KOH (Fisher Scientific, Cat. no. AAJ16924AE), 5 mM MgCl₂ (Thermo Fisher Scientific, Cat. no. AM9530G), 150 mM KCl (Thermo Fisher Scientific, Cat. no. AM9640G), 0.5 mM DTT (AmericanBio, Cat. no. AB00490-00005), Mini Complete EDTA-Free Protease inhibitor (Roche, Cat. no. 11873580001), 100 μg/ml Cycloheximide (Sigma-Aldrich, Cat. no. C7698), 40 U/ml RNasin Plus Ribonuclease Inhibitor (Promega, Cat. no. N2615), and 20 U/ml SUPERase•In RNase Inhibitor (Thermo Fisher Scientific, Cat. no. AM2696)] was homogenized using a bead mill homogenizer (Precellys® evolution, Bertin Instruments) at 5500 rpm for 20 s twice with 1 min ice-bath in between. The homogenate was then centrifuged at 2000 × *g* at 4 °C for 10 min and the soluble fraction was mixed with 10% NP-40 (Thermo Fisher Scientific, Cat. no. 28324) prior centrifugation at 20,000 × *g* at 4 °C for 10 min. The supernatants of the homogenates were mixed with anti-GFP monoclonal antibodies (Monoclonal Antibody Core Facility, Memorial Sloan-Kettering Cancer Center, New York, Cat. no. Clone: 19C8 and 19F7) pre-conjugated to magnetic beads (Dynabeads MyOne Streptavidin T1, Thermo Fisher Scientific, Cat. no. 65601) for overnight immunoprecipitation at 4 °C with gentle rotation. The magnet bound fraction was washed three times with 1 M KCl buffer containing 20 mM HEPES-KOH (pH 7.4), 5 mM MgCl₂, 1 M KCl, 1% NP-40, 0.5 mM DTT and 100 μg/ml Cycloheximide. Any residual DNA was removed with DNase I digestion (Qiagen, Cat. no. 79254) and RNA was isolated with RNeasy MinElute columns (Qiagen, Cat. no. 74204). The RNA eluted in RNase-free water were stored at -80°C until sequencing. mRNA library preparation and sequencing were performed by the Yale Center for Genome Analysis. The sequencing library was prepared using the ribosomal depletion method (KAPA RNA HyperPrep Kit with RiboErase, Roche) and sequencing was run on Illumina's NovaSeq 6000 platform using 150 bp paired end reads with read depth 100 million reads per sample.

## TRAP RNASeq analysis

Raw TRAPSeq fastq files were processed using fastq tool (version 0.20.0)[84]. Sequencing reads with low-quality bases were trimmed or filtered with the default settings. Cleaned reads were aligned using STAR (version 2.7.9)[85] with the mouse reference genome gencode version GRCm38.p6 with vM25 gene annotation. Expression quantification for aligned reads was performed using featureCounts (version 2.0.0)[86]. We used an expression threshold of ≥6 read counts in at least 20% of samples for each sex; genes not meeting this expression threshold were eliminated from downstream analyses. The filtered read counts matrix was normalized by the transcripts per million (TPM) method. Detection of differentially expressed genes was performed using R package DESeq2 (version 1.30.1)[87] and the Benjamini−Hochberg procedure was used for multiple test correction with FDR ≤ 0.05 used as the significance threshold for detection of differentially expressed genes (DEG). The DEGs in the Pkd1ᴷᴼ mutant that when compared to the noncystic and Pkd1ᴷᴼ+ciliaᴷᴼ have the same change direction were defined as "CDCA pattern" DEGs. Heatmaps were generated using the "pheatmap" package in R, with genes and

samples hierarchically clustered using the Pearson correlation method. For the heatmaps generated from microdissected kidney tubule bulk RNASeq data[31], CDCA pattern genes were first ranked based on how many of the 14 microdissected tubule segments showed expression TPM > 1.0, and then further sorted by their significance level (adjusted $p$ value) in the comparison between Pkd1[KO] and non-cystic male samples. The analysis pipeline scripts are available at: https://github.com/StefanSomloLab/TRAPseq.

## Primary cell culture
Primary cells from kidney were isolated using enzymatic digestion procedures. Briefly, adult mouse kidney tissue was collected and minced into small pieces maintaining sterile conditions. Minced tissue pieces were transferred to GentleMACS C-tube (Miltenyi Biotech, Cat. no. 130093237) containing 5 ml of freshly prepared dissociation buffer (DMEM containing D, P, A, Y enzymes). The C-tube was inserted into GentleMACS Octo-Dissociator (Miltenyi Biotech, Cat. no. 130096427) and run with pre-set program "37 multi E 01" (run time of 30 mins). After cell dissociation, 5 ml of DMEM with 10% FBS was added to stop digestion, and the mixture was passed through 70 μm strainer followed by centrifugation at $400 \times g$ for 10 min. The cell pellet was resuspended in REGM medium (Lonza, REGM Renal Epithelial Cell Growth Medium BulletKit, Cat. no. cc-3190) and cells were directly seeded for experiments. Cells were let attach for 48 h and then treated with doxycycline (Sigma, Cat. no. D9891-100G) 1 μg/ml for 72 h, followed by serum starvation in 0.1% FBS containing media for 24 h prior to preparation of protein or RNA. For quality control purposes, cells were seeded in parallel, treated similarly to experimental conditions and tested to assess Cre dependent deletion by genotyping after doxycycline treatment. All experiments were done with freshly isolated primary cells without passaging. This reduced the chance of increasing fibroblast overgrowth in the culture. Fibroblast overgrowth was observed in passaged primary cells. Epithelial composition of isolated primary cells was tested by immunofluorescence for epithelial, fibroblasts markers and lineage markers, e.g., ZO1, COL1A1, PAX2 and PAX8.

## Cell lines
Cell line generation from primary cells was performed using lentivirus of SV40 large T antigen (Gentarget, Cat. no. LVP016-Hygro). Hygromycin antibiotic was used for selection of the transduced and transformed cells. Cell lines were validated by genotyping and by protein analysis for target alleles. IMCD3 cells (ATCC CRL-2123) were cultured in DMEM/F12 (Thermo Fisher Scientific, Cat. no. 10565042) with 5% FBS. IMCD3-3F5, 3F6 *Pkd1* KO clones and the Cas9 parental line have been described previously[49]. IMCD3 cells were transduced with lentiviruses containing Glis2-EGFP or Glis3-EGFP to generate stable cells. HEK293T (ATCC CRL-3216) were cultured in DMEM high glucose (Thermo Fisher Scientific, Cat. no. 11965092) with 10% FBS. Cells were either transiently transfected with indicated plasmids or stable expression was performed using lentiviruses of the indicated constructs and appropriate selection antibiotic. All cell lines were serum starved 24 h in 0.1% FBS media prior experimentation.

## Generation of anti-Glis2 antibodies
Anti-Glis2 antibodies (named YNG2) were custom synthesized by Covance (now Labcorp Drug Development, NC, USA). Two rabbits were injected with 500 μg of each of two peptides corresponding to amino acids 31-46 and 462-476 of the mouse Glis2 sequence (NCBI Accession NP_112461) followed by three rounds of boosting. Serum was collected after final boost and antibodies were obtained from the pooled serum by peptide antigen affinity purification.

## Plasmids, transient transfection, and lentiviral infection
Mouse *Glis2* (NCBI Accession NM_031184) was cloned into several vectors with different epitope tag combinations: pLenti CMV GFP Blast

(Addgene, Cat. no. #17445) with N-terminal V5 tag and C-terminal EGFP tag; pLVX-TetOne (Takara, Cat. no. 631846) with C-term FLAG tag; pHTC HaloTag® CMV-neo was used to clone *Glis2* with C-term Halo tag. Human *GLIS2* (NCBI Accession NM_032575) was purchased as GLIS2-HaloTag® human ORF in pFN21A (Promega, Cat. no. FHC03277). Mouse *Glis3* (NCBI Accession NM_175459) was cloned into pLenti CMV GFP Blast with C-term EGFP tag. The cilia marker Nphp3[(1-200)]-mApple was made by amplifying a fragment corresponding to amino acid 1-200 of mouse *Nphp3* (NCBI Accession NM_028721.3) from mouse kidney cDNA and combining it in-frame with mApple using PCR, followed by subcloning into a pCDH-Hygro vector which was modified from pCDH-EF1-MCS-IRES-Puro (System Biosciences). All constructs were validated by sequencing and immunoblot expression analysis. Transient transfection of HEK293T cells was done with Lipofectamine 2000 (Thermo Fisher Scientific, Cat. no. 11668030). Stable gene expression was achieved using lentivirus transduction and selection with appropriate antibiotic.

## Protein preparation, electrophoresis and immunoblotting
Total lysates from cell pellets or tissues were prepared for immunoblotting using 1x Red Loading Buffer Pack (Cell Signal Technology, Cat. no. 7723S) according to manufacturer's instructions. Briefly, cell pellets or tissue were suspended in the buffer without DTT and sonicated three times, each in 10 pulses of 1–2 s, with a probe sonicator (Misonix, XL-2000; power setting 6). Protein amounts were quantified using a bicinchoninic acid assay (BCA) kit (Thermo Fisher Scientific, Cat. no. 23225). DTT was added to normalized concentrations of protein and samples were boiled for 5 min prior to SDS-PAGE. Fractionation of cells or tissue into cytosolic and nuclear parts was done using either NE-PER™ Nuclear and Cytoplasmic Extraction Reagents (Thermo, Cat. no. 78833) with manufacturer's instructions or by a non-kit method. For the latter, cells pellets were resuspended in ice cold lysis buffer containing 100 mM Tris pH 7.4, 20 mM NaCl, 10 mM EDTA, 0.1% Triton X-100 and incubated on ice for 5 min. For fractionation of tissues, tissue in the same lysis buffer was homogenized in a bead mill homogenizer (Precellys® evolution, Bertin Instruments) at 6800 rpm for 30 s twice with 1 min ice-bath in between. Cell or tissue lysates were resuspended and centrifuged at $21,000 \times g$ for 5 min. The supernatant was transferred to a new tube and comprised the cytosolic fraction. The pellet was resuspended in 1X Red Loading buffer and followed the protein extraction steps described above.

Lysates were run on 4–20% Mini-PROTEAN TGS Precast Protein gels (Biorad, Cat. no. 4561094) for all except for PC1 western blots. PC1 western blots were run on NuPAGE 3-8% Tris-Acetate 1.5 mm Mini Protein gels (Thermo Fisher Scientific, Cat. no. EA0378BOX). Proteins were transferred either on Nitrocellulose membrane (Biorad, Cat. no. 1620115) or PVDF membrane (Biorad, Cat. no. 1620177). Precision Plus Protein™ Dual Color Standards (Biorad, Cat. no. 1610374) or PureView Prestained Protein Ladder (Azura Genomics, Cat. no. AZ-1142-2) were used as size markers. Ponceau S (SCBT, Cat. no. sc-301558) was used to reversibly stain membranes to determine loading and proper transfer from the gel to the membrane. Membrane stripping for re-probing was done using Restore PLUS Western Blot Stripping Buffer (Thermo Fisher Scientific, Cat. no. 46430). Briefly, membranes were washed with running water multiple times. Then, membranes were incubated with buffer enough to cover the membrane at 55 °C for 10 min followed by washing with running water multiple times. Membranes were then blocked with 5% non-fat dry milk for 10 min, followed by incubation with primary and secondary antibodies. Chemiluminescent signals from membranes were determined using either Clarity Western ECL Substrate (Biorad, Cat. no. 1705060) or SuperSignal Femto Maximum Sensitivity Substrate (Thermo Scientific, Cat. no. 34095) on Licor Odyssey Fc. A detailed listing of antibodies used is provided in Supplementary Data 5.

## RNA isolation and qRT-PCR

Total RNA from kidney and cells was isolated using either RNeasy Mini Kit (Qiagen, Cat. no. 74104) or Trizol (Themo Fisher Scientific, Cat. no. 15596026) based method followed by column purification. Cells were serum starved overnight before the day of RNA preparation. For tissue homogenization, kidneys were homogenized in Trizol by bead mill homogenizer (Precellys® Evolution, Bertin Instruments) at 6800 rpm for 30 s twice with 30 s in ice-bath in between. Two µg of total RNA was used for cDNA synthesis using iScript cDNA synthesis kit (Biorad, Cat. no. 1708890). Quantitative RT-PCR was performed using iTaq Universal SYBR green Supermix (Biorad, Cat. no. 18064022) or AzuraView Green Fast qPCR Blue Mix, LoRox (Azura Genomics, Cat. no. AZ2320) in CFX96 Touch Real-Time PCR detection system (Biorad, USA). *Gapdh* or *18s* was used for normalization. qRT-PCR primer sequences are provided in Supplementary Data 5.

## Immunocytochemistry and cystic index measurement

Immunocytochemistry was performed on primary cells, IMCD3 and HEK293T as per standard protocols. Cells were plated on 4-well chambered dishes (Cellvis, Cat. no. C4-1.5P). All adhered cells were washed gently two times with 1X PBS followed by fixation with 4% PFA in PBS for 20 min. Cells were washed 3 times with 1X PBS and permeabilized with 0.25% Triton X-100 for 5 min followed by blocking with 5% BSA in PBS. Primary antibody was diluted in 5% BSA and incubated at room temperature for 1 h. Cells were washed 3 times with PBS followed by secondary antibody in 5% BSA for 1 h at room temperature. Cells were washed another 3 times with PBS and Hoechst was added to stain nuclei. Microscopy of non-fixed live cells was done on adhered cells. Cells were washed in sterile conditions in 1X PBS and imaging was done on cells with 1X PBS to reduce autofluorescence from media. All images were acquired on Nikon Eclipse Ti (Nikon Instruments Inc, Japan) equipped with Yokogawa CSU-W1 spinning disc and Andor lasers (Andor Technology, UK). For movies, gamma correction was done to reduce background noise. For tissue immunohistochemistry, cryosections (5–7 µm) were used according to standard procedures. Cystic index was calculated as previously described[19] using frontal plane sections of kidneys processed for hematoxylin and eosin and scanned by light microscopy under the control of MetaMorph software (Universal Imaging, version 7.7).

## Blood urea nitrogen (BUN) measurements

BUN was measured at the George M. O'Brien Kidney Center at Yale Core using Stanbio Urea Nitrogen (BUN) Procedure 0580 quantitative colorimetric determination of urea nitrogen in serum and plasma. The method is based on the diacetylmonoxime (DAM) methodology. Samples were read on the Excel Chemistry Analyzer (Stanbio Laboratory, Boerne Texas).

## Proliferation assays

Proliferation was measured by EdU incorporation and Ki67 staining. For EdU incorporation, mice received 50 mg/kg of 5-ethynyl-2'-deoxyuridine (Invitrogen, Cat. no. A10044) by intraperitoneal injection 4 h before euthanasia. Kidneys were fixed with 4% PFA overnight and embedded in OCT after 30% sucrose infiltration and processed for immunofluorescence. EdU staining was performed with Click-iT™ EdU Imaging Kit with Alexa Fluor™ 647 (Invitrogen, Cat. no. C10086,). KI-67 expression in kidney was detected with rabbit anti-Ki67 antibody (Thermo Fisher Scientific, RM-9106-S1, dilution 1:200). Numbers of EdU and Ki67-positive nuclei were counted amongst at least 1,000 DBA or LTA positive cells per animal.

## Senescence associated-β-Galactosidase (SA-β-Gal) staining in tissue

SA-β-Gal staining was done using the Senescence β-Galactosidase Staining Kit (Cell Signaling, Cat. no. 9860), adapted for tissue, according to manufacturer's instructions. Briefly, kidneys were collected after perfusing mice with 4% PFA in 1x PBS followed by fixation in 4% PFA at 4 °C overnight. Following fixation, kidneys were treated with 30% sucrose solution in 1x PBS at 4 °C overnight and embedded in OCT compound. Frozen sections (5–7 µm) were air dried for 15 min and washed with 1x PBS three times, then incubated with fresh SA-β-Gal staining solution at 37 °C for 24 h. Sections were counterstained with Nuclear Fast Red solution (Statlab Medical Products, Cat. no. STNFR100).

## RNAScope multiplex fluorescent in situ hybridization (RNAScope-FISH)

RNAScope® Multiplex Fluorescent Kit v2 (Advanced Cell Diagnostics) was used to perform RNA-probe based fluorescent in situ hybridization (FISH) according to the manufacturer's instructions. Briefly, 5–7 µm thick Tissue-Tek OCT Compound (Sakura, Cat. no. 4583) embedded cryosections of mouse kidneys were fixed in 4% PFA for 15 mins at 4 °C prior to serial dehydration for 5 min each with 50, 70 and 100% ethanol. Sections were then treated with hydrogen peroxide for 10 min at room temperature, followed by incubation in RNAscope target retrieval solution for 15 min at 98–102 °C and then treated with RNAscope Protease III for 30 min at 40 °C. RNA-specific probes targeting mouse *Glis2* (Mm-Glis2, Cat. no. 405621) and mouse megalin (Mm-Lrp2, Cat. no. 425881; Advance Cell Diagnostics Inc., USA) were incubated for 2 h at 40 °C. Sections were counterstained with DAPI, and mounted with fluorescent mounting media with anti-fading agent DABCO (Fluoro Gel with DABCO, EMS, USA). All the slides were kept at 4 °C prior to image acquisition. Images were obtained using a confocal microscope (Nikon Eclipse Ti, Japan) under the control of NIS-Elements AR 4.30.02 software (Nikon).

## Single molecule fluorescence in situ hybridization (smFISH)

Kidneys were removed and fixed in 4% PFA in 1 x PBS for 4 h and subsequently incubated in a 30% sucrose, 4% PFA in 1 x PBS solution at 4 °C overnight with gentle agitation. Fixed kidneys were embedded in Tissue-Tek OCT Compound (Sakura, Cat. no. 4583) and 5 µm thick sections placed onto poly L-lysine coated coverslips. Probe libraries for *Glis2* and *Lrp2* were designed using the Stellaris FISH Probe Design Software (Biosearch Technologies). Probe sequences are provided in Supplementary Data 5. Probes for *Glis2* were coupled to Quasar® 670 Dye and probes for *Lrp2* (megalin) were coupled to CAL Fluor® Red 590 Dye. Kidney sections were hybridized with smFISH probe sets based on the Stellaris RNA FISH protocol (Biosearch Technologies). Images were obtained using a confocal microscope (Nikon ECLIPSE Ti, Japan). under the control of NIS-Elements AR 4.30.02 software (Nikon). A pipeline in CellProfiler[88] was written that segmented and identified Megalin puncta, Glis2 puncta and nuclear signal. The intensity of the Glis2 puncta normalized to the number of nuclei were used to calculate signal intensity.

## Mouse Glis2 antisense oligonucleotide (Glis2 ASO) development, characterization, and administration

The *Glis2* ASOs targeting murine *Glis2* were designed, synthesized, and tested in a series of in vitro and in vivo screens and evaluated for efficacy and tolerability in 8–10-week-old male C57BL/6J mice at a dose of 50 mg/kg per week for 3 weeks by Ionis Pharmaceuticals, Inc. (Carlsbad, CA). ASOs that displayed robust reduction of kidney *Glis2* mRNA without inducing elevations in plasma transaminases and mRNA biomarkers of renal tubular injury (KIM-1 and NGAL) were selected for additional in vivo studies in models of ADPKD. The *Glis2* ASO identified from the screening exercise evaluated in the disease models had the following sequence: 5'- CCTTATAAGCTTCTGC -3', with the underlined sequences indicated the 2',4'-constrained ethyl-D-ribose (cEt) modified bases. Additionally, a control ASO was used: 5'- ACGATAAC GGTCAGTA. ASO was dissolved and diluted in sterile D-PBS and

sterilized by filtration through a 0.22 μm filter. ASO was administered by intraperitoneal injection (IP) at a dose of 50 mg/kg beginning at 5 weeks of age following the end of tamoxifen induction (P28-P35). ASO was administered twice during week 5 (initial week), and once a week from 6 weeks of age through 18 weeks age. Mice were euthanized 1 day after the final dose.

## Sample size and statistics

Sample size and power calculations were performed prospectively using STPLAN (v.4.5; University of Texas MD Anderson Cancer Center). Calculations were based on the following inputs: we selected a significance level (α) as 0.05 (one sided) and 0.80 power (1 − β) as our threshold. For kidney weight to body weight ratio in $Pkd1^{fl/fl}$; $Pax8^{rtTA};TetO^{Cre}$ mice at age 18 weeks after receiving oral doxycycline from P28-42, a mean of 10.2% with s.d. ±3.3% was empirically derived from our earlier data. Our alternative hypothesis was that $Glis2^{fl/fl}$; $Pkd1^{fl/fl};Pax8^{rtTA};TetO^{Cre}$ mice would result in at least a 40% decrease in the kidney weight to body weight ratio. Under these expectations, 6 mice could achieve 80% power to detect the difference between the null hypothesis and the alternative hypothesis. For kidney weight to body weight ratio in $Pkd1^{fl/fl};UBC^{CreERT2}$ mice at age 18 weeks after receiving tamoxifen from P28-35, a mean of 6.2% with s.d. ±3.0% was empirically derived from our earlier data. Our alternative hypothesis was that $Pkd1^{fl/fl};UBC^{CreERT2}$+Glis2-ASO mice would result in at least a 40% decrease in the kidney weight to body weight ratio. Under these expectations, 11 mice could achieve 80% power to detect the difference between the null hypothesis and the alternative hypothesis.

Most of the quantitative data were analyzed using one-way analysis of variance (ANOVA) followed by Tukey's multiple-comparison test using GraphPad Prism 9.5.1 software. The data of Fig. 3d, f, h were analyzed using two-tailed, unpaired Student's $t$ test. The data in Supplementary Fig. 5a were analyzed using two-way ANOVA followed by Dunnett's multiple-comparison test. All data are presented as mean ± s.e.m. $p < 0.05$ was considered the threshold for significance throughout. Exact $p$ values are provided for all $p > 0.0001$.

## Data availability

All the raw sequencing data and processed data have been deposited in the Gene Expression Omnibus (GEO) with the following accession number: GSE232556. Access is not restricted. It contains all TRAP RNASeq data reported in this study. All other data are contained in the Supplementary Information and Source Data files with this manuscript. Source data are provided with this paper.

## Code availability

No custom code was used for any part of the data processing or analysis. Details of the analytic methods described are available at https://github.com/StefanSomloLab/TRAPseq. The TRAP RNASeq can also be accessed at the "Metabolism and Genomics in Cystic Kidney (MAGICK)" (https://pkdgenesandmetabolism.org/) website.

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

## Acknowledgements

This work was supported by NIH/National Institute of Diabetes and Digestive and Kidney Diseases grants (nos. R01 DK120911, R01 DK100592, and RC2 DK120534 to S.S.) and a grant from the Amy P. Goldman Foundation to S.S. We are grateful for the generous support from Mr. and Mrs. Robert Roth. A.M.J. is supported by the Intramural Research Program of the NIEHS, NIH (NIH Z01-ES-100485).

## Author contributions

Co-first authors C.Z., M.R. and X.T. designed and performed experiments, data acquisition and analysis, drafted figures and wrote the methods. Part of this work was performed in fulfillment of the doctoral thesis requirements for C.Z. M.R. and X.T. collated all the primary source data provided. S.L.C.P. performed the TRAP RNASeq and additional experiments and provided artwork. J.G., supervised by H.Z., was responsible for all bioinformatic analyses and the data output from the analyses. T.A.B.3rd, while an employee of Ionis Pharmaceuticals, designed and tested the ASO reagents. K.D., M.S.T., Y.C., Z.W. and F.B. performed selected experiments and provided reagents. A.M.J. provided critical reagents. M.L. contributed to data analysis and produced the website where the TRAP RNASeq and related data are presented. S.S. conceived the study, developed and supervised the overall research plan and wrote the manuscript text. All authors reviewed the text and M.R. assisted with editing.

## Competing interests

S.S. is a consultant for, and a scientific co-founder with equity interest in Sen Therapeutics. T.A.B.3rd was an employee of Ionis Pharmaceuticals at the time of this study. There are no more competing interests.
