## [Peer Review File · Nature Communications]

Glis2 is an early effector of polycystin signaling and a target for therapy in polycystic kidney diseaseREVIEWER COMMENTS

Reviewer #1 (Remarks to the Author):

Comments to the Authors

In the manuscript "Glis2 is an early effector of polycystin signaling and a target for therapy in polycystic kidney disease", Zhang and collaborators investigate the phenomenon that cyst formation following the inactivation of polycystin-1 or polycystin-2 in mice kidney tubules requires the presence of structurally and functionally intact primary cilia (a phenomenon defined by the authors as cilia dependent cyst activation or CDCA). To identify potential molecular mechanisms that could explain this observation, the authors compared the transcriptome profiles of kidney tubular cells from transgenic mice with kidney-specific genetic inactivation of Pkd1, Pkd1;Kif3a double knockouts (no cilia), and Pkd1 heterozygous mice as a control, to prioritize differentially translated genes for further studies. Among the differentially translated genes, the authors chose to focus on the gene Glis2 as a potential contributor to the CDCA, and generated several models of Pkd1- and Pkd2- Glis2 double knockouts in the presence and the absence of cilia. The authors show that genetic and transcriptional (by using an antisense oligonucleotide) inactivation of Glis2 resulted in slower cyst growth and improved kidney function in several mouse models of PKD. The authors conclude that the expression of Glis2 is a biomarker of polycystin function, a functional target of CDCA, and a potential target to treat PKD.

Numerous critical methodological, conceptual, and technical shortcomings limit the scientific soundness and relevance of this manuscript.

The first is the complete absence of mechanistic insight into how glis2 is upregulated after the genetic inactivation of PKD1/2 and how the inactivation of Glis2 in Pkd1 and Pkd2 results in a milder cystic phenotype.

The definition of cilia-dependent cyst activation (CDCA) itself is an abstract semantic construct that designates experimental observations that have not been mechanistically explained. In the absence of such mechanistic insights, it needs to be clarified why and how the loss of functional cilia invariably results in a cystic phenotype except in the absence of PC1 and PC2. Thus, the fact that cyst formation following the inactivation of PC1 and PC2 requires the presence of structurally and functionally intact primary cilia does not imply that a CDCA pathway actually exists.

Similarly, Glis2 overexpression in the absence of Pkd does not imply that Glis2 is part of a putative ciliary PKD signaling, especially considering that cilia are not required for Glis2 function (PMID: 27181777). The overexpression of Glis2 could simply be an indirect, secondary effect of Pkd inactivation, and the partial rescue of the cystic Pkd KO phenotype by Glis2 RNA interference could represent a bystander effect of the perturbation of other unrelated signaling pathways. None of these possibilities have been experimentally investigated.

The manuscript is conceptually based on the unproven assumption that the function of the polycystins in the cilia drives the cystic phenotype, excluding the role of polycystins functions in other cell compartments.

It is quite surprising that the upregulation of Glis2 and Pkd2 has not been reported in the plethora of other gene expression studies performed in PKD animal and human studies. This point has not been mentioned or discussed by the authors, as it has not been tested whether Glis2 expression also increased in human PKD cysts or human organoids.

The use of mouse models with pan-tubular inactivation of PKD is disputable, considering that in human PKD cysts arise in the distal nephron. Additionally, using such different transgenic mouse models generates variability and confusion.

The manuscript is burdened with errors, uninterpretable words, expressions, and sentences and would benefit from extensive editing.

More specific comments are listed below:

Page 2, lines 17/18: The statement “Glis2 transcript and protein is an in vivo and in vitro biomarker of polycystin function” is not supported by experimental data. Based on the data, Glis2 change of expression is associated with PC inactivation, not function.

Page 3, line 19: The effect of Tulp3 inactivation on cyst growth is debated, given contrasting results from other publications (PMID: 30799239).

Page 4, line 5: “Polycystin-dependent cyst formation” is incorrect; is it meant cilia-dependent cyst formation?

Page 4, lines 8-9: It is not clear what the novelty is in using TRAP as an exploratory analysis.

Page 4, line 18: The use of the pan-nephron Pax8rtTA;TetOCre model is debatable, given that only the distal nephron develops cysts in ADPKD.

Page 4, line 25: The sentence “To limit discovery to cell-autonomous transcriptional changes following loss of polycystin-1” is unjustified: TRAP only measures cell-autonomous changes.

Page 4, line 34: It is unclear what the meaning/relevance of “cilium organization” differentially translated genes is.

Page 4, line 35: The missed identification of pathways previously implicated in ADPKD pathogenesis may suggest the opportunity for new discoveries but also underlines the lack of validation from other studies.

Page 5, line 37 - page 6, lines 1-2: The inactivation of cilia in all mouse models causes cysts, does not suppress them. Ablation of cilia is only reported to suppress cyst growth in the concomitance of PKD inactivation (citation 19).

Page 5, lines 2-4: It is unclear why transcripts expressed along the entire tubule would be expected of candidate transcriptional changes related to CDCA. Wouldn't it be proper to focus only on genes expressed in the distal nephron, where cysts develop in humans?

Page 6, lines 26/27: why were most genes not involved in cystic kidney diseases tested by RT-PCR?

Page 7, line 33: “Developing kidney” refers to the kidney during embryonic development. The

sentence, instead, refers to Pkd2 KO in which Pkd2 is re-expressed (as per Figure 2).

Page 8, line 10: Does the expression validated Glis2 targets, such as Wnt4 and Snai1, also vary accordingly?

Page 8, lines 21/22: It should be clarified that the cyst-forming potential refers to the specific model used here, i.e., Pkd KO cells, and not to the general cyst-forming potential attributed to cilia in the context of PKD inactivation.

Page 9, lines 2 and 6: The working hypothesis is that a putative CDCA controls cyst growth in Pkd-null kidneys, not that a "polycystin function dependent cyst forming potential" exists: PCs loss of function causes cysts.

Page 9, line 28: Has it been tested if embryonic deletion of Glis/f mice reproduces the phenotype of the constitutive knockouts?

Page 10, line 23: "Glis2^{-/-}, Pkd2WS25^{-/-}, and Glis2^{-/-}; Pkd2WS25^{-/-} mice are the same genotype.

Page 12, line 29: The association between the hedgehog pathway and polycystin signaling is obscure.

Page 13, line 21: It is not clear why prioritizing differentially expressed genes in all nephron segments instead of focusing on the distal nephron only.

Page 13, line 29: "This transcriptional signature can be applied...". This needs to be experimentally proven.

Page 13, line 37: "We found". Based on the cited reference, this should read, "We confirmed".

Page 14, lines 1-2: The parallel between postnatal kidney expression of Glis2 and polycystin-1 expression is unclear.

Page 14, line 6: the statement that there are no reported extrarenal defects in Glis2 knockout mice is false. See PMID: 30523147 and PMID: 34705506. It is also not unlikely that more subtle extrarenal manifestations have been described in patients with mutations in NPHP7, given these subjects' early onset and mortality.

Page 14, lines 9-11: It would be interesting to test whether mice with Glis2 inactivation after development show signs of disease.

Page 15, line 24: The relevance of establishing Glis2 as a therapeutic target in preclinical models is unclear.

Page 15, lines 29-30: See comments to Page 14, line 6.

Page 15, line 35: The relevance of using a model of PKD with proximal tubule inactivation is unclear.

Page 16, lines 2-3: Although there may have been some variability in using ASO after the formation of cysts, this experiment should have been performed.

Page 16, lines 6-13: The statement that Glis2 expression is an indicator of polycystin function is an overstatement not supported by experimental data.

Page 16, Figure 1: Why are some genes in panel b marked in red?

Reviewer #2 (Remarks to the Author):

The reviewed manuscript titled "Glis2 is an early effector of polycystin signaling and a target for therapy in polycystic kidney disease" by Zhang et al propose Glis2 as a novel biomarker in and potential therapeutic target of ADPKD. The researchers used unbiased cell type-specific transcriptional profiling to analyze actively translating mRNA in various mouse models of ADPKD at early time points and identified 167 differentially expressed mRNA that correlated with the CDCA pattern in both male and female mice.

Among these genes, Glis2 was chosen and investigated in-depth as a candidate functional effector downstream of polycystin signaling and CDCA. Glis2 transcript and protein expression in cell cultures followed the same polycystin and cilia-dependent changes as in kidney tissue, suggesting Glis2 to potentially be a useful in vitro assay of polycystin function related to cyst formation.

The researchers genetically inactivated Glis2 in mouse kidneys in a conditional fashion and observed a suppression of polycystic kidney disease progression in different mouse models of ADPKD. Additionally, they used antisense oligonucleotides to target Glis2 pharmacologically, which also was effective in suppressing PKD progression. The authors conclude that Glis2 transcript and protein may serve as an in vivo and in vitro biomarker of polycystin function, may be a functional target of CDCA and a potential therapeutic target for treating polycystic kidney disease.

Overall, this study is a very impressive piece of work with an elegant design, elaborate models, rigorous experiments -- a tour-de-force. The new insights, that have been gained, offer substantial value to the broader field of kidney research as the study indeed fills several gaps in the ADPKD field. The article is well written and should be of considerable interest to the readership of Nature Communications.

Major comments:

1) Herein, the candidate Glis2 was exclusively studied and validated in mouse tissue and cultured cells. Demonstration of at least some validating data on Glis2 transcript and protein expression in renal tubular epithelial cells of at least one other species, ideally in human kidney/ADPKD tissue (by e.g. Western Blot, FISH, IHC...), would add important additional value to the study, as parallels and common patterns may be confirmed, thus further underscoring the rationale and translational potential of Glis2 targeting in ADPKD patients.

Minor comments:

1) The evaluation of Glis2-ASO-mediated Glis2 knockdown was conducted using 15 single doses over the course of 13 weeks. It is unclear why this particular regimen was chosen and if titration experiments had been performed to determine the proper in vivo ASO dosage for robust and sustained Glis2 suppression in renal tubular epithelial cells, which ultimately achieved protective effects with regard to PKD progression. Some data/explanation should be provided.

2) Since pharmacological Glis2 targeting was initiated immediately after conditional Pkd1 knockout (i.e. before manifest cyst formation), it remains an open question, whether Glis2 targeting not only “preemptively”, but also at more advanced PKD stages may be of any benefit, as patients are often diagnosed with ADPKD when cysts are already present. This circumstance (and potential ways to address it) may deserve a bit more consideration in the discussion.

3) In the introduction, page 3, line 31: I recommend to include the kidney TRAP study published by Grgic I, Hofmeister AF, Genovese G et al titled “Discovery of new glomerular disease-relevant genes by translational profiling of podocytes in vivo” (Kidney Int. 2014 Dec;86(6):1116-29. doi: 10.1038/ki.2014.204) in the references, since it is the first study to successfully apply kidney cell-specific TRAP in genetic kidney disease models and underscoring its potentials as an unbiased discovery strategy.

4) In the discussion, page 14, line 22-23: please check/clarify sentence.

Reviewer #3 (Remarks to the Author):

In this study by Steve Somlo and co-workers, the authors provide an extensive and thorough investigation to try identifying components of the CDCA pathway that they have previously described.

They first perform an RNAseq approach using inducible animal models of PKD (Pax8TetON Cre) by combining various floxed alleles resulting in inactivation of the Pkd1 gene alone, or in combination with Kif3a to remove cilia and hence the CDCA signal. To identify relevant transcripts early in disease initiation they use a TRAP system, that not only allows the sequencing of transcripts associated with ribosomes (thus being translated) but also allow to use a EGFP protein to select for ribosomes derived from cells in which the Cre has been indeed activated, thus that are destined to form cysts (in Pkd1 mutants) and have been rescued by concomitant inactivation of Kif3 (in double mutants). The investigators perform an extensive analysis of the transcripts that change in the expected direction (i.e. that respond to a CDCA behavior) and in addition they filter the DEGs for genes that are expressed throughout major tubular segments in line with the expected distribution of the CDCA signaling (based on their previous findings of cyst rescue in multiple renal tubular cells by cilia removal and based on the assumption that the same signaling is responsible for rescuing in all segments). Finally, given differences between females and males, they retain in the analysis only DEGs shared in both genders. These several filterings ultimately provide a list of 167 DEGs responding to a CDCA signature profile. Of interest, almost none of these genes has been previously implicated in the multiple attempts to delineate pathways deregulated in PKD, possibly supporting the notion that these genes change early during disease initiation. Among these 167, the investigators identify the transcription factor Glis2 as one of the most significant hits and go on to demonstrate: i) a strict correlation between Glis2 overexpression (in the nucleus) and nuclear translocation upon inactivation of Pkd1 or Pkd2; ii) a robust improvement in disease progression when the Glis2 is conditionally inactivated at the same time as Pkd1 and Pkd2 in the mouse; importantly, inactivation in the WS25 animal model also provides an improvement; iii) that Glis2 is an excellent candidate for therapy as ASOs directed against this transcription factor provide a prominent phenotypic amelioration of the PKD phenotype in adult animal models carrying proximal

tubules-induced inactivation of Pkd1.

The study reported is quite impressive both in the effort of identifying early CDCA signaling events and in the robust validation of the results, that required, among other things, the generation of new antibodies against Glis2 and a new inducible floxed allele of Glis2. Thus, this work is a typical tour-de-force leading to a quite clearcut outcome of the Somlo team.

This reviewer only has a few remarks and suggestions to add.

Major:

- 1) The authors should provide a characterization of the animal model employed for the TRAP analysis, and in particular it would be important to show that indeed the EGFP-expressing cells will end up forming cysts (i.e. what is the percentage of cystic cells that are EGFP positive, and what is the percentage of EGFP-positive cells that form cysts? In other words: is there any cell that will not develop cysts? How long would it take to form cysts and what is the variability?)
- 2) It is interesting to note that despite the fact that Glis2 is a transcription factor, no effort has been placed in trying to validate its activity in the multiple models described. Without going to ChIP-Seq analysis with their newly characterized anti-Glis2 antibody, the authors could simply test whether among the DEGs that respond to a CDCA signature-like behavior are present known or presumed targets of Glis2. How many of the 167 DEGs are direct targets of Glis2? Or, maybe, specifically for this type of analysis the investigators could go back and analyze the differentially expressed genes in a slightly less stringent manner to determine whether they can identify a Glis2 signature, so to speak. It would also be particularly important and interesting to determine whether Glis2 and/ or some of its targets (i.e. a Glis2 "signature") can be identified in the multiple different datasets that have been published from mice and human tissues by multiple labs. Interestingly, while Glis2 loss of function was associated with nephronophthisis, its gain of function has been associated with proliferation and tumorigenesis, i.e. Glis2 has been classified as an oncogene. Its upregulation in PKD might therefore be linked to increased proliferation and it would be interesting to check this aspect by testing for expression levels of direct targets in the multiple models provided (particularly interesting the Glis2 KO in the Pkd background)
- 3) In Figure 2, it would be ideal to see a proper loading control for the multiple western blots shown, rather than the ponceau which, in multiple instances, only shows a blurry smear.
- 4) The investigators propose that Glis2 is a quite direct and proximal marker of Pkd1 or Pkd2 inactivation, as the nuclear translocation seems to be quite robustly and rapidly occurring after inactivation of the genes. How would this occur? Is Glis2 directly binding the Polycystins or what could be an alternative explanation?

Minor:

Figure 3m is missing

In sum, this is an extensive study which, as the authors state, fills several gaps in the ADPKD field. And, therefore, it certainly deserves publication in a prominent journal.

Reviewer #4 (Remarks to the Author):

Zhang and colleagues present an experimental study on the role of Glis2 in cilia dependent cyst activation in PKD. The study consists of two main parts. First, the authors use TRAP to identify novel differentially regulated genes between mice with cyst formation +/- cilia. They use three genotypes: Pkd1KO, Pkd1KO+ciliaKO 23 and Pkd1 heterozygous mice. The chosen strategy is elegant, comprehensive and state-of-the art. Results of the unbiased translomic approach are made available by the authors, which will be very helpful for the PKD field to screen for specific pathways. Among the identified DEGs the authors selected Glis2 as an attractive candidate gene. In the second part of the paper, the authors focus on Glis2 function in vitro and in vivo. They very conclusively demonstrate the involvement of Glis2 as an effector of PKD-dependent cyst formation. The data is very clean and the line of evidence is convincing. The experiments are performed and presented on a very high level. The amount of data is impressive and the paper is nicely written. There are only few points I would like to mention:

1. Reference 52 by the Attanasio group has already shown that loss of Glis2 suppresses cyst growth in the Kif3a ko mouse. This previous publication does not make the current results less important, but given the parallel observations, Reference 52 should be cited and discussed accordingly with respect to the known effects of Glis2 on cyst formation/growth.
2. The authors identified a possible concern about potential senescence-inducing effects of knocking down Glis2 (based on Ref 52). To exclude increased accumulation and development of cellular senescence they stained for SA-beta-Galactosidase activity. SA-b-Gal is one of many non-specific markers of cellular senescence. It simply detects a higher lysosomal galactosidase activity, which is also seen in many non-senescent conditions. Therefore, there are clear recommendations that SA-b-Gal has to be combined with other markers of senescence (e.g. Cdkn2a and Cdkn1a on transcriptional level, reduced lamin b1 staining, enhanced DNA damage markers, absence of proliferation markers, etc.). Please provide additional markers or alternatively refrain calling the observed state cellular senescence if SA-b-Gal is used alone (increased lysosomal activity could be an alternative terminology).
3. Is anything known about the expression pattern of Glis2 in human PKD? E.g. from scRNAseq (Nat Commun. 2022; 13: 6497.)? This would be interesting from a translational standpoint.
4. A challenging aspect of the study is the impact of sex (in particular difference between male and female mouse TRAP data). It is generally reasonable and important to elucidate sex differences in PKD disease models. However, in the current study the added value of female mice is not really clear. In the human PKD field sex differences are not as relevant. While it might be an advantage to use female mice for a less stringent PKD model the authors should describe the background and advantages in more detail for researchers less familiar with this specific model.
5. It was not the focus of the study to elucidate the mechanistic downstream effects of Glis2 (how is the effector working?). However, it would be desirable to obtain at least some speculative explanation on the effector pathway of Glis2, which is responsible for the observed differences.

Response to REVIEWER COMMENTS

Reviewer #1 (Remarks to the Author):

In the manuscript “Glis2 is an early effector of polycystin signaling and a target for therapy in polycystic kidney disease”, Zhang and collaborators investigate the phenomenon that cyst formation following the inactivation of polycystin-1 or polycystin-2 in mice kidney tubules requires the presence of structurally and functionally intact primary cilia (a phenomenon defined by the authors as cilia dependent cyst activation or CDCA). To identify potential molecular mechanisms that could explain this observation, the authors compared the transcriptome profiles of kidney tubular cells from transgenic mice with kidney-specific genetic inactivation of Pkd1, Pkd1;Kif3a double knockouts (no cilia), and Pkd1 heterozygous mice as a control, to prioritize differentially translated genes for further studies. Among the differentially translated genes, the authors chose to focus on the gene Glis2 as a potential contributor to the CDCA, and generated several models of Pkd1- and Pkd2- Glis2 double knockouts in the presence and the absence of cilia. The authors show that genetic and transcriptional (by using an antisense oligonucleotide) inactivation of Glis2 resulted in slower cyst growth and improved kidney function in several mouse models of PKD. The authors conclude that the expression of Glis2 is a biomarker of polycystin function, a functional target of CDCA, and a potential target to treat PKD. Numerous critical methodological, conceptual, and technical shortcomings limit the scientific soundness and relevance of this manuscript.

The first is the complete absence of mechanistic insight into how glis2 is upregulated after the genetic inactivation of PKD1/2 and how the inactivation of Glis2 in Pkd1 and Pkd2 results in a milder cystic phenotype.

The reviewer is correct to say that this manuscript does not describe the mechanism by which Glis2 is upregulated following the inactivation of *Pkd1* or *Pkd2* nor does it directly address the mechanisms by which inactivation of Glis2 suppresses cyst growth. What the manuscript does do, however, is make the novel discovery of the relationship between polycystin inactivation and Glis2 up-regulation and demonstrates the functional relevance of Glis2 up-regulation to the progression of cyst formation in PKD models. These findings are in and of themselves substantive contributions and the data supporting these conclusions are clearly presented in the manuscript. The mechanistic questions the reviewer refers to are appropriate subjects for future investigations that we and others interested in this area will undertake.

The definition of cilia-dependent cyst activation (CDCA) itself is an abstract semantic construct that designates experimental observations that have not been mechanistically explained. In the absence of such mechanistic insights, it needs to be clarified why and how the loss of functional cilia invariably results in a cystic phenotype except in the absence of PC1 and PC2. Thus, the fact that cyst formation following the inactivation of PC1 and PC2 requires the presence of structurally and functionally intact primary cilia does not imply that a CDCA pathway actually exists.

The reviewer unfortunately conflates cyst formation due to loss of polycystins with cyst formation due to loss of cilia. The impetus for the study that defined CDCA was the observation clearly outlined in reference 19 [Ma, M., Tian, X., Igarashi, P., Pazour, G.J. & Somlo, S. Loss of cilia suppresses cyst growth in genetic models of autosomal dominant polycystic kidney disease. *Nat Genet* **45**, 1004-12 (2013)] that inactivating cilia resulted in a much more slowly progressive and indolent form of cyst formation compared to inactivation of polycystins. Therefore, the fact that inactivating cilia “invariably causes cyst formation” does not in any way imply that the cyst formation due to inactivation of cilia and that of polycystins are comparable. Only loss of polycystins results in severe aggressive PKD.

Similarly, Glis2 overexpression in the absence of Pkd does not imply that Glis2 is part of a putative ciliary PKD signaling, especially considering that cilia are not required for Glis2 function (PMID: 27181777). The overexpression of Glis2 could simply be an indirect, secondary effect of Pkd inactivation, and the partial rescue of the cystic Pkd KO phenotype by Glis2 RNA interference could represent a bystander effect of the perturbation of other unrelated signaling pathways. None of these possibilities have been experimentally investigated.

We agree with the reviewer that “Glis2 could simply be an indirect, secondary effect of Pkd inactivation”—this is what our data show. We took pains to show that Glis2 is not expressed in cilia, which supports the likelihood that the effect of polycystin inactivation on Glis2 expression is an indirect effect.’

The reviewer's statement that "the partial rescue of the cystic Pkd KO phenotype by Glis2 RNA interference could represent a bystander effect of the perturbation of other unrelated signaling pathways" only makes sense if one ignores all the data in Figures 2 and 3 which extensively document genotype dependence of the Glis2 effects with no opportunity for "off target" ASO effects. That is why we did those studies first and presented them first.

The manuscript is conceptually based on the unproven assumption that the function of the polycystins in the cilia drives the cystic phenotype, excluding the role of polycystins functions in other cell compartments.

There is ample literature over the past two decades to support the importance of cilia in polycystin function and cyst formation. At no point in the manuscript do we exclude potential roles for polycystins in other compartments in the cell. We simply chose to investigate the role of polycystins in one well established and scientifically accepted compartment, the primary cilium, but we made no effort to exclude alternative hypotheses. Our work will add to the literature supporting the importance of cilia in polycystin function but in no way detracts from investigators who feel other compartments of cell may be more important.

It is quite surprising that the upregulation of Glis2 and Pkd2 has not been reported in the plethora of other gene expression studies performed in PKD animal and human studies. This point has not been mentioned or discussed by the authors, as it has not been tested whether Glis2 expression also increased in human PKD cysts or human organoids.

Most of the existing data are from bulk kidney transcriptomics (mouse) and human studies invariably examine late-stage tissues. Bulk kidney or end stage kidney is a mixed bag with significant contribution from secondary process such as inflammation and fibrosis, often occurring over long-time courses. In such studies, it becomes difficult to focus in on what is important due to that large number of changes observed. This is why we chose to do TRAP RNA-seq at an early (pre-cystic) time point—to better focus in on early changes directly resulting from loss of PC1. Specifically, to answer the reviewer comment: *Glis2* upregulation has been observed in other data sets related to ADPKD, including human studies. It just has not previously been recognized for its potential importance. Below are some instances showing or supporting the occurrence of *Glis2* up-regulation in the context of ADPKD:

Table S1. List of genes in mutant-signature.

Column ID	qvalue(p-value(P * 14 vs. N * 14))	Fold-Change(P * 14 vs. N * 14)	qvalue(p-value(P * 12 vs. N * 12))	Fold-Change(P * 12 vs. N * 12)
Glis2	1.80E-04	1.43	3.26E-02	1.28
Pkd2	3.26E-05	1.38	4.07E-03	1.29
Impa1	1.08E-02	1.25	1.37E-01	1.20
Necap1	2.30E-02	1.23	1.62E-01	1.20
Lad1	5.32E-03	-1.35	1.30E-01	-1.26

In 2012, Menzes et al performed microarray on mouse models of *Pkd1* KO (PMID: 23209428). They presented supplementary data in which *Glis2* showed upregulation. Table at left shows some of the filtered and highlighted data for *Glis2* as well as some additional genes in our TRAP from their supplementary table S1.

In 2017, Malas et al (PMID: 28148532) performed microarray on mouse *Pkd1* KO models in a meta-analysis with other published expression profiles. They reported: "For instance, *Glis2* an important gene in kidney function (22, 26) is part of the PKD signature and was consistently upregulated in all qPCR measured time points of the iKsp-*Pkd1* del."

RNAseq from *Pkd2* KO kidney: 3 biological replicates

In 2021, data from our lab using bulk RNAseq from kidneys of *Pkd2* mutant mice with and without cilia knockout (PMID: 33046531) found that *Glis2* (as well as other genes in our TRAP dataset) were differentially expressed in the *Pkd2* knockouts compared to both noncystic and *Pkd2*/cilia double knockouts. The figure at left highlights some of that data.

This same data is part of the responses to Reviewers 2 and 4 as well:

In 2009, Song et al. (PMID: 19346236) performed microarray on human ADPKD cyst samples. They observed significantly increased expression of *GLIS2* in more cystic disease (Figure at left). In addition, several other genes in the TRAP gene list from manuscript Figure 1d showed relative changes in expression that correlated with cyst severity in the human studies (Figure at left). Specifically, in addition to *GLIS2*, *PKD2*, *PTPDC1*, *TSPAN5* also showed upregulation in the human studies and *LAD1* showed downregulation.

In 2022, Muto et al. (PMID: 36310237) performed snRNAseq from 5 healthy kidneys and 8 ADPKD patient kidneys. Authors also reported *GPRC5A* as a marker of ADPKD kidneys. They reported that the cyst lining cells in their sample were in the connecting tubule and principal cell cluster (CNT_PC). We re-analyzed their snRNAseq dataset (GSE185948) to examine *GLIS2* expression which was not reported in per-cell gene expression analysis in the original paper. We used the authors' processing code on GitHub (https://github.com/TheHumphreysLab/Multimodal_analysis_ADPKD) to reproduce the UMAP plots as published (not shown). Table 1 at left shows the number of cells with expression of *GLIS2* and *GPRC5A* in the CNT_PC cluster in control and PKD samples. We used an adjusted count threshold of >1 for gene expression detection to count cells, and we used *GPRC5A* as a positive control for per-cell

expression analysis. On average, there was a ~30-fold increase in the cells expressing *GPRC5A* in the PKD samples compared to control; there was a two-fold increase in cells expressing *GLIS2* in the PKD samples. Given the small absolute number of cells expressing *GLIS2*, we were not able to do determine per-cell expression for *GLIS2* as was done for *GPRC5A* in the paper. We therefore used a pseudobulk strategy with the DEGseq2 package to assess *GLIS2* expression in the PKD and control groups. Pseudobulk analysis refers to the use of single-cell expression data to compute average gene expression in a cluster of cells and thereby simulate bulk gene expression (PMID: 34321199, PMID: 36550119,

Table 1:

Cell numbers in segment

Patient type	CNT_PC		Patient type	CPM Normalized Gene Expression Values	
	GLIS2	GPRC5A		GLIS2	GPRC5A
control1	1	13	control1	50.4	63.4
control2	4	6	control2	50.3	55.9
control3	1	3	control3	57.5	54.8
control4	10	18	control4	56.3	57.4
control5	0	2	control5	56.2	52.2
PKD1	7	420	PKD1	64.4	101.5
PKD2	6	107	PKD2	56.7	74.4
PKD3	15	392	PKD3	68.7	97.4
PKD4	2	249	PKD4	62.7	107.9
PKD5	3	53	PKD5	66.7	84.6
PKD6	17	236	PKD6	53.8	67.2
PKD7	1	4	PKD7	55.6	78.8
PKD8	6	638	PKD8	60.6	107.4
control_average	3.2	8.4	control_average	54.1	56.7
PKD_average	7.1	262.4	PKD_average	61.2	89.9

PMID: 35361816, PMID: 35864314, PMID: 37400500); for context, our TRAP-RNASeq is a cell type specific bulk RNASeq approach. Table 2 shows the counts per million (CPM) normalized pseudobulk expression levels of *GLIS2* and *GPRC5A*. Using pseudobulk analysis, *GPRC5A* shows significant increase in PKD samples compared to control, in line with the published data of Muto et al. *GLIS2* average CPM is 61.2 in PKD vs 54.1 in controls, but these averages are based on far fewer cells so pAdj is not significant. Still, there is higher average expression in a larger number of cells for *GLIS2* in the PKD samples than in the controls. All we can say is that *GLIS2* shows a trend towards increased expression in the PKD cell cluster that includes cells of cyst origin. These results, while not conclusive, are at least "permissive" for translation of our findings in mouse models to human samples.

The use of mouse models with pan-tubular inactivation of PKD is disputable, considering that in human PKD cysts arise in the distal nephron. Additionally, using such different transgenic mouse models generates variability and confusion.

The use of the pan-tubular knockout for studying orthologous models of Pkd1 into his well-established for over a decade in peer reviewed literature. It is supported by the resources provided for the PKD Renal

Fig. 2. Glomerulus (1), proximal convoluted tubule (2) and the loop of Henle (3). Cysts are situated on the proximal convoluted tubule and loop of Henle (arrow-heads). (Magnification $\times 12.5$, enlargements $\times 51.25$ and $\times 200$.)

Fig. 3. Glomerulus (1), proximal convoluted tubule (2). A single cyst is situated along the path of an otherwise normal proximal tubule. In A and B, the tubule distal to the cyst has been disrupted during the maceration. In C, the proximal and distal part are present.

Research Consortium (PKD-RRC; Pkd1 Flox; Pax8-rtTA; tetO-7-Cre; Nephron specific deletion of Pkd1 after Doxycycline induction – PKD Research Resource Consortium (pkd-rrc.org)) developed by the NIH. As discussed in the manuscript, the use of multiple animal models involving both Pkd1 and Pkd2 and early and late-stage inactivation is actually critically important validation step to ensure that the results are consistent rather than being model dependent and such an approach is becoming the standard in literature with varied combinations of models based on orthologous ADPKD genes (e.g. PMID: 28205547; PMID: 23892607; PMID: 34635846).

There is no question that all segments of the nephron give rise to cysts in human ADPKD. We offer the pictures at left from a classic paper from over four decades ago demonstrating cysts in multiple segments from human kidneys (Kidney International, Vol. 13 (1978), pp. 519—525). This data has been substantiated over the ensuing decades and supported by numerous animal model studies as well. It is likely true that distal nephron cysts grow faster and are larger, but this does not alter the fact that all segments of the nephron give rise to cysts and the relative contributions to disease progression from different segments is not experimentally or clinically defined.

The manuscript is burdened with errors, uninterpretable words, expressions, and sentences and would benefit from extensive editing.

We have revised the manuscript to improve the presentation.

More specific comments are listed below:

Page 2, lines 17/18: The statement “*Glis2* transcript and protein is an *in vivo* and *in vitro* biomarker of polycystin function” is not supported by experimental data. Based on the data, *Glis2* change of expression is associated with PC inactivation, not function.

The statement as written is correct and supported by experimental data presented. We present far more data for *Glis2* genotype dependence than just “PC inactivation.” Inactivation of polycystin, which is a surrogate for polycystin loss-of-function, does result in increased nuclear *Glis2*. However, in the presence of persistent inactivation of polycystin, increased *Glis2* expression reverts to normal when cilia are also inactivated. This is a response to change in the CDCA signaling that also abrogates cyst progression and therefore abrogates the functional effects of polycystin loss. So *Glis2* changes with functional changes related to polycystins, not just with PC inactivation. Furthermore, we show that re-expression of polycystin also reverts *Glis2* expression to normal levels. Therefore, PC1 reactivation also influences *Glis2* expression. We feel that these and other experiments detailed in the paper warrant the conclusion that change of expression of *Glis2* is associated with PC function, not just inactivation.

Page 3, line 19: The effect of *Tulp3* inactivation on cyst growth is debated, given contrasting results from other publications (PMID: 30799239).

Actually, the “contrasting results” appear in both papers—the citation that the reviewer provides as well as the paper that we cite (PMID: 30799240). The “contrast” is that *Tulp3* inactivation does not suppress *Pkd1* cysts in early (embryonic/perinatal) inactivation models, only in the adult model. The citation the reviewer offers does not address the adult models at all, whereas the citation we provide addresses both and notes the different outcomes. We have independently confirmed the cyst suppressive effects of adult inactivation of *Tulp3* in *Pkd1* models. There is no doubt (or debate) that *Tulp3* inactivation results in very impressive rescue of cyst formation in adult conditional inactivation models of *Pkd1*.

Page 4, line 5: “Polycystin-dependent cyst formation” is incorrect; is it meant cilia-dependent cyst formation?

It is meant as written. When loss of polycystin function drives cyst formation, *Glis2* expression is elevated. When loss of polycystin function does not drive cyst formation (i.e., with cilia knockout, with *Tulp2* knockout), *Glis2* is not elevated. Therefore, *Glis2* expression is a surrogate for polycystin dependent cyst formation—the relevant subject of this statement is “cyst formation.” Of note, “polycystin” in this instance is inclusive of PC1 and PC2.

Page 4, lines 8-9: It is not clear what the novelty is in using TRAP as an exploratory analysis.

Here is the sentence the reviewer refers to as it appears in the manuscript: “In aggregate, we have developed a discovery platform for transcriptional targets of polycystin function and discovered *Glis2* as a novel and potentially tractable therapeutic target for treatment of ADPKD.” The word “novel” is modifying “*Glis2*,” not “TRAP,” which does not appear in the sentence.

Page 4, line 18: The use of the pan-nephron Pax8rtTA;TetOCre model is debatable, given that only the distal nephron develops cysts in ADPKD.

This has been addressed above when the reviewer made the same comments. This model is well accepted in the peer reviewed literature and the reviewer is incorrect in stating that “only the distal nephron develops cysts in ADPKD.”

Page 4, line 25: The sentence “To 24 limit discovery to cell-autonomous transcriptional changes following loss of polycystin-1” is unjustified: TRAP only measures cell-autonomous changes.

The complete sentence the reviewer refers to is: “To limit discovery to cell autonomous transcriptional changes following loss of polycystin-1 and avoid confounding by “outside-in” signaling from inflammatory and other responses that occur as cyst formation progresses, we performed TRAP RNASeq at a time point at which polycystin-1 and cilia had disappeared¹⁹ but cysts had not yet begun to form.” The reviewer missed the critical concept that events outside of the cyst cell such as immune infiltration can, through cell-extrinsic signaling, alter the properties of tubule cells in such a way that their transcriptional profiles will be affected by these extrinsic signals and therefore not be strictly cell autonomous. Therefore, the critical focus in our approach was not the use of TRAP to capture cell autonomous events but rather to use the early model before extensive secondary changes such as inflammation occur to limit the “outside-in” effects that could result in transcriptional changes that are not fundamentally cell autonomous.

Having said that, we have updated the wording somewhat to make things clearer so that readers will not be confused the way that the reviewer was. It now reads: “We sought to identify cell autonomous transcriptional changes resulting from polycystin inactivation in tubule cells. Since cyst formation causes inflammatory and other tissue level responses that can result in “outside-in” signaling that is not cell-autonomous, we sought to limit the latter effects by doing TRAP RNASeq at a time point at which polycystin-1 and cilia had disappeared¹⁹ but cysts had not yet begun to form.”

Page 4, line 34: It is unclear what the meaning/relevance of “cilium organization” differentially translated genes is.

The following sentence occurred on page 5, line 34: “The 167 gene overlap group showed enrichment primarily of metabolic pathways although “cilium organization” also appeared (Supplementary Figure 6; Supplementary Table 3).” In this sentence and the data presented in the supplementary figure and table referenced, “cilium organization” refers to a phrase output by the pathway analysis using Metascape provided that is in Supplementary Figure 6. This is akin to “GO terms,” only from Metascape. The genes included in this Metascape output term are provided in Supplementary Table 3. We believe that readers who look at the Figure and Table will understand that this is reference to a Metascape output term.

Page 4, line 35: The missed identification of pathways previously implicated in ADPKD pathogenesis may suggest the opportunity for new discoveries but also underlines the lack of validation from other studies.

It is hard to understand how science can advance if one only accepts findings that have been previously validated. Our studies are well controlled, and we validated the findings related to *Glis2* in several relevant biological systems. We also provide some biological validation of other transcriptional changes as well (e.g., Figure 1g). As noted above, many of the transcriptional changes found in our TRAP studies were present in previously published studies in humans and mice—only their potential significance may have been missed.

Page 5, line 37 - page 6, lines 1-2: *The inactivation of cilia in all mouse models causes cysts, does not suppress them. Ablation of cilia is only reported to suppress cyst growth in the concomitance of PKD inactivation (citation 19).*

We have clarified the text as suggested by the reviewer. The sentence now reads: "All kidney tubule segments and tubule epithelial cell types, with the likely exception of intercalated cells which lack cilia³⁹, have the capacity to form cysts *following polycystin inactivation* that can be suppressed by inactivation of cilia¹⁹." (Italics to highlight changes)

Page 5, lines 2-4: *It is unclear why transcripts expressed along the entire tubule would be expected of candidate transcriptional changes related to CDCA. Wouldn't it be proper to focus only on genes expressed in the distal nephron, where cysts develop in humans?*

We have addressed this concern in the earlier comments. Humans develop cysts in all segments of the nephron. Distal nephron cysts likely grow faster and larger, but that does not change the fact that all segments can give rise to cysts. Furthermore, the CDCA mechanism (i.e., cilia suppression of cyst growth after PC inactivation) has been demonstrated to be active in all segments of the nephron for both *Pkd1* and *Pkd2* and it has also been shown in bile ducts in the liver. Therefore, it is appropriate to expect that the most central components of the CDCA pathway response are present in all nephron segments. There may be elements that are more specific or active in certain segments to account for the differential cyst growth responses in nephron segments, but our focus is on the most common elements of CDCA. The most parsimonious hypothesis for identifying the most common transcriptional responses to CDCA is that they are shared in all segments that can give rise to cysts in human ADPKD and that can be suppressed by cilia inactivation in animal models.

Page 6, lines 26/27: *why were most genes not involved in cystic kidney diseases tested by RT-PCR?*

How does the reviewer know that these genes are not involved in cystic kidney diseases? We keep an open mind on whether or not these genes may be involved based on our in vivo discovery experiments. We used qRT-PCR of these genes as a secondary validation method in primary cells. The in vivo TRAP RNASeq data is "gold standard" data since it is in vivo derived and robust to multiple independent biological samples. The genes that we tested were previously not known to be associated with transcriptional responses following PC loss specifically associated with the propensity for cyst formation in the presence of intact cilia. Their identification as transcriptionally altered in cystic disease is a novel discovery in our work. The reproducible changes by qRT-PCR in primary cells in vitro validates these cells as in vitro models beyond just *Glis2* expression changes while also providing an in vitro supporting model for the gold standard in vivo data for the expression changes of these additional genes.

Page 7, line 33: *"Developing kidney" refers to the kidney during embryonic development. The sentence, instead, refers to Pkd2 KO in which Pkd2 is re-expressed (as per Figure 2).*

"Developing kidney" refers to kidneys undergoing active nephrogenesis which continues for ~2-3 weeks postnatally in the mouse.

Page 8, line 10: *Does the expression validated Glis2 targets, such as Wnt4 and Snai1, also vary accordingly?*

We did not evaluate putative targets of *Glis2* transcription factor activity in this study. We limit our conclusions to genotype-dependent changes in nuclear *Glis2* expression agnostic to its downstream activity. Such questions will be interesting to investigate in the future based on our novel discovery of the altered expression of *Glis2* and its direct relevance to polycystic kidney disease in orthologous gene models of ADPKD.

Page 8, lines 21/22: *It should be clarified that the cyst-forming potential refers to the specific model used here, i.e., Pkd KO cells, and not to the general cyst-forming potential attributed to cilia in the context of PKD inactivation.*

The study and the manuscript addresses ADPKD models of cyst formation in the context of orthologous genes related to that human disease. It is a tautology that this refers to *Pkd* KO and we trust the readers to recognize this is what the topic of the sentence is based on the extensive presentation of information leading up to this sentence. That said, we have modified the sentence to read: "*Glis2* and *PC2* protein expression in *Pkd1;Tulp3* double knockout cells remained unchanged from non-knockout controls further

supporting the strict correlation of Glis2 levels in vitro with *polycystin-dependent* cyst forming potential in vivo (Figure 2k).” (Italics to highlight the change)

We are unsure what the reviewer means by “*the general cyst-forming potential attributed to cilia in the context of PKD inactivation.*”

Page 9, lines 2 and 6: The working hypothesis is that a putative CDCA controls cyst growth in Pkd-null kidneys, not that a “polycystin function dependent cyst forming potential” exists: PCs loss of function causes cysts.

“Polycystin function” addresses both its function when present and the absence of its function when absent as well as its reduced function when hypomorphic.

We have revised the sentence in line 2 to read: “*Glis2* transcript and protein expression changes in primary cells are an in vitro surrogate indicator of in vivo cyst forming potential resulting from polycystin dysfunction.”

We left line 6 unchanged: “...we used a series of mouse allele combinations to determine whether inactivation of *Glis2* affected polycystin-dependent cyst progression.” “Polycystin-dependent” is dependence on polycystin.

Page 9, line 28: Has it been tested if embryonic deletion of Glis/f mice reproduces the phenotype of the constitutive knockouts?

Yes, this is presented in Supplementary Figures 15 and 18. The embryonic deletion allele is labeled *Glis2*^{Δex3} to distinguish it from the original null allele.

Page 10, line 23: “Glis2^{-/-}, Pkd2WS25^{-/-}, and Glis2^{-/-}; Pkd2WS25^{-/-} mice are the same genotype.

We corrected this typo.

Page 12, line 29: The association between the hedgehog pathway and polycystin signaling is obscure.

We have removed this comparison.

Page 13, line 21: It is not clear why prioritizing differentially expressed genes in all nephron segments instead of focusing on the distal nephron only.

This has been addressed on multiple occasions earlier in this response.

Page 13, line 29: “This transcriptional signature can be applied...”. This needs to be experimentally proven.

This is a discussion. The transcriptional changes in the current study are well validated internally in vivo and to some extent in vitro. We are proposing that transcription signature can be applied to other systems. It is within the scope of the data presented in this manuscript that such a proposal can be made in the context of discussion.

Page 13, line 37: “We found”. Based on the cited reference, this should read, “We confirmed”.

We chose our words carefully here. Previous literature showed that *Glis2* mRNA expression by RT-PCR (Fig. 6D of the cited paper, PMID: 21816948) was elevated in the early postnatal period, but it did not examine protein levels. As most investigators know, there can be a disconnect between transcript levels and protein levels so one should not assume the latter based on the former. Our data, which examines protein levels directly, is a “finding,” not a confirmation. That said, we have revised the statement to make this clearer: “We found that *Glis2* protein expression is highest in the first two postnatal weeks in the mouse kidney, confirming that the previously reported increased transcript levels during development are correlated with protein levels⁶⁵.”

Page 14, lines 1-2: The parallel between postnatal kidney expression of Glis2 and polycystin-1 expression is unclear.

There are higher levels of *Glis2* and PC1 expression in the first two weeks postnatally and then they both decrease in expression level. Around two weeks postnatally is also when the developmental switch from rapid to slower cyst growth occurs. We are just noting the co-occurrences of these events based on our data combined with the literature cited.

Page 14, line 6: the statement that there are no reported extrarenal defects in Glis2 knockout mice is false. See PMID: 30523147 and PMID: 34705506. It is also not unlikely that more subtle extrarenal manifestations have been described in patients with mutations in NPHP7, given these subjects' early onset and mortality.

We have amended the statement regarding mice and included the references provided by the reviewer. The statement now reads: “*Glis2*^{-/-} mice develop progressive nephronophthisis-like atrophic tubulointerstitial fibrotic kidney disease associated with tubule cell senescence, but with relatively few reported extra-renal defects^{32,48,52,62,67}.”

We have edited the following sentence incorporate the reviewers perspective: “Similarly, patients with *NPHP7* develop early onset end stage kidney disease, and although it is possible that subtle extrarenal manifestations also occur, these have not been reported to date^{32,33}.”

Page 14, lines 9-11: It would be interesting to test whether mice with Glis2 inactivation after development show signs of disease.

Supplementary Figures 15 and 18 address this in part. No NPHP-like phenotype, no increased senescence after ~3 months of inactivation.

Page 15, line 24: The relevance of establishing Glis2 as a therapeutic target in preclinical models is unclear.

We disagree with the reviewer based on the data presented and the discussion of that data provided in the manuscript.

Page 15, lines 29-30: See comments to Page 14, line 6.

We have modified this sentence to read: “Furthermore, a drug with specificity for *Glis2* may be tolerated given the limited known extra-renal effects in germline knockout mice and paucity of extrarenal manifestation reported in patients with *NPHP7*. This is further supported by the apparent difference in kidney phenotypes between germline inactivation which results in early onset NPHP in mice and adult inactivation which has no effect on the kidney for at least 3 months.”

Page 15, line 35: The relevance of using a model of PKD with proximal tubule inactivation is unclear.

This has been addressed in the above responses.

Page 16, lines 2-3: Although there may have been some variability in using ASO after the formation of cysts, this experiment should have been performed.

This will be a good future study as is clearly stated in the following sentence to the one the reviewer cites. Here is the relevant part of the discussion in its entirety: “Finally, we began treatment with ASO soon after inactivation of *Pkd1* before cyst had already formed. This choice eliminated potential variation in drug availability in more advanced cysts that may not receive glomerular filtrate. This was beneficial in a proof-of-concept studies, but future studies should determine if inhibition of *Glis2* at later stages of ADPKD can nonetheless slow disease progression.”

Page 16, lines 6-13: The statement that Glis2 expression is an indicator of polycystin function is an overstatement not supported by experimental data.

All the data presented in this manuscript point to our conclusion with an abundance of in vivo and in vitro validation. Inactivation of polycystins leads to increased transcript and protein expression for *Glis2* and simultaneous inactivation of cilia abrogates this response both in vitro and in vivo. This is true for *Pkd1* and for *Pkd1* across a spectrum of mouse models and cilia related mutations. Therefore, we respectfully disagree with the reviewer's comment.

Page 16, Figure 1: Why are some genes in panel b marked in red?

We want to draw attention to them based on high FDR and possible functional relevance and because we further highlight them in Figure 1e.

Reviewer #2 (Remarks to the Author):

The reviewed manuscript titled "Glis2 is an early effector of polycystin signaling and a target for therapy in polycystic kidney disease" by Zhang et al propose Glis2 as a novel biomarker in and potential therapeutic target of ADPKD. The researchers used unbiased cell type-specific transcriptional profiling to analyze actively translating mRNA in various mouse models of ADPKD at early time points and identified 167 differentially expressed mRNA that correlated with the CDCA pattern in both male and female mice.

Among these genes, Glis2 was chosen and investigated in-depth as a candidate functional effector downstream of polycystin signaling and CDCA. Glis2 transcript and protein expression in cell cultures followed the same polycystin and cilia-dependent changes as in kidney tissue, suggesting Glis2 to potentially be a useful in vitro assay of polycystin function related to cyst formation.

The researchers genetically inactivated Glis2 in mouse kidneys in a conditional fashion and observed a suppression of polycystic kidney disease progression in different mouse models of ADPKD. Additionally, they used antisense oligonucleotides to target Glis2 pharmacologically, which also was effective in suppressing PKD progression. The authors conclude that Glis2 transcript and protein may serve as an in vivo and in vitro biomarker of polycystin function, may be a functional target of CDCA and a potential therapeutic target for treating polycystic kidney disease.

Overall, this study is a very impressive piece of work with an elegant design, elaborate models, rigorous experiments -- a tour-de-force. The new insights, that have been gained, offer substantial value to the broader field of kidney research as the study indeed fills several gaps in the ADPKD field. The article is well written and should be of considerable interest to the readership of Nature Communications.

We thank the reviewer for their careful evaluation of this manuscript.

Major comments:

Herein, the candidate Glis2 was exclusively studied and validated in mouse tissue and cultured cells. Demonstration of at least some validating data on Glis2 transcript and protein expression in renal tubular epithelial cells of at least one other species, ideally in human kidney/ADPKD tissue (by e.g. Western Blot, FISH, IHC...), would add important additional value to the study, as parallels and common patterns may be confirmed, thus further underscoring the rationale and translational potential of Glis2 targeting in ADPKD patients.

The challenge in validating these studies with human tissues or cells is that the relevant stage material is not accessible from patient samples. Human kidney tissue and cells from ADPKD patients are invariably derived from nephrectomies of patients with end stage kidney disease, which is permeated by many secondary changes including inflammation, fibrosis, infections, calcifications, etc. that markedly change the composition and responses of the cell types in the kidney. One of the critical advantages of the mouse model was the ability to interrogate the early, precystic changes; we assiduously avoided tissues even in the mouse that had undergone extensive secondary effects from the disease. Similarly, human iPSC derive organoids are not known to reproduce our well specified biological system. Specially, most conventional organoids lack vascularization and ultrafiltration which may well be critical to CDCA signaling. Furthermore, most organoids that form cysts with *PKD1* mutations have reverse topology with cilia outside the cyst.

Since the human material is significantly different from the biological material used in this study, in the interest of maintaining rigor, we hesitate in interpreting results as either confirmatory or not. On the one hand, if Glis2 is upregulated in these systems, it does not mean that this is not due to secondary effects such as inflammatory cells invading the tissue (as an example); it cannot be rigorously concluded that this is due to cell autonomous primary effects of *PKD1* inactivation. Similarly, if Glis2 is not upregulated, it does not mean that upregulation was not a feature of the early changes following *PKD1* loss that is now confounded by processes in the late-stage kidneys that are part of altered terminal phenotype after decades of tissue damage. So, without the ability to draw rigorous conclusions from the available human samples, we have opted not to pursue these studies.

That said, we offer two circumstantial examples from existing data sets that at least are at least permissive for the human applicability of our findings. The upregulation of *GLIS2* have been suggested in two human models previously reported:

This same data is part of the responses to Reviewers 1 and 4 as well:

In 2009, Song et al. (PMID: 19346236) performed microarray on human ADPKD cyst samples. They observed significantly increased expression of *GLIS2* in more cystic disease (Figure at left). In addition, several other genes in the TRAP gene list from manuscript Figure 1d showed relative changes in expression that correlated with cyst severity in the human studies (Figure at left). Specifically, in addition to *GLIS2*, *PKD2*, *PTPDC1*, *TSPAN5* also showed upregulation in the human studies and *LAD1* showed downregulation.

In 2022, Muto et al. (PMID: 36310237) performed snRNAseq from 5 healthy kidneys and 8 ADPKD patient kidneys. Authors also reported *GPRC5A* as a marker of ADPKD kidneys. They reported that the cyst lining cells in their sample were in the connecting tubule and principal cell cluster (CNT_PC). We re-analyzed their snRNAseq dataset (GSE185948) to examine *GLIS2* expression which was not reported in per-cell gene expression analysis in the original paper. We used the authors' processing code on GitHub (https://github.com/TheHumphreysLab/Multimodal_analysis_ADPKD) to reproduce the UMAP plots as published (not shown). Table 1 at left shows the number of cells with expression of *GLIS2* and *GPRC5A* in the CNT_PC cluster in control and PKD samples. We used an adjusted count threshold of >1 for gene expression detection to count cells, and we used *GPRC5A* as a positive control for per-cell

Table 1:
Cell numbers in segment

Patient type	CNT_PC	
	GLIS2	GPRC5A
control1	1	13
control2	4	6
control3	1	3
control4	10	18
control5	0	2
PKD1	7	420
PKD2	6	107
PKD3	15	392
PKD4	2	249
PKD5	3	53
PKD6	17	236
PKD7	1	4
PKD8	6	638
control_average	3.2	8.4
PKD_average	7.1	262.4

Table 2:
CPM Normalized Gene Expression Values

Patient type	CNT_PC	
	GLIS2	GPRC5A
control1	50.4	63.4
control2	50.3	55.9
control3	57.5	54.8
control4	56.3	57.4
control5	56.2	52.2
PKD1	64.4	101.5
PKD2	56.7	74.4
PKD3	68.7	97.4
PKD4	62.7	107.9
PKD5	66.7	84.6
PKD6	53.8	67.2
PKD7	55.6	78.8
PKD8	60.6	107.4
control_average	54.1	56.7
PKD_average	61.2	89.9

expression analysis. On average, there was a ~30-fold increase in the cells expressing *GPRC5A* in the PKD samples compared to control; there was a two-fold increase in cells expressing *GLIS2* in the PKD samples. Given the small absolute number of cells expressing *GLIS2*, we were not able to do determine per-cell expression for *GLIS2* as was done for *GPRC5A* in the paper. We therefore used a pseudobulk strategy with the DEGseq2 package to assess *GLIS2* expression in the PKD and control groups. Pseudobulk analysis refers to the use of single-cell expression data to compute average gene expression in a cluster of cells and thereby simulate bulk gene expression (PMID: 34321199, PMID: 36550119,

PMID: 35361816, PMID: 35864314, PMID: 37400500); for context, our TRAP-RNASeq is a cell type specific bulk RNASeq approach. Table 2 shows the counts per million (CPM) normalized pseudobulk expression levels of *GLIS2* and *GPRC5A*. Using pseudobulk analysis, *GPRC5A* shows significant increase in PKD samples compared to control, in line with the published data of Muto et al. *GLIS2* average CPM is 61.2 in PKD vs 54.1 in controls, but these averages are based on far fewer cells so pAdj is not significant. Still, there is higher average expression in a larger number of cells for *GLIS2* in the PKD samples than in the controls. All we can say is that *GLIS2* shows a trend towards increased expression in the PKD cell cluster that includes cells of cyst origin. These results, while not conclusive, are at least “permissive” for translation of our findings in mouse models to human samples.

Minor comments:

1) *The evaluation of Glis2-ASO-mediated Glis2 knockdown was conducted using 15 single doses over the course of 13 weeks. It is unclear why this particular regimen was chosen and if titration experiments had been performed to determine the proper in vivo ASO dosage for robust and sustained Glis2 suppression in renal tubular epithelial cells, which ultimately achieved protective effects with regard to PKD progression. Some data/explanation should be provided.*

The initial assessment of Glis2-ASO dosing is detailed in part in the Methods:

“The *Glis2* ASOs targeting murine *Glis2* were designed, synthesized, and tested in a series of *in vitro* and *in vivo* screens and evaluated for efficacy and tolerability in 8–10-week-old male C57BL/6J mice at a dose of 50 mg/kg per week for 3 weeks by Ionis Pharmaceuticals, Inc. (Carlsbad, CA). ASOs that displayed robust reduction of kidney *Glis2* mRNA without inducing elevations in plasma transaminases and mRNA biomarkers of renal tubular injury (KIM-1 and NGAL) were selected for additional *in vivo* studies in models of ADPKD.”

Since this regimen, which was based on Ionis’ experience with ASO in mouse kidneys, was effective in the above initial evaluation as described in the methods, we opted to adopt the dosage regimen for the preclinical experimental phase as well.

2) *Since pharmacological Glis2 targeting was initiated immediately after conditional Pkd1 knockout (i.e. before manifest cyst formation), it remains an open question, whether Glis2 targeting not only “preemptively”, but also at more advanced PKD stages may be of any benefit, as patients are often diagnosed with ADPKD when cysts are already present. This circumstance (and potential ways to address it) may deserve a bit more consideration in the discussion.*

This very important point was already addressed in our discussion:

“Finally, we began treatment with ASO soon after inactivation of *Pkd1* before cyst had already formed. This choice eliminated potential variation in drug availability in more advanced cysts that may not receive glomerular filtrate. This was beneficial in a proof-of-concept studies, but future studies should determine if inhibition of *Glis2* at later stages of ADPKD can nonetheless slow disease progression.”

3) *In the introduction, page 3, line 31: I recommend to include the kidney TRAP study published by Grgic I, Hofmeister AF, Genovese G et al titled “Discovery of new glomerular disease-relevant genes by translational profiling of podocytes in vivo” (Kidney Int. 2014 Dec;86(6):1116-29. doi: 10.1038/ki.2014.204) in the references, since it is the first study to successfully apply kidney cell-specific TRAP in genetic kidney disease models and underscoring its potentials as an unbiased discovery strategy.*

Thank you for this comment. We have added the reference for the reason cited by the reviewer.

4) *In the discussion, page 14, line 22-23: please check/clarify sentence.*

We have taken the reviewers suggestion and tried to clarify and shorten this entire paragraph.

Reviewer #3 (Remarks to the Author):

In this study by Steve Somlo and co-workers, the authors provide an extensive and thorough investigation to try identifying components of the CDCA pathway that they have previously described.

*They first perform an RNAseq approach using inducible animal models of PKD (Pax8TetON Cre) by combining various floxed alleles resulting in inactivation of the *Pkd1* gene alone, or in combination with *Kif3a* to remove cilia and hence the CDCA signal. To identify relevant transcripts early in disease initiation they use a TRAP system, that not only allows the sequencing of transcripts associated with ribosomes (thus being translated) but also allow to use a EGFP protein to select for ribosomes derived from cells in which the Cre has been indeed activated, thus that are destined to form cysts (in *Pkd1* mutants) and have been rescued by concomitant inactivation of *Kif3* (in double mutants). The investigators perform an extensive analysis of the transcripts that change in the expected direction (i.e. that respond to a CDCA behavior) and in addition they filter the DEGs for genes that are expressed throughout major tubular segments in line with the expected distribution of the CDCA signaling (based on their previous findings of cyst rescue in multiple renal tubular cells by cilia removal and based on the assumption that the same signaling is responsible for rescuing in all segments). Finally, given*

differences between females and males, they retain in the analysis only DEGs shared in both genders. These several filterings ultimately provide a list of 167 DEGs responding to a CDCA signature profile. Of interest, almost none of these genes has been previously implicated in the multiple attempts to delineate pathways deregulated in PKD, possibly supporting the notion that these genes change early during disease initiation. Among these 167, the investigators identify the transcription factor *Glis2* as one of the most significant hits and go on to demonstrate: i) a strict correlation between *Glis2* overexpression (in the nucleus) and nuclear translocation upon inactivation of *Pkd1* or *Pkd2*; ii) a robust improvement in disease progression when the *Glis2* is conditionally inactivated at the same time as *Pkd1* and *Pkd2* in the mouse; importantly, inactivation in the WS25 animal model also provides an improvement; iii) that *Glis2* is an excellent candidate for therapy as ASOs directed against this transcription factor provide a prominent phenotypic amelioration of the PKD phenotype in adult animal models carrying proximal tubules-induced inactivation of *Pkd1*.

The study reported is quite impressive both in the effort of identifying early CDCA signaling events and in the robust validation of the results, that required, among other things, the generation of new antibodies against *Glis2* and a new inducible floxable allele of *Glis2*. Thus, this work is a typical tour-de-force leading to a quite clearcut outcome of the Somlo team.

This reviewer only has a few remarks and suggestions to add.

We thank the reviewer for their careful evaluation of this manuscript.

Major:

1) The authors should provide a characterization of the animal model employed for the TRAP analysis, and in particular it would be important to show that indeed the EGFP-expressing cells will end up forming cysts (i.e. what is the percentage of cystic cells that are EGFP positive, and what is the percentage of EGFP-positive cells that form cysts? In other words: is there any cell that will not develop cysts? How long would it take to form cysts and what is the variability?)

Unfortunately, we did not produce mice that would directly address this request. We used all the TRAP mice by 10 weeks and therefore we did not age TRAP mice out to 16 weeks to let them form cysts that we could interrogate in this manner. Based on allele combination (*Pkd1^{flox}* with *R26^{Rpl10a}*), we reasonably expect that any cells from which ribosomes can be pulled down due to activation of the EGFP-L10a transgene by removal of the flox-STOP will also have inactivation of *Pkd1^{fl}* (and *Kif3a^{fl}* if present). So,

within the precision allowed by the genetic models and the specificity of the ribosomal pulldown, we captured only cells with inactivation of *Pkd1*. We also showed in Supplementary Figure 1c that the efficiency of *Pkd1^{fl}* inactivation in all mice was comparable. By way of indirect evidence that we are capturing only cell types where the *Pax8^{rtTA}* is active, in the figure at right we show evidence of depletion in the TRAP RNA of cell type specific markers from cell types in which *Pax8^{rtTA}* is not active. From left to right, qRT-PCR for endothelial marker *Flt1*, fibroblast marker *Pdgfrb*, podocyte marker *Nphs2*

and macrophage marker *F4/80*. By contrast, *Nhe3* and *Aqp2* from proximal tubule and collecting duct, respectively, showed strong expression in the TRAP RNA (not shown). So given the specificity of the TRAP pulldown for *Pax8^{rtTA}*-expressing cells, the conclusions that the transcriptional changes are related to in vivo *Pkd1* loss are clear and well supported.

Therefore, the question the reviewer is really asking whether *Pkd1* loss leads to cyst formation. For this, admittedly, we relied on some transitivity from data in our previous publication defining CDCA [Ma, M., et al. *Nat Genet* **45**, 1004-12 (2013)]. In that paper (Supplementary Figure 7), we used the mTmG Cre reporter in combination with the *Pax8^{rtTA}* system to define the segments of the proximal tubule (S1, S2 but not S3) and distal nephron (mTAL, DCT, CD) where Cre was expressed, and we showed which segments (PT, CD) gave rise to cysts or had cilia inactivation (Figure 3; Supplementary Figure 7 of the Ma et al. paper). mTAL also form cysts (data not shown). The transitivity is that *Pax^{rtTA}* activity results in

inactivation of *Pkd1* in the same segments as form cysts, so we equate inactivation of *Pkd1* in the current study with the destiny to form cysts. The image at left shows early (12 week) cystic dilation of proximal tubules labeled with megalin (blue) in a *Pax8^{rtTA}* model with mTmG (*Pkd1^{fl/fl};Pax8^{rtTA};TetO^{Cre};mTmG*). EGFP marks cells in which Cre recombinase has been active (i.e., *Pkd1* knockout) and Tomato red cells marks cells with no Cre activity. The EGFP positive tubule segments show dilation indicative of early cyst formation.

Overall, we conclude that based on the aggregate literature on *Pkd1* inactivation in mice from many investigators in addition to our data, there is compelling evidence that if *Pkd1* is inactivated, cysts will form in affected segments. Our TRAP RNASeq data is specific for *Pkd1* inactivation and reasonably anticipates cells with a high propensity for future cyst formation.

2) *It is interesting to note that despite the fact that Glis2 is a transcription factor, no effort has been placed in trying to validate its activity in the multiple models described. Without going to ChIP-Seq analysis with their newly characterized anti-Glis2 antibody, the authors could simply test whether among the DEGs that respond to a CDCA signature-like behavior are present known or presumed targets of Glis2. How many of the 167 DEGs are direct targets of Glis2? Or, maybe, specifically for this type of analysis the investigators could go back and analyze the differentially expressed genes in a slightly less stringent manner to determine whether they can identify a Glis2 signature, so to speak. It would also be particularly important and interesting to determine whether Glis2 and/ or some of its targets (i.e. a Glis2 “signature”) can be identified in the multiple different datasets that have been published from mice and human tissues by multiple labs. Interestingly, while Glis2 loss of function was associated with nephronophthisis, its gain of function has been associated with proliferation and tumorigenesis, i.e. Glis2 has been classified as an oncogene. Its upregulation in PKD might therefore be linked to increased proliferation and it would be interesting to check this aspect by testing for expression levels of direct targets in the multiple models provided (particularly interesting the Glis2 KO in the Pkd background)*

We agree with the reviewer that identifying targets of *Glis2* in the specific context of its upregulation following polycystin inactivation in ADPKD is an important objective and these studies are ongoing. As the reviewer notes, however, these are beyond the scope of the current report. The reviewer suggests that we take existing data sets for *Glis2* targets from other studies and aggregate these with our data to glean potential downstream insights. We chose not to use such an approach because we do not have confidence that the results of such an in-silico analysis from disparate biological systems would be sufficiently reliable for us to include in this manuscript.

Among the datasets available that the reviewer alludes to are two microarray expression studies on *Glis2* null mouse kidneys (PMID: 17618285, 18227149) and ChIP studies from ATCC cancer cell lines (PMID: 32483180) and from immortalized mouse stellate cell line with *Glis2* knockout (PMID: 36397300). None of these bear biological relevance or tissue cell type specificity to our studies. To highlight the differences that limit interpretation of any of these proposed comparisons, *Glis2* null kidneys (arguably the closest available data to our models) show senescence whereas adult inactivation as used in our TRAP studies does not. *Glis2* targets relevant to ADPKD are, unsurprisingly, likely to be biological context dependent and the available data does not offer any biologically relevant context in this regard.

Our study was predicated on finding a constrained data set with high confidence—the TRAP gene set we provide is very well supported and will identify to strongest transcriptional signals, but it is likely not comprehensive (high positive predictive value, but uncertain negative predictive value for putative transcriptional changes). We have added text to make this clearer for the reader: “We chose to focus on DEG shared by male and female mice as a more stringent selection criterion to provide the highest confidence for definition of the ‘CDCA pattern’ transcriptional changes. This does not, however, preclude potential functional importance for other genes, e.g., those showing significant ‘CDCA pattern’ change only in male mice.”

That said, we do not wish to “analyze the differentially expressed genes in a slightly less stringent manner” as suggested by the reviewer as this is counter to our experimental design and intent. Integration

of the available biologically unrelated data sets with loosened stringency for our TRAP data may find potential overlap patterns. We have no confidence in such an analysis—e.g., looking for a proliferation connection with *Glis2* runs the risk of confirmation bias and is contrary to our intent in this study. Our intent in the current study was to do an unbiased analysis and we would like to stick with that. That said, all of our data will be available in the public domain so investigators who wish to do these types of studies with existing or future *Glis2* datasets will be able to do so.

3) In Figure 2, it would be ideal to see a proper loading control for the multiple western blots shown, rather than the ponceau which, in multiple instances, only shows a blurry smear.

All blots with total cell lysates have Hsp90 loading controls in addition to the Ponceau staining. All nuclear fractions have either lamin A/C or lamin B1 nuclear protein loading controls in addition to Ponceau staining. Only the cytosolic fractions in the latter blots lack a cytosolic loading control, but all the conclusions from these blots are based on nuclear expression and the cytosolic fraction is only shown for completeness. We also include the whole gel images for immunoblots and Ponceau in the Source Data files so the entire lane, not just a small part, of the Ponceau stained membranes can be evaluated by the readers.

4) The investigators propose that *Glis2* is a quite direct and proximal marker of *Pkd1* or *Pkd2* inactivation, as the nuclear translocation seems to be quite robustly and rapidly occurring after inactivation of the genes. How would this occur? Is *Glis2* directly binding the Polycystins or what could be an alternative explanation?

We propose that *Glis2* is a tightly correlated marker of *Pkd1* and *Pkd2* inactivation, but this is not meant to imply that this is a direct or even proximal target of polycystins. We suspect that the connection, while specific, is likely indirect, with several unknown more proximal intervening steps resulting in increased transcription of *Glis2*. *Glis2* is not in cilia, so the polycystin effect is not likely to directly impact *Glis2* in this compartment. We do not have evidence of direct interaction of *Glis2* with either polycystin outside the cilia. Finally, heterologous, *Pkd1*-independent over-expression of *Glis2* (e.g. Supplementary Figure 8b) results in strong nuclear localization, so the *Pkd1* effect may be primarily to increase steady state transcriptional and posttranscriptional levels of *Glis2* with the attendant nuclear localization being an inherent property of *Glis2* itself. The steps coupling loss of polycystins with increased transcription of *Glis2* are unknown but are of great interest if they can be discovered.

Minor:

Figure 3m is missing

We confirmed that Figure 3m is present.

In sum, this is an extensive study which, as the authors state, fills several gaps in the ADPKD field. And, therefore, it certainly deserves publication in a prominent journal.

Reviewer #4 (Remarks to the Author):

*Zhang and colleagues present an experimental study on the role of *Glis2* in cilia dependent cyst activation in PKD. The study consists of two main parts. First, the authors use TRAP to identify novel differentially regulated genes between mice with cyst formation +/- cilia. They use three genotypes: *Pkd1*KO, *Pkd1*KO+ciliaKO 23 and *Pkd1* heterozygous mice. The chosen strategy is elegant, comprehensive and state-of-the art. Results of the unbiased translomic approach are made available by the authors, which will be very helpful for the PKD field to screen for specific pathways. Among the identified DEGs the authors selected *Glis2* as an attractive candidate gene. In the second part of the paper, the authors focus on *Glis2* function in vitro and in vivo. They very conclusively demonstrate the involvement of *Glis2* as an effector of PKD-dependent cyst formation. The data is very clean and the line of evidence is convincing. The experiments are performed and presented on a very high level. The amount of data is impressive and the paper is nicely written. There are only few points I would like to mention:*

1. Reference 52 by the Attanasio group has already shown that loss of *Glis2* suppresses cyst growth in the *Kif3a* ko mouse. This previous publication does not make the current results less important, but given the parallel observations, Reference 52 should be cited and discussed accordingly with respect to the known effects of *Glis2* on cyst formation/growth.

We thank the reviewer for this suggestion. We have added this to the discussion as follows:

“Interestingly, *Glis2* inactivation suppresses cyst formation due to loss of cilia in an early onset model of kidney selective inactivation of *Kif3a*⁵³. While the cadence and mechanisms of cyst formation due to cilia inactivation differ from polycystic kidney disease due to inactivation of *Pkd1* or *Pkd2* [19], this observation does raise the intriguing possibility that *Glis2* may be a common downstream effector for both mechanisms. Such a hypothesis raises the possibility that transcriptional targets of *Glis2* are common elements of kidney cyst formation from diverse causes.”

2. The authors identified a possible concern about potential senescence-inducing effects of knocking down *Glis2* (based on Ref 52). To exclude increased accumulation and development of cellular senescence they stained for SA-beta-Galactosidase activity. SA-b-Gal is one of many non-specific markers of cellular senescence. It simply detects a higher lysosomal galactosidase activity, which is also seen in many non-senescent conditions. Therefore, there are clear recommendations that SA-b-Gal has to be combined with other markers of senescence (e.g. *Cdkn2a* and *Cdkn1a* on transcriptional level, reduced lamin b1 staining, enhanced DNA damage markers, absence of proliferation markers, etc.). Please provide additional markers or alternatively refrain calling the observed state cellular senescence if SA-b-Gal is used alone (increased lysosomal activity could be an alternative terminology).

We have added quantitative RT-PCR for *Cdkn1a* and *Cdkn2a* to Supplementary Figure 18, panels f, g. This new data also shows absence of senescence following conditional inactivation of *Glis2* in adult kidney; we also confirmed the previously reported changes in germline *Glis2*^{-/-} kidneys that serve as positive control in our studies. We used positive staining in *Glis2*^{-/-} kidneys as a positive control based on the reference cited (PMID: 27181777). As the reviewer points out, SA-β-Gal positive staining is not specific for marking senescent cells in tissues. However, absence of SA-β-Gal is fairly effective in excluding senescence in the tissue sample, which was our point. In other parts of the results text, we have taken the reviewers suggestion and were more careful in how we described the sporadic positive SA-β-gal staining seen in a few scattered cysts (not attributing this to senescence). We interpreted absence of SA-β-Gal, coupled with not change in qRT-PCR for *Cdkn1a* and *Cdkn2a* in the adult inactivation of *Glis2* and *Glis2*-ASO treated models to show absence of senescence.

3. Is anything known about the expression pattern of *Glis2* in human PKD? E.g. from scRNAseq (Nat Commun. 2022; 13: 6497.)? This would be interesting from a translational standpoint.

This same data is part of the responses to Reviewers 1 and 2 as well:

In 2009, Song et al. (PMID: 19346236) performed microarray on human ADPKD cyst samples. They observed significantly increased expression of *GLIS2* in more cystic disease (Figure at left). In addition, several other genes in the TRAP gene list from manuscript Figure 1d showed relative changes in expression that correlated with cyst severity in the human studies (Figure at left). Specifically, in addition to *GLIS2*, *PKD2*, *PTPDC1*, *TSPAN5* also showed upregulation in the human studies and *LAD1* showed downregulation.

In 2022, Muto et al. (PMID: 36310237) performed snRNAseq from 5 healthy kidneys and 8 ADPKD patient kidneys. Authors also reported GPRC5A as a marker of ADPKD kidneys. They reported that the cyst lining cells in their sample were in the connecting tubule and principal cell cluster (CNT_PC). We re-analyzed their snRNAseq dataset (GSE185948) to examine GLIS2 expression which was not reported in per-cell gene expression analysis in the original paper. We used the authors' processing code on GitHub (https://github.com/TheHumphreysLab/Multimodal_analysis_ADPKD) to reproduce the UMAP plots as published (not shown). Table 1 at left shows the number of cells with expression of GLIS2 and GPRC5A in the CNT_PC cluster in control and PKD samples. We used an adjusted count threshold of >1 for gene expression detection to count cells, and we used GPRC5A as a positive control for per-cell

Table 1: Cell numbers in segment			Table 2: CPM Normalized Gene Expression Values		
Patient type	CNT_PC		Patient type	CNT_PC	
	GLIS2	GPRC5A		GLIS2	GPRC5A
control1	1	13	control1	50.4	63.4
control2	4	6	control2	50.3	55.9
control3	1	3	control3	57.5	54.8
control4	10	18	control4	56.3	57.4
control5	0	2	control5	56.2	52.2
PKD1	7	420	PKD1	64.4	101.5
PKD2	6	107	PKD2	56.7	74.4
PKD3	15	392	PKD3	68.7	97.4
PKD4	2	249	PKD4	62.7	107.9
PKD5	3	53	PKD5	66.7	84.6
PKD6	17	236	PKD6	53.8	67.2
PKD7	1	4	PKD7	55.6	78.8
PKD8	6	638	PKD8	60.6	107.4
control_average	3.2	8.4	control_average	54.1	56.7
PKD_average	7.1	262.4	PKD_average	61.2	89.9

expression analysis. On average, there was a ~30-fold increase in the cells expressing GPRC5A in the PKD samples compared to control; there was a two-fold increase in cells expressing GLIS2 in the PKD samples. Given the small absolute number of cells expressing GLIS2, we were not able to determine per-cell expression for GLIS2 as was done for GPRC5A in the paper. We therefore used a pseudobulk strategy with the DEGseq2 package to assess GLIS2 expression in the PKD and control groups. Pseudobulk analysis refers to the use of single-cell expression data to compute average gene expression in a cluster of cells and thereby simulate bulk gene expression (PMID: 34321199, PMID: 36550119,

PMID: 35361816, PMID: 35864314, PMID: 37400500); for context, our TRAP-RNASeq is a cell type specific bulk RNASeq approach. Table 2 shows the counts per million (CPM) normalized pseudobulk expression levels of GLIS2 and GPRC5A. Using pseudobulk analysis, GPRC5A shows significant increase in PKD samples compared to control, in line with the published data of Muto et al. GLIS2 average CPM is 61.2 in PKD vs 54.1 in controls, but these averages are based on far fewer cells so pAdj is not significant. Still, there is higher average expression in a larger number of cells for GLIS2 in the PKD samples than in the controls. All we can say is that GLIS2 shows a trend towards increased expression in the PKD cell cluster that includes cells of cyst origin. These results, while not conclusive, are at least "permissive" for translation of our findings in mouse models to human samples.

4. A challenging aspect of the study is the impact of sex (in particular difference between male and female mouse TRAP data). It is generally reasonable and important to elucidate sex differences in PKD disease models. However, in the current study the added value of female mice is not really clear. In the human PKD field sex differences are not as relevant. While it might be an advantage to use female mice for a less stringent PKD model the authors should describe the background and advantages in more detail for researchers less familiar with this specific model.

The sex difference in progression in *Pkd1* mice was first reported by Menzes et al., 2016 (PMID: 27077126) in a different adult inducible model, so our data confirms this phenomenon with the *Pax8^{rtTA}* model. Our mice strains are all >95% C57/BL6J. We believe both sexes develop cysts but the females do so more slowly, so the differences are really in how fast those cysts develop, not in whether they develop cysts.

Our goal in this report is to provide the most constrained list of high certainty transcriptome changes related to ADPKD. To identify the highest confidence CDCA pattern changes, we focused on TRAP DEG that were in common between male and female mice, with the understanding that the male CDCA pattern DEG would capture more true signals, but also more false positive signals. We have high confidence in the DEG we present in the TRAP but acknowledge the possibility that there are other genes that respond in the CDCA pattern, but which are excluded from our core list due to the stringency of our criteria. This is described with the following text on page 5, lines 28-34:

"We chose to focus on DEGs shared by male and female mice as a more stringent selection criterion to provide the highest confidence for definition of the 'CDCA pattern' transcriptional changes. This does not, however, preclude potential functional importance for other genes, e.g., those showing significant 'CDCA

pattern' change only in male mice. Nonetheless, we hypothesize that this 167 gene set has the highest likelihood of containing subsets of genes that define a transcriptional 'signature' for polycystin loss in vivo. This gene set likely also contains genes whose dysregulation is functionally related to in vivo cyst progression in ADPKD."

In our view, if one wanted a more expanded set, then the male-only CDCA pattern TRAP-DEG would be the better starting set since we know the male mice are in the "incipient" cyst state and should show more of the changes than the females at 7 weeks; these changes may emerge in female mice as they progress.

5. It was not the focus of the study to elucidate the mechanistic downstream effects of Glis2 (how is the effector working?). However, it would be desirable to obtain at least some speculative explanation on the effector pathway of Glis2, which is responsible for the observed differences.

As discussed under Reviewer 3 (item #2), we would like to resist the temptation to speculate on the downstream effects. Glis2 is a transcription factor, and its upregulation supports cyst progression whereas inactivation suppresses cyst progression. We can speculate that the transcriptional targets of Glis2 specifically in the context of its action following inactivation of polycystins are part of the machinery that drives cyst progression. As noted in earlier comment to this Reviewer (item #1), the transcriptional targets of Glis2 may have broader implications for kidney health. Furthermore, if Glis2 functions primarily a transcriptional repressor, then the downstream mechanisms promoting cyst formation may be the result of repression of critical factors maintaining the normal state of the kidney—these may prove to be important biologically but may be difficult from a drug development standpoint since they will require activators to suppress cysts downstream of Glis2. We believe that defining these potentially impactful targets specifically in the context of polycystin inactivation will be an important next step in furthering understanding of mechanism in ADPKD. This discussion, however, is too speculative to put in the manuscript, but we did add some general statements to raise these points in the discussion.

REVIEWER COMMENTS

Reviewer #2 (Remarks to the Author):

Dear author,
all my comments have been considered and concerns satisfactorily met. Thank you.

Reviewer #3 (Remarks to the Author):

Unfortunately, the authors were unable to address two of my requests. They were not able to provide evidence that in the mouse model utilized all cells carrying the Cre and acquiring green ribosomes after induction will eventually end up forming cysts. Their argument based on previous data and other indirect evidence is only partially convincing. But the implications of not knowing this with certainty are not such to jeopardize the importance of the results.

They were also unable to show that a few known targets of Glis2 could be identified as being upregulated in the PKD mutants and rescued in the double mutants, hence providing initial evidence that Glis2 transcriptional activity aligns with its expression, the last convincingly shown to mediate the effects on phenotype. While it is appreciable that they are already placing their efforts in defining the targets of this transcription factor in future studies, it is highly unlikely that the targets will be so uniquely specific to the renal tissue studied here. Again, this was a suggestion that would have strengthened the results but would not change the conclusions.

Based on this, and as stated in the previous round of revision, this is a strong and important study, deserving rapid publication.

Reviewer #4 (Remarks to the Author):

My questions have been adequately addressed.

Reviewer #5 (Remarks to the Author):

We would like to thank you for the opportunity to review this manuscript. In addition to our comments on Reviewer 1's (#R1) comments and the author's reply, we here provide our outstanding comments on the manuscript.

Remarks to the Author

In the manuscript "Glis2 is an early effector of polycystin signaling and a target for therapy in polycystic kidney disease", Zhang et al. investigate Glis2 as one of the previously unknown components of the cilia-dependent cyst activation (CDCA). CDCA refers to the phenomenon that signaling that is dependent on the presence of intact primary cilia leads to cyst formation in mouse models of autosomal dominant polycystic kidney disease (ADPKD). Using unbiased transcriptional profiling in the form of translating ribosome affinity purification (TRAP) in kidney tissue, Zhang et al. identify 167 differentially expressed mRNAs associated with CDCA in both male and female mice. Among these, Glis2 was identified as a potential effector of polycystin signaling and CDCA. Validating Glis2 as an assay for polycystin function, its genetic inactivation in mouse kidneys suppressed polycystic kidney disease in ADPKD models based on *Pkd1* and *Pkd2*. Targeting Glis2 with antisense oligonucleotides suppressed polycystic kidney disease in a mouse model of ADPKD based on *Pkd1*. The study demonstrates that increased Glis2 transcript and protein levels indicate a loss of polycystin function and suggests Glis2 as a therapeutic target for treating polycystic kidney disease.

The manuscript presents a compelling exploration by identifying differentially expressed mRNAs in an ADPKD model. The strategic choice of a unique time point after *Pkd1* knockout but before cyst manifestation offers an insightful opportunity to unveil crucial players in cystogenesis. The approach of comparing the *Pkd1* knockout with both a non-cystic model and a *Pkd1* and cilia double knockout model is commendable. By identifying mRNAs showing consistent regulation in both comparisons, the study aims to discern factors crucial for cyst formation, attributing a cilia-dependent role to these genes due to alteration in their expression in the presence of cilia.

The specific emphasis on *Glis2* as a differentially regulated gene provides a targeted avenue for subsequent investigation, potentially yielding valuable insights into its role in cystogenesis within the context of ADPKD. However, tempering the description of *Glis2* as the definitive Pkd effector and the incorporation of mechanistic data would contribute to the reader's understanding of *Glis2* as (one of the) downstream effector(s). Particularly noteworthy is the validation of data in various ADPKD mouse models and the identification of *Glis2* as a potential therapeutic target.

Overall, the study is well-designed, and the especially the dataset of differentially regulated genes contributes substantially to the PKD community's knowledge by filling several gaps.

Major comments

1. We appreciate the newly identified and previously underestimated significance of *Glis2* shown by this manuscript. Simultaneously, a minor weakness arises from the absence of functional data on how *Glis2* contributes to cyst formation, also in consideration of its prior association with PKD [e.g. PMID: 31773180] and the title, which implies a focus on *Glis2* as a PC effector ("Glis2 is an early effector of polycystin signaling"). An investigation of the relationship between *Glis2* and, for example, the hedgehog signaling pathway [PMID: 21816948] would be of particular interest to the reader and would contribute to the clarity of the presented concept of *Glis2* as an effector of polycystin signaling.
2. The differences between the blots in Figure 5 g and i are not clear to us. In both representations, both bands are labeled as TNF-alpha and cleaved caspase-1, respectively. While the inserted marker suggests a slight height difference between the bands, the bands on the original membrane, according to the marker, appear to be at the same level. We kindly request clarification regarding whether the same membrane is shown in i and g and an explanation of what exactly the bands represent.
3. Figure 1c: We are wondering whether the numbers of DEGs in the first two volcano plots of male and female mice are the same or whether the numbers need to be corrected.

Minor comments

1. The definition of the CDCA represents an abstract term that, in our understanding, serves the PKD community more as a working hypothesis for identifying the cystogenic signal originating from cilia. A consistent definition maintained throughout the manuscript would facilitate the reader's comprehension of the concept. For instance, the definition used in the Abstract could be made more specific from our perspective, for example, by replacing it with “a putative ciliary signal that drives cyst growth when PC function is compromised” [PMID: 36805211]. Another reason why the CDCA remains abstract for the reader even after reading the manuscript is the absence of a (possibly graphical) representation of the outlined concept that contextualizes *Glis2* as a component of the CDCA. Here, the lack of functional data, which renders such a presentation relatively speculative, represents a limitation.
2. The aspect of different signaling pathways underlying cyst formation due to the loss of ciliary function versus the loss of polycystins is particularly interesting, considering that, in our understanding, *Glis2* appears to be a component involved in signaling in both cases. The prevention of cyst formation by knocking out *Glis2* within a *Kif3a* knockout model [PMID: 27181777] suggests an overlap between the CDCA signaling pathway and the pathway relevant in ciliary knockout. This, combined with the absent ciliary localization of *Glis2*, could indicate its role as a downstream effector. While the authors mention the data regarding *Glis2* in the *Kif3a* model in the discussion, a more detailed discussion of this connection would be interesting. However, as mentioned above, due to the lack of functional data, it might understandably be too speculative to provide such a discussion.
3. Correction and uniformity in several text passages would enhance the readability. (Of course, these aspects are redundant if they have already been addressed as part of the correction).
 - The use of the central term “CDCA pattern” is inconsistent.
 - o CDCA pattern: sometimes with ` (e.g. p6 vs p7), sometimes with – (e.g. p5, l26), sometimes activation itself, sometimes cilia-dependent cyst activating (CDCA) pathway (p3)
 - Part of the citations within the text are written in [] or [[(e.g. p6, l26)
 - The caption of the figures is inconsistent.
 - o “Figure 1:” versus “Figure 2.”
 - o Use of bold letters as a reference to the figures is inconsistent (e.g. p6, l30)
 - o Use of (bold) commas in the caption of all the figures is inconsistent (e.g. p18, l19), (**e,g**) (p20, l15), m.n, (p.22, l17)
 - o Figure 2a: (genotypes and) timepoints on the x-axis would be consistent with the labeling of the other graphs and would increase readability
 - o The captions are cumbersome to read. It would be helpful if, for instance, repetitions are avoided (e.g. “All mice have x in addition to..” on p.20, l20 and l23)
 - o With the expression “i, * is non-specific band” maybe “k, * is non-specific band” is meant on p.20, l20?
 - o Mouse model (instead of models) (p24, l1)
 - o At least one section [...] **was** examined (p24, l22)
 - o Labeling of the red and blue bar is missing in Figure 5 k
 - Cells destined **to** form cysts (p3, l35)
 - Dot is missing behind “has been described” (p4, l21)
 - Signaling that results in (p4, l27)
 - Are the first molecular marker shown to (instead of **to** shown to) (p8, l32)
 - Set of [...] represents (p13, l29)
 - With TPM >1.0 (e.g. p13, l31)
 - Website (instead of web site) (p13, l14)
 - Interventions that suppress [...], similarly suppress (instead of suppresses) (p15, l5) increased expression **of** *Glis2* in vitro (p15, l5)

Response to REVIEWER COMMENTS

Reviewer #2 (Remarks to the Author):

All my comments have been considered and concerns satisfactorily met. Thank you.

We thank the reviewer for their careful evaluation of this manuscript and for helping us improve the presentation.

Reviewer #3 (Remarks to the Author):

Unfortunately, the authors were unable to address two of my requests. They were not able to provide evidence that in the mouse model utilized all cells carrying the Cre and acquiring green ribosomes after induction will eventually end up forming cysts. Their argument based on previous data and other indirect evidence is only partially convincing. But the implications of not knowing this with certainty are not such to jeopardize the importance of the results.

They were also unable to show that a few known targets of Glis2 could be identified as being upregulated in the PKD mutants and rescued in the double mutants, hence providing initial evidence that Glis2 transcriptional activity aligns with its expression, the last convincingly shown to mediate the effects on phenotype. While it is appreciable that they are already placing their efforts in defining the targets of this transcription factor in future studies, it is highly unlikely that the targets will be so uniquely specific to the renal tissue studied here. Again, this was a suggestion that would have strengthened the results but would not change the conclusions.

Based on this, and as stated in the previous round of revision, this is a strong and important study, deserving rapid publication.

We thank the reviewer for their careful evaluation of this manuscript and their support of this effort despite the above limitations.

Reviewer #4 (Remarks to the Author):

My questions have been adequately addressed.

We thank the reviewer for their careful evaluation of this manuscript and for helping us improve the presentation.

Reviewer #5 (Remarks to the Author):

We would like to thank you for the opportunity to review this manuscript. In addition to our comments on Reviewer 1's (#R1) comments and the author's reply, we here provide our outstanding comments on the manuscript.

Remarks to the Author

In the manuscript "Glis2 is an early effector of polycystin signaling and a target for therapy in polycystic kidney disease", Zhang et al. investigate Glis2 as one of the previously unknown components of the cilia-dependent cyst activation (CDCA). CDCA refers to the phenomenon that signaling that is dependent on the presence of intact primary cilia leads to cyst formation in mouse models of autosomal dominant polycystic kidney disease (ADPKD). Using unbiased transcriptional profiling in the form of translating ribosome affinity purification (TRAP) in kidney tissue, Zhang et al. identify 167 differentially expressed mRNAs associated with CDCA in both male and female mice. Among these, Glis2 was identified as a potential effector of polycystin signaling and CDCA. Validating Glis2 as an assay for polycystin function, its genetic inactivation in mouse kidneys

suppressed polycystic kidney disease in ADPKD models based on Pkd1 and Pkd2. Targeting Glis2 with antisense oligonucleotides suppressed polycystic kidney disease in a mouse model of ADPKD based on Pkd1. The study demonstrates that increased Glis2 transcript and protein levels indicate a loss of polycystin function and suggests Glis2 as a therapeutic target for treating polycystic kidney disease.

The manuscript presents a compelling exploration by identifying differentially expressed mRNAs in an ADPKD model. The strategic choice of a unique time point after Pkd1 knockout but before cyst manifestation offers an insightful opportunity to unveil crucial players in cystogenesis. The approach of comparing the Pkd1 knockout with both a non-cystic model and a Pkd1 and cilia double knockout model is commendable. By identifying mRNAs showing consistent regulation in both comparisons, the study aims to discern factors crucial for cyst formation, attributing a cilia-dependent role to these genes due to alteration in their expression in the presence of cilia.

The specific emphasis on Glis2 as a differentially regulated gene provides a targeted avenue for subsequent investigation, potentially yielding valuable insights into its role in cystogenesis within the context of ADPKD. However, tempering the description of Glis2 as the definitive Pkd effector and the incorporation of mechanistic data would contribute to the reader's understanding of Glis2 as (one of the) downstream effector(s).

Particularly noteworthy is the validation of data in various ADPKD mouse models and the identification of Glis2 as a potential therapeutic target.

Overall, the study is well-designed, and the especially the dataset of differentially regulated genes contributes substantially to the PKD community's knowledge by filling several gaps.

We thank the reviewer for their very thoughtful comments on the manuscript.

Major comments

1. *We appreciate the newly identified and previously underestimated significance of Glis2 shown by this manuscript. Simultaneously, a minor weakness arises from the absence of functional data on how Glis2 contributes to cyst formation, also in consideration of its prior association with PKD [e.g. PMID: 31773180] and the title, which implies a focus on Glis2 as a PC effector ("Glis2 is an early effector of polycystin signaling"). An investigation of the relationship between Glis2 and, for example, the hedgehog signaling pathway [PMID: 21816948] would be of particular interest to the reader and would contribute to the clarity of the presented concept of Glis2 as an effector of polycystin signaling.*

We have added the citation PMID: 31773180 that showed that *Glis2* was one of 92 transcription factors upregulated amongst 1515 gene transcripts proposed as a PKD Signature in an inducible Ksp model of *Pkd1*. In that study, *Glis2* was the transcription factor most correlated with the PKD genotype, although the authors did not investigate it further, highlighting the challenges of working with very broad gene lists.

We have previously published that the Hedgehog (Hh) pathway has no role in ADPKD cyst progression in orthologous models (PMID: 31451534). We therefore opted not to discuss putative roles of *Glis2* in Hh signaling since such roles are unlikely to be related to its effector role in polycystin dependent signaling. We now cite this work in the discussion and explicitly state that its putative function in the Hh pathway is unlikely to be linked to the role of *Glis2* in ADPKD pathogenesis.

2. *The differences between the blots in Figure 5 g and i are not clear to us. In both representations, both bands are labeled as TNF-alpha and cleaved caspase-1, respectively. While the inserted marker suggests a slight height difference between the bands, the bands on the original membrane, according to the marker, appear to be at the same level. We kindly request clarification regarding whether the same membrane is shown in i and g and an explanation of what exactly the bands represent.*

First, a correction. The labeling of the marker positions on TNF α panel in Figure 5g were placed incorrectly and have now been corrected to properly match the source data provided with the manuscript.

Regarding the similarity between the two blots, we shared the reviewer's concern when we first used these two antibodies in our previous publication on polycystin gene re-expression (PMID: 34635846) in Figure 5e,g,i,k (and the associated source data). At that time, we verified that the primary antibodies were not confused—they came from different tubes from different vendors—and yet gave very similar migration and band pattern. We repeated the blots at that time to verify the outcome.

For the data we present in this manuscript, the two blots are different, and they were not re-probed but rather were run in parallel. One can see, for example, that the marker lanes (rightmost lanes) on the TNF α and cleaved caspase-1 blots appear different. Similarly, in the Hsp90 blots, there are differences in the curvature of some of the lanes. We affirm that these are different blots probed with different commercial antibodies. We ran multiple iterations of these immunoblots using the same 9 biological samples to get the best quality blots for the final figure, so we provide images of another iteration of these immunoblots for the reviewer's and editors' reference:

3. Figure 1c: We are wondering whether the numbers of DEGs in the first two volcano plots of male and female mice are the same or whether the numbers need to be corrected.

The numbers are correct as presented. We wanted to show two pieces of information:

1. the number genes meeting our criterion of significant “same direction change” in Pkd1^{KO} compared to both noncystic and Pkd1^{KO}+cilia^{KO} mice and
2. the number of genes within this group for each direction change (up and down).

Figure 1b gives the numerical information for item 1 (total number with same direction change; 440 for male mice, 526 for female mice), but information regarding how many are up and how many are down is lost in this aggregate data. So is the information that most of these same genes did not differ in expression between noncystic and Pkd1^{KO}+cilia^{KO} mice. We therefore used the volcano plots in Figure 1c to provide the latter information both visually and numerically.

We will use the first volcano plots in the male group to illustrate what we are showing:

Based on the data in Figure 1b, 440 genes showed the same direction change in male Pkd1^{KO} mice compared to both noncystic and Pkd1^{KO}+cilia^{KO} mice. So, by definition, the total numbers of up and down genes in both comparisons must be the same and add up to 440. Put another way, 177 genes met our criteria for being significantly upregulated in the Pkd1^{KO} vs. noncystic group and in the Pkd1^{KO} vs. Pkd1^{KO}+cilia^{KO} group and 283 met the criterion and were downregulated. The rightmost panel shows that most of these 440 genes were not significantly different between the noncystic and Pkd1^{KO}+cilia^{KO} groups, which we considered phenotypically similar as far as ADPKD is concerned (i.e., neither are destined to form cysts).

We have reviewed the figure legend in an effort to be as clear as possible.

Minor comments

1. The definition of the CDCA represents an abstract term that, in our understanding, serves the PKD community more as a working hypothesis for identifying the cystogenic signal originating from cilia. A consistent definition maintained throughout the manuscript would facilitate the reader's comprehension of the concept. For instance, the definition used in the Abstract could be made more specific from our perspective, for example, by replacing it with "a putative ciliary signal that drives cyst growth when PC function is compromised" [PMID: 36805211]. Another reason why the CDCA remains abstract for the reader even after reading the manuscript is the absence of a (possibly graphical) representation of the outlined concept that contextualizes *Glis2* as a component of the CDCA. Here, the lack of functional data, which renders such a presentation relatively speculative, represents a limitation.

We appreciate the reviewer's guidance. We have deliberately tried to avoid being specific with the definition of CDCA because we do not have sufficient information. We like (and agree) with the reviewer's suggested definition as "a putative ciliary signal that drives cyst growth when PC function is compromised" [PMID: 36805211]. Our hesitation stems from the fact that there is a subtle difference between saying CDCA requires intact cilia (which we have shown) and saying that it is "a putative ciliary signal," which we have not shown. As a reasonable compromise, we have reworded the opening sentences of the abstract as follows:

Studies in mouse models of autosomal dominant polycystic kidney disease (ADPKD) have shown that putative signaling pathways requiring structurally and functionally intact primary cilia drive cyst growth following inactivation of polycystin-1 or polycystin-2. The molecular components of these pathways, which we have termed cilia dependent cyst activation (CDCA), are unknown.

We have also made some changes in the text to be more declarative in describing CDCA.

2. The aspect of different signaling pathways underlying cyst formation due to the loss of ciliary function versus the loss of polycystins is particularly interesting, considering that, in our understanding, *Glis2* appears to be a component involved in signaling in both cases. The prevention of cyst formation by knocking out *Glis2* within a *Kif3a* knockout model [PMID: 27181777] suggests an overlap between the CDCA signaling pathway and the pathway relevant in ciliary knockout. This, combined with the absent ciliary localization of *Glis2*, could indicate its role as a downstream effector. While the authors mention the data regarding *Glis2* in the *Kif3a* model in the discussion, a more detailed discussion of this connection would be interesting. However, as mentioned above, due to the lack of functional data, it might understandably be too speculative to provide such a discussion.

We agree with all the reviewer's points including the concern that such a discussion may be too speculative to be informative. In theory, intact cilia drive cyst growth through CDCA and PCs are in cilia to regulate CDCA as a physiological response to stimuli, rather than an unchecked maladaptive process as occurs in ADPKD. Loss of cilia may remove the 'brake' function on CDCA since PCs outside of cilia do not function to regulate CDCA. The absence of cilia may result in a low-level activity

of cellular effectors of CDCA (hence much slower cyst growth) due to the loss of regulation of downstream effectors, but without the strong driving signal that CDCA emanating from intact cilia offers. Glis2 may be an effector of both the high amplitude signal in ADPKD (intact cilia, absent PC, positive CDCA input) and the low amplitude signal that occurs when cilia are lost (absent cilia, present PC, absent regulation but no CDCA driver input). We are concerned that this type of discussion is a little too speculative and is better suited to a broader opinion piece on CDCA than the current manuscript. That said, we have a brief version of this discussion to offer a potential unifying reconciliation for the cyst suppressive effects of Glis2 in ADPKD and in Kif3a knockouts.

3. Correction and uniformity in several text passages would enhance the readability. (Of course, these aspects are redundant if they have already been addressed as part of the correction).

- *The use of the central term “CDCA pattern” is inconsistent.*

o *CDCA pattern: sometimes with ` (e.g. p6 vs p7), sometimes with – (e.g. p5, l26), sometimes activation itself, sometimes cilia-dependent cyst activating (CDCA) pathway (p3)*

We have made these all consistent using ‘CDCA pattern.’

- *Part of the citations within the text are written in [] or [] (e.g. p6, l26)*

[] are used when citations follow immediately after either a number or a superscript in the text to avoid confusion. We have made the absence of a space before the brackets consistent throughout the manuscript.

- *The caption of the figures is inconsistent.*

o *“Figure 1:” versus “Figure 2.”*

We have made this all consistent using a period.

o *Use of bold letters as a reference to the figures is inconsistent (e.g. p6, l30)*

We will defer to final editing on this. We use bold letters for panels labeled within Figures, but do not use bold letters in Table file names (e.g., 2a, 2b).

o *Use of (bold) commas in the caption of all the figures is inconsistent (e.g. p18, l19), (e,g) (p20, l15), m.n, (p.22, l17)*

We have made all commas in the figure legends not bold.

o *Figure 2a: (genotypes and) timepoints on the x-axis would be consistent with the labeling of the other graphs and would increase readability*

We have updated the labeling as suggested. We have kept the color legend insert to ensure clarity.

o *The captions are cumbersome to read. It would be helpful if, for instance, repetitions are avoided (e.g. “All mice have x in addition to..” on p.20, l20 and l23)*

We have edited the legends for Figures 1-5 to improve the efficiency and clarity of presentation.

o *With the expression “i, * is non-specific band” maybe “k, * is non-specific band” is meant on p.20, l20?*

The non-specific band * was meant for both Figure 2i,k. We have updated the legend accordingly.

o *Mouse model (instead of models) (p24, l1)*

We have corrected this.

- o *At least one section [...] was examined (p24, l22)*

We have corrected this.

- o *Labeling of the red and blue bar is missing in Figure 5 k*

We have corrected this.

- *Cells destined **to** form cysts (p3, l35)*

We have corrected this.

- *Dot is missing behind “has been described” (p4, l21)*

We have corrected this.

- *Signaling that results in (p4, l27)*

We are not sure about this change. Our reading suggests that “result” is modifying “other tissue level responses” which is plural. That said, we have rewritten this sentence in an effort to improve its overall structure.

- *Are the first molecular marker shown to (instead of **to** shown to) (p8, l32)*

We have corrected this.

- *Set of [...] represents (p13, l29)*

We have corrected this.

- *With TPM >1.0 (e.g. p13, l31)*

We have corrected this.

- *Website (instead of web site) (p13, l14)*

We have corrected this.

- *Interventions that suppress [...], similarly suppress (instead of suppresses) (p15, l5) increased expression **of** Glis2 in vitro (p15, l5)*

We have corrected this.

REVIEWERS' COMMENTS

Reviewer #5 (Remarks to the Author):

Dear Authors,

our comments have been addressed appropriately.

Thank you!